



# Diurnal, synoptic and seasonal variability of atmospheric $CO_2$ in the Paris megacity area

Irène Xueref-Remy[1], Elsa Dieudonné[1,2], Cyrille Vuillemin[1,3], Morgan Lopez[1,4], Christine Lac[5], Martina Schmidt[1,6], Marc Delmotte[1], Frédéric Chevallier[1], François Ravetta[7], Olivier Perrussel[9], Philippe Ciais[1], François-Marie Bréon[1], Grégoire Broquet[1], Michel Ramonet[1], T. Gerard Spain[8] and Christophe Ampe[9]

[1]Laboratoire des Sciences du Climat et de l'Environnement (LSCE), Gif-sur-Yvette, France
[2]Now at Laboratoire de Physico-chimie de l'Atmosphère (LPA), Dunkerque, France
[3]Now at European Organization for Nuclear Research (CERN), Meyrin, Switzerland
[4]Now at Environment Canada, Climate Research Division, Toronto, Ontario, Canada
[5]Centre National de la Recherche Météorologique (CNRM-GAME), Toulouse, France
[6]Now at Institute of Environmental Physics (IEP), Heidelberg, Germany
[7]Laboratoire Atmosphères, Milieux, Observations Spatiales (LATMOS), Guyancourt, France
[8]National University of Ireland (NUI), Galway, Ireland
[9]Association de Surveillance de la Qualité de l'Air en Île-de-France (AIRPARIF), Paris, France

*Correspondance to:* Irène Xueref-Remy (irene.xueref@lsce.ipsl.fr)

**Abstract.** Most of the global fossil fuel $CO_2$ emissions arise out of urbanized and industrialized areas. Bottom-up inventories quantify them but with large uncertainties. In 2010-2011, the first atmospheric in-situ $CO_2$ measurement network for Paris, the capital of France, has been operated with the aim of monitoring the regional atmospheric impact of the emissions out coming from this megacity. Five stations sampled air along a northeast-southwest axis that corresponds to the direction of the dominant winds. Two stations are classified as rural (TRN and MON), two are peri-urban (GON and GIF) and one is urban (EIF, located on top of the Eiffel tower). In this study, we analyze the diurnal, synoptic and seasonal variability of the in-situ $CO_2$ measurements over nearly one year (8 August 2010–13 July 2011). We compare these datasets with remote $CO_2$ measurements made at Mace Head (MHD) on the Atlantic coast of Ireland, and support our analysis with atmospheric boundary layer height (ABLH) observations made in the centre of Paris and with both modeled and observed meteorological fields. The average hourly $CO_2$ diurnal cycles observed at the regional stations are mostly driven by the $CO_2$ biospheric cycle, the ABLH cycle, and the proximity to urban $CO_2$ emissions. Differences of several μmol·mol⁻¹ (ppm) can be observed from one regional site to the other. The more the site is surrounded by urban sources (mostly traffic, residential and commercial heating), the more the $CO_2$ concentration is elevated, as is the associated variability which reflects the variability of the urban sources. Furthermore, two elevated sites (EIF and TRN) show a phase shift of the $CO_2$ diurnal cycle of a few hours compared to lower sites due to a strong coupling with the boundary layer diurnal cycle. As a consequence, the existence of a $CO_2$ vertical gradient above Paris can be inferred, whose amplitude depends on the time of the day and on the





season, ranging from a few tenths of ppm during daytime to several ppm during nighttime. The $CO_2$ seasonal cycle inferred from monthly means at our regional sites are driven by the biospheric and anthropogenic $CO_2$ flux seasonal cycles, by the ABLH seasonal cycle and also by synoptic variations. Gradients of several ppm are observed between the rural and peri-urban stations, mostly from the influence of urban emissions that are in the footprint of the peri-urban station. The seasonal

cycle observed at the urban station (EIF) is specific and very sensitive to the ABLH cycle. At both the diurnal and the seasonal scales, noticeable differences of several ppm can be observed between the measurements made at regional rural stations and the remote measurements made at MHD, that are shown not to define background concentrations appropriately for quantifying the regional atmospheric impact of urban $CO_2$ emissions. For wind speeds less than 3 m s$^{-1}$, the accumulation of the local $CO_2$ emissions in the urban atmosphere forms a dome of several tens of ppm at the peri-urban stations, mostly

under the influence of relatively local emissions including those from the Charles-De-Gaulle (CDG) airport facility and from aircrafts in flight. When wind speed increases, ventilation transforms the $CO_2$ dome into a plume. Higher $CO_2$ background concentrations of several ppm are advected from the remote Benelux-Ruhr and London regions, impacting concentrations at the five stations of the network even at wind speeds higher than 9 m s$^{-1}$. For wind speeds ranging between 3 and 8 m s$^{-1}$, the impact of  Paris emissions can be detected in the peri-urban stations when they are downwind of the city, while the rural

stations often seem disconnected from the city emission plume. As a conclusion, our study highlights a high sensitivity of the stations to wind speed and direction, to their distance from the city, but also to the ABLH cycle depending on their elevation. We learn some lessons regarding the design of an urban $CO_2$ network : 1/  careful attention should be paid to properly setting background sites that will be representative of the different wind sectors; 2/ the downwind stations should as much as possible be positioned symmetrically  in relation to the city centre, at the peri-urban/rural border; 3/ the stations should be

installed at ventilated sites (away from strong local sources) and the air inlet set-up above the building or biospheric canopy layer, whichever is the greatest; and 4/ high resolution wind information should be available with the $CO_2$ measurements.

**Keywords**: Carbon dioxide, $CO_2$ urban plume, anthropogenic emissions, variability, boundary layer height, wind, turbulence, fossil fuel, biospheric fluxes.

# 1 Introduction

Urbanized and industrialized areas are estimated to produce more than 70% of the global $CO_2$ emissions based on the consumption of fossil fuels (IEA, 2008, Seto and Dhakal, 2014). Furthermore, due to increased urbanization especially in emerging countries, urban $CO_2$ emissions are projected to grow rapidly in the next decades (e.g. Wolf et al., 2011). Understanding the contribution of cities to climate change will help stakeholders to become active at the city level in taking proper decisions regarding $CO_2$ emissions reduction (United Nations, 2011a). Especially, megacities are places where human




activities release large quantities of $CO_2$ in the atmosphere and they require scientific and political interest (Rosenzweig et al., 2010; Duren and Miller, 2012).

Based on the 2010 population criteria, the Paris conurbation is with 10.5 million inhabitants the 21[tst] megacity in the world and the 2[nd] in Europe after Moscow (United Nations, 2011b). Paris is centered in the region Île-de-France (IdF) that contains 18% of the French population (INSEE, 2012) while covering only 2% of the territory. The emission inventory reported by AIRPARIF (Association de surveillance de la qualité de l'air en IDF: http://www.airparif.asso.fr) estimates that IdF emitted a total of 41.9 Mt of $CO_2$ in 2010, i.e. 12% of French anthropogenic $CO_2$ emissions (source: CITEPA, 2012, www.statistiques.dvpt-durable.gouv.fr). However, there is no independent assessment of the regional $CO_2$ emission estimates given by the AIRPARIF inventory, which is based on the combination of benchmark emission factors and activity data. The associated uncertainties are estimated to be 20% of the total $CO_2$ emitted by month, but they are also sector dependent and can reach several tens of percent for some sectors, as also discussed in Rayner et al. (2010).

In the last years, there has been a growing international interest in quantifying urban $CO_2$ fluxes from atmospheric top-down approaches (e.g. Duren and Miller, 2012; Mc Kain et al., 2012). Large projects emerged in Indianapolis (Influx: http://influx.psu.edu), Boston (http://www.bu.edu/today/2013/the-climate-crisis-measuring-boston-carbon-metabolism/), Los Angeles (Megacities: http://megacities.jpl.nasa.gov/portal/) and in our case Paris ($CO_2$-Megaparis: http://co2-megaparis.lsce.ipsl.fr). These projects rely on the development of urban atmospheric in-situ $CO_2$ monitoring networks that should ideally include, all along the dominant wind paths: 1/ regional stations upwind of the city to characterize the regional background $CO_2$ dry air mole fraction (i.e. without having the impact of the regional emissions); and 2/ regional stations in the city and downwind of it (that will integrate both the background signal and the peri-urban/urban ones). In the following, the term dry air mole fraction is simplified by concentration and is expressed in the part per million (ppm) unit.

Several studies highlighted the fact that the $CO_2$ concentration measured in and around cities are directly sensitive to factors that control the $CO_2$ fluxes: proximity to urban centers and industrial sources, ground and air traffic, vegetation distribution, and rates of primary productivity (e.g. Wentz et al., 2002; Apadula et al., 2003; Nasrallah et al., 2003; Gratani and Varone, 2005; Strong et al., 2011). Furthermore, advection and vertical mixing strongly influences the urban $CO_2$ signal (e.g. Idso et al., 2002; Moriwaki et al., 2006). At low wind speeds, urban $CO_2$ emissions that accumulate over the city were observed to generate a $CO_2$ urban dome of several tens of ppm at night and several ppm in the afternoon compared to surrounding rural areas, reaching for example 100 ppm in Phoenix, Arizona just before pre-dawn (Idso et al., 1998, 2001). At higher wind speeds, the strength of the $CO_2$ urban dome decreases through ventilation processes to take the shape of a plume, and is considered in some former studies for other cities to reach an asymptotic value (e.g. Rice et al., 2011) which was sometimes considered representative of the regional background $CO_2$ concentration (Garcia et al., 2012; Massen and Beck, 2011).

In the Paris region, no continuous atmospheric $CO_2$ observation network was developed before the present study, albeit a couple of intensive campaigns: 1/ Widory and Javoy (2003) performed $CO_2$ measurements very close to the ground level (mostly under the influence of car exhausts) that we think is not representative of the urban scale; and 2/ in winter 2010, Lopez et al (2013) showed an increase of several ppm in the atmospheric $CO_2$ concentration in Paris (30 m above ground





level, AGL) in comparison with the $CO_2$ levels measured in the Gif-sur-Yvette station (GIF, 12 m AGL), located in a remote peri-urban area ~20 km SW of Paris. Furthermore, the Mace Head station (MHD - west coast of Ireland) is generally used as the reference site for European $CO_2$ background measurement (Bousquet et al. 1996), as it has been the case in the Heidelberg (Germany) study of Vogel et al. (2010) or in the Paris study of Lopez et al. (2013). The relevance of this remote

coastal site as a regional background site, especially for studying the regional impact of the Paris megacity on atmospheric $CO_2$ remains to be assessed at the diurnal to the seasonal scales as no regional in-situ network measurements were available to tackle this question yet.

In the framework of the $CO_2$-Megaparis project, we deployed a network of in-situ $CO_2$ stations along the path of the dominant winds and developed high-resolution top-down modeling frameworks dedicated to study the Paris $CO_2$ emissions

(Lac et al., 2013; Bréon et al., 2015). Our observation network consisted of three new continuous sites installed in and around the Paris megacity, among which one on top of the Eiffel tower (317 m AGL). These three stations (named MON, GON and EIF) were deployed in summer 2010 within the AIRPARIF infrastructure. They ran for several months of the $CO_2$-Megaparis project lifetime and delivered almost one year of $CO_2$ concentration datasets in the Paris megacity area. Additional datasets were provided by two long-term stations operated by LSCE named TRN (Schmidt et al., 2014) and GIF

(Lopez et al., 2012) that are part of the national monitoring network SO-RAMCES (now called ICOS-Fance: https://icos-atc.lsce.ipsl.fr/).

This work aims at understanding the diurnal, synoptic and seasonal variability of the atmospheric $CO_2$ concentration observed at each of the five stations of the Paris megacity network from the analysis of the first ~1-year long time series (8 August 2010 - 13 July 2011). We also compare the regional $CO_2$ concentration datasets to those at MHD ones in order to

assess how relevant this remote site is in defining the $CO_2$ background level in the Paris region. Section 2 introduces the observation network and reports the data treatment and the quality of the $CO_2$ time series. We also present the meteorological fields used over the period of study as well as observations of the atmospheric boundary layer (ABL) height in the Paris center that cover a large part of the period of study (8 August 2010 – 31 March 2011). In section 3, we present air mass back trajectories and the different wind sectors covered to assess the variability of the time series over the year of

study (section 3.1). We then analyze the diurnal variations of the $CO_2$ concentration at the 5 sites that we compare to the MHD record (section 3.2). A specific focus is carried out on the case of the Eiffel tower station. We also estimate the weekday versus weekend variability (section 3.3). We then analyze the seasonal variations of the $CO_2$ concentration at each site (section 3.4). Finally, we study the role of wind speed and direction on the $CO_2$ signal collected at the five regional network stations (section 3.5) and we assess the impact of local, regional and remote fluxes on the observed $CO_2$

concentrations. We conclude on the representativeness of each site for assessing the Paris $CO_2$ emission plume and on the lessons for regional urban network design that we learned from this study.

## 2 Experimentals





### 2.1 The measurement network

### 2.1.1 Geography of IdF and $CO_2$ emissions from the Paris region and Western Europe

Paris is located in the region of IdF in a relatively flat area and benefits from a temperate climate, with frequent rain events in all seasons and changing weather conditions. IdF covers 12011 km$^2$ i.e. only 2.2% of the national territory. In 2010, land

usage was 47% by agriculture, 31% by forests and natural areas and 22% by urbanized areas (http://www.insee.fr/fr/themes/tableau.asp?reg_id=20&ref_id=tertc01201), the last sector increasing in recent decades (United Nations, 2011b). In 2010, anthropogenic $CO_2$ emissions of IdF came from the residential and commercial buildings (43%), road traffic (29%), industry and energy production (14%), agriculture (5%), wastes (4%), aircrafts (0-915 m ASL) and airport infrastructures (4%) and worksites (1%) (AIRPARIF, 2010). The CDG airport (relatively close to GON, see

below) represents about 78% of the aircrafts and airports $CO_2$ emissions in IDF, with ~60% issued from airplane traffic on the tarmac and in flight (below 915 m ASL) (ADP, 2013; AIRPARIF, 2013). The Orly airport (16 km east of GIF) emits ~27% of the CDG airport $CO_2$ emissions (AIRPARIF, 2013). Le Bourget airport (close to GON, see below) $CO_2$ emissions are much smaller (~1.6% of the CDG one, AIRPARIF, 2013).

As shown on Fig. 1, there is a large spatial variability of $CO_2$ emissions in IdF which is mainly driven by the population

density and the location of highways. Each year, average emissions in the center of Paris are estimated to be ~70 000 t$CO_2$ km$^{-2}$ compared to ~5000 t$CO_2$ km$^{-2}$ at the surburban borders. Emissions have a temporal variability on diurnal, synoptic and seasonal scales, mainly because $CO_2$ emitted by heating varies with temperature and season, and $CO_2$ emitted by traffic changes with the time of the day, day of the week and vacation periods. Figure 2 shows the distribution of fossil fuel and cement $CO_2$ emissions in Western Europe extracted from the EDGAR v4.0 emission inventory

(http://edgar.jrc.ec.europa.eu/, 2009), highlighting large anthropogenic emissions spots in the Paris megacity, but also in the Benelux area, the Ruhr valley and the London megacity that may enrich the synoptic air masses with high $CO_2$ concentrations before they reach the Paris region.

### 2.1.2 Sampling sites

The location of the observation sites are represented on Fig. 1 and Fig. 2. Table 1 gives their exact geographical coordinates.

The Eiffel tower station (EIF) was installed on the highest floor accessible to tourists, in a closed room of 1.5 m$^2$ under the stairs providing access to the Tower communication antennas. To prevent contamination by the visitors' respiration, the air inlet was elevated to about 15 m above the last floor accessible to tourists, at the antenna level (317m AGL), where it was protected from uplifted air by several intermediate metallic floors. The instrument was set-up into a Faraday cage to avoid interferences with strong electromagnetic radiations from the antennas. The location of the Eiffel tower is not fully central

within Paris, and the 0-180° (N, E and S) wind sector of the station is exposed to a larger urbanized area than the 180-270° sector (S to W). In the 0-180° wind sector, the urbanized area covers a radius of about 20 km and includes two large emitting point sources that are the waste burning facility of Ivry (in the SE direction of the Eiffel Tower) and the heating facility of



Saint-Ouen (in the North). In the 180-270° wind sector, the urbanized area extends barely within a 10 km radius before entering into broad-leaved trees forests covering ~2300 ha. The 270°-360° wind sector is also mostly urbanized over a radius of about 15 km, although it comprises the woods of Boulogne (about 840 ha) which are located only 2 km NW of the Eiffel tower.

The Gonesse station (GON) was set-up about 20 km north east of the Eiffel tower at the local fire station in a residential area comprising a combination of streets and lawn gardens with a few trees around. The analyzer was hosted in a shelter equipped with a mast of ~4 m standing below the canopy level (~15m AGL). However the distance from the mast to the closest trees was at least 20m and the station was well exposed to wind from all directions. GON is located on a small hill relative to the centre of Paris and in the southerly direction, the station benefits from an open view of the Paris megacity. About 3 to 4 km

to the southeast and east of the station is a highway which carries high traffic during rush hours, as early as 5 am local time. The highway connects the centre of Paris and CDG airport, which is located about 7 km northeast of GON. The station is also close to the Bourget Airport located about 2.5 km to the south.

The Montgé-en-Goële station (MON) was installed in the small village of the same name with approximately 700 inhabitants located on the middle of the slope of a small hill (~20m high). The analyzer was installed on the top of the 3-floor city hall

building (~9 m AGL). The air inlet was set-up on an arm pointing about 1.5m outside of the window towards the south direction (200°) opened on fields. The north sector was covered by a few houses settled at the limit of a broad-leaved trees wood. The city hall is located on the southern side of the main road of the village which follows approximatively the northwest- southeast axis. Most of its close surroundings are agricultural fields and small villages connected by secondary roads. Montgé-en-Goële is located approximately 10 km east of CDG airport. It was considered as a NE rural site for the

Paris megacity.

The Gif-sur-Yvette station (GIF), previously described in Lopez et al (2012, 2013), has been running continuously since 2001 at LSCE (Laboratoire des Sciences du Climat et de l'Environnement). The air inlet is set up on the roof of a building at 7 m AGL. The site is located ~20 km south-west of the centre of Paris on the Plateau de Saclay and surrounded mainly in the 0°-90° sector by agricultural fields and by a few villages. Approximately 1 km away in this direction, a national road passes

on a north-south axis with high traffic levels in the morning and in the evening during rush hours. In the 270-360° sector, the atomic and environmental research agency (CEA of Saclay) has approximately 7000 employees, and is further surrounded by agricultural fields. In the last wind sector (90°-270°), a band of forest of about 1 km depth extends along the west to east axis down to the bed of the Yvette river. GIF is located roughly at the same distance from the Eiffel tower as GON. However, the environment is more rural in GIF than in GON so that we can label GON as a residential peri-urban site and

GIF as a remote peri-urban site - although it is not as rural as the site at MON. About 16 km East of GIF is located the Orly airport.

The Traînou station (TRN), previously described by Schmidt et al (2014) has been running continuously since 2007. It is located about 120 km south of the center of Paris in the region "Centre", within the Orleans forest (50000 ha). A 200 m transmitter mast was equipped with four sampling levels: 5 m, 50 m, 100 m and 180 m AGL. TRN is located ~13 km



northeast of the city of Orléans which has about 120 000 inhabitants. There are a few villages around the station, including Traînou village with 3195 inhabitants in 2012 (http://www.insee.fr/fr/themes/comparateur.asp?codgeo=com-45327). The station area is covered by agriculture fields and by a mixed forest composed of deciduous and evergreen trees. In this study, we use the datasets sampled from the 50m and 180m levels. TRN is considered as a rural site for the Paris megacity although quite already remote from it.

Mace Head station (MHD) has already described by Biraud et al (2000) and Messager et al (2008). Atmospheric $CO_2$ has been continuously measured there since 1992. This station, located on the west coast of Ireland, is an important site for atmospheric research in the northern hemisphere, as its remote location facilitates the investigation of trace constituent changes in marine and continental air masses. Most often, the station receives maritime air masses, although sometimes it is in the footprint of continental air masses coming from Europe, or more locally from Ireland and the UK (see Messager et al., 2008 for further details).

The Qualair station (QUA) is located in the Paris city center on the campus of Université Pierre et Marie Curie in Jussieu on the last floor of a building (25 m AGL), about 4 km east of the Eiffel tower along the Seine river. It is briefly described in Dieudonné et al (2012). This station is ideally located to monitor the height of the urban atmospheric boundary layer (ABL) above the Paris megacity.

## 2.2 $CO_2$ measurements

### 2.2.1 Measurement system and calibration procedure

The $CO_2$ datasets of the $CO_2$-Megaparis stations (MON, GON and EIF) were collected from 8 August 2010 to 13 July 2011 using CRDS (Cavity Ring Down Spectroscopy) analyzers (Picarro, model G1302) at 0.5 Hz. These three stations were identically setup: atmospheric air was pumped through short inlet lines made of Synflex® (4.3 mm inner diameter) with a flowrate of 0.15 L.min$^{-1}$. The cell temperature of the analyzers was controlled at 45° C and the cell pressure at 140 Torr. At EIF, the analyzer was specifically designed to undergo higher temperatures inherent to the metallic structure of the tower and the cell temperature set point was set higher (60° C). Air was not dried before analysis at the 3 stations and we applied on our datasets the automatic $CO_2$ water correction implemented on the CRDS instruments (Rella, 2010).

The GON and MON stations were equipped with four high pressure aluminum cylinders containing gas mixtures of $CO_2$ in synthetic air (matrix of $N_2$, $O_2$ and Ar) for instrumental calibration. Before on-site deployment, the $CO_2$ concentration of the cylinders was assigned at LSCE on the WMO-X2007 scale by a gas chromatograph (GC) described in Lopez et al (2012). It spanned a range from 370 ppm to 500 ppm. At each site, three of the tanks were used for the instrument calibration and measured every 2 weeks. The calibration sequence consisted of four cycles (6 h total). One cycle measured the tank one after the others for 30 minutes each. The fourth tank called "target" was run for 30 minutes every 12 hours. The target was used to monitor the instrumental drift and to assess the dataset accuracy and repeatability. At EIF, for safety reasons it was not possible to leave any gas tanks on the site so the target tank was measured every two weeks and the calibration gases every 3



months only (two calibration cycles of 20 minutes for each gas, for a total sequence of 2 hours). The instrumentation and the calibration procedure of the two SO-RAMCES stations (GIF and TRN) have already been described in Lopez et al. (2012, 2013) and Schmidt et al (2014).

### 2.2.2 Data processing and quality control

The CRDS $CO_2$ data were calibrated by applying the linear fit equation of the $CO_2$ concentration of the calibration tanks measured by the CRDS analyzer vs the GC $CO_2$ concentration of these tanks. Only the last calibration cycle over the 4 ones at MON and GON (over the 2 ones at EIF) was retained. For all of the calibration and target gas cylinders, the CRDS $CO_2$ concentration was calculated as the average of the last 5 minutes of each gas. The accuracy of the datasets was calculated as the mean difference between the $CO_2$ concentration given by the CRDS analyzer and by the GC for the target gas. The long-

term repeatability of each dataset was calculated as the standard deviation of the mean concentration of the target gas given by the CRDS analyzer over the year of observations.

Table 2 summarizes the accuracy ($\leq 0.13$ ppm) and repeatability ($\leq 0.38$ ppm) calculated from the 5 minute averaged data for MON, GON and EIF. As expected, the dataset of EIF shows larger deviations compared to GON and MON due to less frequent calibration and target gas measurements and a shorter calibration procedure.

The data of GON, EIF and MON were automatically filtered against cavity pressure (P) and cavity temperature (T) departure to the set points ($P_0$ and $T_0$) according to the ICOS procedure, keeping only points for which $|P-P_0|<0.1$ Torr and $|T-T_0|<0.004°$ C for MON and EIF (0.006° C for EIF). Furthermore, dead volumes in the set-up lead to instabilities in the response of the analyzer until 2 minutes after switching from one gas line to the other. These 2 minute periods were systematically removed from the datasets. In total, more than 92% of the raw data were validated after these filtering steps.

The datasets of MON, GON and EIF were manually inspected to remove spikes of $CO_2$ from very local influences (e.g. fire training at the GON station, breathing of a maintenance operator on the sampling inlet...). In total less than 1% of the total datasets were removed, leading to a final amount of 91% of the data validated after the (P, T) filtering and the quality control step.

The GIF, TRN and MHD data processing and quality check were assessed in former studies by Schmidt et al (2014) and

Messager et al (2008): the repeatability of the 1 h average $CO_2$ concentration of the target gas is 0.05 ppm at GIF, 0.06 ppm at TRN and 0.05 ppm at MHD. The instrumentation of these 3 sites is directly linked to the WMO-X2007 scale.

At each station, some instrumental failures occurred during the $CO_2$-Megaparis period of study. The amount of available data points in the final datasets which are all provided on hourly averages is reported in Table 4 for each month and for each site, and is in most cases above 80%.

In the following study, we will use $CO_2$ hourly means for all of the stations. Apart from a few exceptions that will be mentioned, time is always given in hours UTC. Local time in Paris is UTC+2 from April to October and UTC+1 from November to March.





## 2.3 Atmospheric boundary layer height measurements

ABL heights over Paris were determined using the 532 nm elastic lidar of the QUALAIR station (http://qualair.aero.jussieu.fr/) from 8 August 2010 to 31 March 2011. A description of the instrumental setup and data processing can be found in Dieudonné (2012, 2013). The ABL height (ABLH) can be retrieved from elastic lidar

measurements because the lidar signal is proportional to the backscattering coefficient of aerosols. In fair weather, this leads to a sharp signal decrease between the polluted boundary layer (where aerosols emitted from the surface are trapped) and the clean free troposphere. The altitude where the signal first derivative reaches its absolute minimum corresponds to the center of the entrainment zone (Menut et al. 1999). The depth of the layer where the signal first derivative is lower than 80 % of its absolute minimum is used to estimate the base of the entrainment zone, which corresponds here to the lowest ABL height

(LBLH) estimate. More complex situations can occur, when elevated layers of aerosols are present in the free troposphere. In that case the absolute minimum of the signal gradient can be located elsewhere than at the ABL top. To resolve such situations, threshold conditions are applied to discriminate significant minima of the signal gradient (Dieudonné, 2012) and results are manually inspected to check for temporal continuity (as the altitude of a layer cannot vary much from one lidar profile to the next). When the ABL is capped by a cloud, the very strong light scattering by water droplets creates a sharp

increase of the lidar signal at the ABL top. In such cloudy weather, the cloud base height is the best estimation for the ABLH. The LBLH is calculated as in fair weather.

The ABL height database was constructed by applying this detection method on hourly average lidar data, leading to hourly average ABL depth values. The data were acquired during daytime and weekdays, since an operator had to be on site to shut down the system in case of rain. The dataset covers 70% of the year of study.

## 2.4 Meteorological fields

Urbanized areas are characterized by specific meteorological patterns (e.g. Masson et al., 2000). For example, the urban heat island effect was observed to generate a gradient of temperature of a few degrees and a gradient in the ABLH of several percent between Paris city center and its rural surroundings (Pal et al, 2012 ; Lac et al, 2013). As far as possible, it is thus appropriate to use local meteorological fields for each of the regional atmospheric $CO_2$ stations. Since our sites were not

equipped with their own meteorological sensors, the Meso-NH model was run over the full period of study at a time step of 60 s and a spatial resolution of 2 km to generate wind speed and direction over a domain including Île-de-France (Lac et al., 2013). This modeling framework includes the land and-surface–atmosphere interaction model SURFEX with an urban scheme (Town Energy Balance (TEB); Masson, 2000) and a vegetation scheme (Interactions between Soil, Biosphere, and Atmosphere (ISBA-A-gs); Calvet et al., 1998; Noilhan and Planton, 1989). It was already validated against observations for

one week of March 2011 in Lac et al (2013), where it is described in detail. The meteorological fields were extracted for the present study from the model with an output frequency of 1 h at the sampling height of each station. About 1.5 km north of GIF at the CEA of Saclay (SAC), a mast equipped with meteorological sensors provided wind fields data at 10m AGL from



August 2010 to April 2011. In that period, the SAC and the GIF Meso-NH meteorological datasets match each other on average within 0.8 m s$^{-1}$ for wind speed, and 3.7° for wind direction, giving additional confidence in the average behavior of the model, at least in such peri-urban areas.

For wind fields at MHD, we use a local meteorological hourly observation dataset provided by Met Eireann.

Fig. 3 shows the wind roses at GIF for each season, given that the synoptic features are broadly similar to all of the regional stations. Two dominant wind regimes were observed according to the general meteorological features of the region: the southwest regime dominates mostly in summer, autumn and winter, and a northern regime (northeast and northwest sectors) mostly in spring and winter. Wind speed varied from ~0 m s$^{-1}$ on 18 September 2010 to a maximum of 11.1 m s$^{-1}$ on 13 November 2010, the mean wind speed being 3.4 m s$^{-1}$. The first (25%) and third (75%) quartiles were 2.2 m s$^{-1}$ and 4.4 m s$^{-1}$,

respectively. The main variations of wind speed occurred during changes of synoptic conditions. In MHD, winds blew mostly from the Atlantic Ocean in all seasons, including both the southwest and the southeast sectors. MHD also sometimes received continental air masses mostly in winter, spring and autumn. At this station, wind speeds ranged from 0.1 to 25.3 m s$^{-1}$ with a mean at 7 m s$^{-1}$ and the first and third quartiles standing at 4.1 and 9.5 m s$^{-1}$, respectively.

Regarding temperature, field observations were available over the full period of study at 100 m AGL at SAC (but not closer

to the surface). Since we are here mostly interested in relative variations of the temperature at the seasonal scale, we use this dataset as a proxy of the air temperature for all stations located in IdF (although we know that the urban heat island can generate differences of a few degrees between the city and its surroundings, as shown in Pal et al., 2012). The hourly temperature dataset collected at SAC 100 m AGL over the whole period of study is shown on Fig. 4. Temperature ranges from a minimum monthly mean of 0° C in December to a maximum monthly mean of 18.8° C in August.

**3 Results**

**3.1 CO$_2$ concentration time series and air mass trajectories**

In order to understand where the air masses were originating from, we ran back trajectories over the full period of study using the HYSPLIT (Hybrid Single Particle Lagrangian Integrated Trajectory: http://www.arl.noaa.gov/HYSPLIT_info.php) model at a 2.5° x 2.5° and 6 h resolution (see supplementary material S1). In all cases, the monthly clusters illustrate the high

variability of the origin of the air masses, which can pass over high CO$_2$ emissions areas such as the megacity of London, the Benelux or the Ruhr regions before reaching IdF. The air masses can also be advected from clean areas such as the Atlantic Ocean, or from biospheric regions such as in the middle of France. This high atmospheric transport variability implies that the Paris regional CO$_2$ background signal may be highly variable depending on the synoptic conditions and that wind direction and speed are key parameters to take into account in order to understand the CO$_2$ concentrations recorded at the

different sites.

The hourly time series of CO$_2$ used in this work are shown in the supplementary material S2 and are colored according to six wind classes. The *local class* is defined for wind speed lower than 3 m s$^{-1}$ and the *remote class* for wind speed higher 9 m s$^{-1}$.



For wind speeds comprised between 3 and 9 m s$^{-1}$, we defined four remaining classes according to the wind direction: *northeast* (NE), *northwest* (NW), *southeast* (SE) and *southwest* (SW). As an example, in GIF the partition of the air masses between the different wind sectors over the full period of study is the following: 16% from the NE, 15% from the NW, 24% from the SW, 7.5% from the SE, 36% from the *local class* and 1.5% from the *remote class*. These classes will be used to better assess the general features of the $CO_2$ seasonal cycles, although a much finer wind analysis will be conducted in section 3.5.2.

## 3.2 $CO_2$ diurnal cycles

### 3.2.1 Mean $CO_2$ diurnal cycles

Diurnal cycles of atmospheric $CO_2$ are affected by local sources and sinks, regional transport and ABL dynamics (Fang et al., 2014; Garcia et al., 2012; Rice et al., 2011; Artuso et al., 2009; Gerbig et al., 2006). The mean $CO_2$ diurnal cycles and associated 1-σ standard deviation are shown in Fig. 5 for the different stations. Noticeable differences can be observed between the sites.

The diurnal amplitude of the $CO_2$ concentration from the lowest to the highest is 2.6 ppm (MHD), 6.5 ppm (TRN180), 11.2 ppm (EIF), 14.9 ppm (MON), 15.5 ppm (TRN50), 18.2 ppm (GIF) and 30.6 ppm (GON). While the $CO_2$ diurnal pattern at TRN can mostly be explained by the biosphere activity and vertical dilution in the ABL, the peri-urban and urban stations are also expected to be strongly influenced by the diurnal cycle of the Parisian anthropogenic sources. For all sites except EIF, the maximum concentration occurs in the late night/early morning (4-5 h for TRN50, MON, GIF and GON; 7-8 h for TRN180) when the ABL is the most shallow, vegetation respires and traffic gets dense. The minimum of the cycle occurs in the afternoon (14 h to 17 h) when the ABL is the deepest and well mixed and at seasons when the vegetation photosynthesis is active. The case of EIF is specific due to its elevation and a strong interaction of urban $CO_2$ emissions with the ABL cycle (see section 3.2.3). As a consequence, the maximum of the $CO_2$ concentration at EIF is in the mid-morning (10h) and its minimum is at night (0h).

Comparing the 50 and 180 m levels at TRN, we observe that a vertical gradient of the $CO_2$ concentration exists, along with a phase shift of the diurnal cycle: the maximum concentration is observed at 5 h UTC at TRN50 against 7 h UTC at TRN180, due to the coupling of the $CO_2$ fluxes with the ABL cycle. Indeed, $CO_2$ emitted during the night and early morning by anthropogenic sources and by the biosphere's respiration accumulates near the ground into the shallow nocturnal boundary layer (Schmidt et al., 2014) until the ABL develops in the morning, uplifting $CO_2$ (from 5 h to 7 h UTC) to the 180m level. In the afternoon, when the ABL is well-mixed and deeper than 180m, the mean difference between the concentration at the 50 and 180 m levels is very low (0.3 ppm). Furthermore, as noticed in Schmidt et al (2014), the amplitude of the diurnal cycle decreases with increased sampling height as elevated sampling levels are decoupled from the $CO_2$ sources during the night. As noted in Fang et al (2014), this covariance between the biospheric $CO_2$ activities and the ABL dynamics can make




it difficult for inversion models to properly reproduce the $CO_2$ vertical gradient and thus, use nighttime data for inversions. During daytime, the ABL is well mixed and the vertical bias would be very tiny.

There is a significant positive gradient in the $CO_2$ concentration observed between the regional stations, and also with MHD. The amplitude of the gradient between each site depends on the time of the day and its variation is mainly driven by the $CO_2$ diurnal cycle at the continental sites. Treating EIF apart, the more the station is surrounded by urbanization, the higher is the gradient with MHD, as the average levels of the $CO_2$ concentration recorded at a station increases with a higher proximity to anthropogenic emissions from Paris. The left panels (a-g) on Fig. 6 show that the hourly 1-$\sigma$ variability of the mean diurnal cycle remains quite constant over the day at TRN50, TRN180 and MHD. It is a bit more variable for the rural and remote peri-urban stations that are located within IdF (MON and GIF). The variability changes significantly with the time of the day at EIF and even more at GON. We can conclude that: 1/ the more the station is within the urbanized part of the city, the more variable is the collected $CO_2$ signal, which reflects the spatial and temporal variability of anthropogenic emissions coupled to atmospheric transport fluctuations; and 2/ the MHD signal is several ppm below the continental signals and does not properly reproduce the diurnal variability of the regional stations: it is clearly not a relevant background site for continental Europe urban studies at the diurnal scale.

The right panels (a'-g') of show the mean diurnal cycle at each site by season. The influence of anthropogenic activities on the observed $CO_2$ is expected to be the highest in wintertime when emissions from heating superimpose to traffic and other sources, photosynthesis is minimal and the diurnal ABL is thinner. Although they vary with the time of the day, on average $CO_2$ emissions from traffic are quite constant all over the year (according to AIRPARIF 2010 inventory: about 7 kt.hr$^{-1}$, a bit less in summer with about 5.5 kt.hr$^{-1}$). On the contrary, emissions from gas combustion (from the residential, the public and the commercial infrastructures that include mostly heating, production of hot water, air conditioning and cooking) show a seasonal cycle (mostly from heating), releasing about 12 kt.hr$^{-1}$ of $CO_2$ in the atmosphere in winter against approximately 5 kt.hr$^{-1}$ in summer (AIRPARIF, 2010). The influence of the biospheric fluxes is expected to be the highest in spring followed by summer and autumn (see Fig.4 in Bréon et al., 2015) with little influence in winter. In the Supplementary material S3 for each site we give the annual and seasonal averages of the daily minimum and of the daily maximum of the hourly concentration, along with the annual and seasonal averages of the diurnal cycle amplitude (max-min concentration difference). The lines entitled "variation" give the mean of the hourly 1-$\sigma$ standard deviation of the min and of the max of each diurnal cycle.

It is noticeable that the mean winter concentration is about 6 ppm higher at MON than in TRN50. Both stations are in rural environment, but MON is closer to Paris than TRN. As the signals are quite similar in summer, this difference can not likely be explained by biospheric activity, and is more probably due to a higher anthropogenic influence in MON. This influence could be due to local sources and/or to $CO_2$ emission plume of the Paris megacity, a point that will be further inferred from the wind analysis in section 3.5.

The influence of the urban emissions in GIF, MON and GON results in a higher mean diurnal concentration of atmospheric $CO_2$ at these sites compared to the others for all seasons (and mainly in winter) and of its variability. The impact of traffic





emissions is visible in GIF, MON and GON on the winter cycle with two $CO_2$ maxima during rush hours (morning and evening). Although traffic occurs throughout the year, these peaks are more or less masked by the biospheric activity during the other seasons. In addition, the ABL is shallower during winter leading to higher $CO_2$ concentrations. The amplitude of the morning and evening peaks is higher in GON than in GIF and MON and denotes a stronger impact of traffic emissions in

GON than in the two other stations. GON also shows the maximum inter seasonal difference between summer and winter (31.3 ppm in the afternoon) which is higher than the mean annual afternoon dispersion. Actually, the whole diurnal cycle is shifted towards higher concentrations at GON, the mean concentration being higher in GON than in GIF, TRN50, TRN180 and MHD for all seasons, with the largest differences in winter. The full variability observed at GON over the year can thus be explained partly by the seasonal variation of the biosphere activity and ABL dynamics, but also by a strong impact of the

regional anthropogenic $CO_2$ emissions variability. The impact of the Paris emissions and of more local sources around the station (highways, airports) will be further assessed in Section 3.5.

### 3.2.2 The specific case of the top of the Eiffel tower

In all seasons, the $CO_2$ diurnal cycle at EIF is out of phase with the other stations, with a maximum occurring later, in the mid-morning instead of the late night/early morning (Fig.7). EIF is significantly higher (317 m AGL) than TRN180 (180 m

AGL) so when comparing these elevated sites to ground stations, the effect of the $CO_2$ coupling with the ABL dynamics can be expected to appear stronger at EIF than at TRN180. Such coupling was already mentioned in the framework of a direct $CO_2$ transport modeling study in March 2011 (Lac et al., 2013). Furthermore, Dieudonné et al (2013) demonstrated the existence of a vertical concentration gradient between the bottom and the top of the Eiffel tower for $NO_2$, a species co-emitted with $CO_2$ during combustion processes especially by the traffic sector, and this vertical gradient was shown to be

correlated with the ABL dynamics.

We show in the supplementary material S4 the hourly means of the LBLH observed at the QUALAIR station during daytime, colored by hour, and compared with the level of the EIF station. These data are summarized in Table 3. We recall that the LBLH dataset does not cover the whole period of study, but the most interesting of it as it includes the cold months during which the LBLH and dynamics are at their lowest. The period of August to March allows us to observe a large

portion of the seasonal cycle of the LBLH which is characterized by a change in its maximum value (on average 1200 m in summer, 400m in winter) and in the phase of its development, which starts earlier in summer. We do not have the proper data to quantify precisely this starting time, however we note that the LBLH is always above the level of EIF in summer, while it stands below (at 301m on average) before 6 UTC in winter (see Table 3). We can thus infer that the EIF station could be often above the nocturnal layer at night, inside the residual layer (but not in the free troposphere).

In Fig.7, we also show the $CO_2$ diurnal cycle for each season computed using only the data that were collected at the EIF station at the same hours than the LBLH data. The $CO_2$ signal increases in the morning when the growing ABL brings to EIF the nighttime and early morning $CO_2$ emissions that got trapped into the nocturnal and/or nascent boundary layer. However ,





compared to TRN180, the effect in EIF is much stronger due to larger emissions in the city, especially from the morning traffic peak (from 6 h to 10 h local time i.e. 4-8 h UTC in summer and 5-9 h UTC in winter) [http://www.dir.ile-de-france.developpement-durable.gouv.fr/les-comptages-a174.html]. Later, the $CO_2$ signal dilutes into the growing ABL to reach a minimum in the afternoon.

5   **Winter**. As expected, the process of vertical mixing is quite slow in wintertime. The $CO_2$ concentration increases in the morning (~ + 6 ppm) with the maximum concentration encountered at 13 h UTC for a development of the LBLH of only ~157 m within a 7 hour time frame. After the morning flush of the surface emissions due to the growth of the ABL, the concentration decreases quite rapidly to reach its daily minimum at 16 h. At the end of the day, the LBLH falls and gets quite rapidly below the EIF station level, decoupling the EIF station from the surface. Although we do not have Lidar data after 18

10  h UTC to confirm it, this likely explains the relatively low level of $CO_2$ concentrations observed in the late night.

**Spring**. In spring, the $CO_2$ signal increases until 10 h to a maximum of 420 ppm while the ABL height increases by ~287 m. The shape of the $CO_2$ mean concentration and LBLH diurnal cycles suggests that the relatively high $CO_2$ concentrations encountered in the late night/early morning result from the evening high $CO_2$ emissions trapped into the previous day ABL that became at night the residual layer.

**Summer**. The $CO_2$ concentration is on average lower than in the other seasons due to local and regional photosynthesis activity, lower anthropogenic emissions levels and higher LBLH. In particular, the observed LBLH during daytime is always above the EIF station level (Fig. S4) so that one would expect $CO_2$ concentrations to peak in phase with the traffic counter records, between 6 h and 7 h. However, the $CO_2$ diurnal cycle at EIF remains out of phase with those recorded at ground level stations, though the delay with the morning peak is reduced compared to other seasons. The $CO_2$ concentration remains

quite stable between 7 h and 9 h, despite the increasing LBLH (+460m) and of the decreasing traffic counts. However, one must keep in mind that until late morning, the air dragged into the ABL by entrainment does not come from the clean free troposphere but from the polluted residual layer, explaining why high $CO_2$ concentration can maintain. After 9 h, the $CO_2$ concentration steadily decreases, though the average LBLH still increases. This drop in concentration can be explained both by an increase in the photosynthetic activity with increasing solar flux, and by vertical dilution. Indeed, though the LBLH

still rises after 10h, the entrainment zone goes on growing until the mid-afternoon (Dieudonné, 2012) blending in clean air from the free troposphere. During the late afternoon, the $CO_2$ concentration increases again as vertical mixing, decays, and as the evening traffic peak starts (around 15 h).

**Autumn**. The LBLH is close to the EIF altitude. The moderate development of the ABL during the morning does not compensate for the accumulation of the peak traffic emission in the ABL, so that the $CO_2$ concentration increases from 5 h to

10 h, leading to a $CO_2$ increase of 17.1 ppm for an LBLH increase of 470 m. At the end of the afternoon, the LBLH decreases and it gets close to the level of EIF, decoupling the station from the surface. This could explain why the late night/early morning concentrations are relatively low and the morning bump of $CO_2$ quite large. However this remains an hypothesis as we do not have enough points for a robust demonstration.



This analysis confirms that the coupling of the urban $CO_2$ emissions together with the dynamics of the ABL height is very likely a major controlling factor of the specific $CO_2$ diurnal pattern observed at EIF. We lack data at night and in the early morning to make a deeper analysis on the ABL dynamics and especially on the role of turbulence on the $CO_2$ variability. We can conclude that a vertical and fluctuating gradient of $CO_2$ likely exists above the Paris megacity, between the ground level and 317m AGL (and likely higher). This vertical gradient can be estimated by subtracting the EIF signal from the GON or the GIF one. In the early morning (4-5 h am) the GON-EIF (respectively GIF-EIF) gradient is +35 ppm (+18 ppm) in spring, +31 ppm (+17 ppm) in summer, +30 ppm (+10 ppm) in autumn, and +14 ppm (+4ppm) in winter. In the afternoon (14-16 h), the GON-EIF (respectively GIF-EIF) gradient is lower in absolute values and changes of sign: -7 ppm (-8 ppm) in spring, -4 ppm (-3 ppm) in summer, -4 ppm (-7 ppm) in autumn and -2 ppm (-5 ppm) in winter. The gradient is thus at its maximum at night and in the warm seasons, which may also reflect the influence of the biospheric respiration at the stations close to the ground level, compared to EIF.

### 3.3 Weekday versus weekend

According to the AIRPARIF inventory, the total $CO_2$ emissions of IdF are lower during weekends than during weekdays, with mean differences of the order of 30-40% during daytime and 50-60% during nighttime. We infer here the impact of such variations on the atmospheric concentrations. In Fig. 8, we show the mean diurnal cycles of the $CO_2$ concentrations at each site for each day of the week, as well as the associated standard deviation (1-σ).
In GON, the $CO_2$ concentrations are systematically lower over the weekend days, especially on Sundays (5-10% of decrease during daytime, 25-35% of decrease during nighttime). A similar pattern is observed for MON. However, the weekdays-to-weekend ratios observed for the $CO_2$ concentrations are lower than those computed from the emissions given by the inventories. This could be due to an overestimation of the difference from the inventory. Note that while the variability of the $CO_2$ means is very large in GON, it is lower during weekends than during weekdays. Surprisingly, the $CO_2$ diurnal cycle does not change so much in GIF between a working weekday and a weekend (except for a small decrease during nighttime over the weekend), nor at EIF and TRN. And, during nighttime at GIF we observed the highest concentrations from Sundays to Wednesdays, with concentrations lower by 3-5 ppm (a 20-25% decrease) from Thursdays to Saturdays. This could be due to a specific traffic pattern within the footprint of the station, but we currently do not have access to local traffic data for each day of the week to verify this hypothesis.

### 3.4 $CO_2$ seasonal cycle

We computed the seasonal cycle of $CO_2$ at each site, based on monthly means of our ~1 year datasets (Fig. 9a). The seasonal cycles of the air temperature and available LBLH data (at QUA) are also shown on the same figure.
Ignoring the specific case of EIF (section 3.2.3), throughout the year we observe that the monthly mean $CO_2$ concentration increases with the vicinity of the station to larger $CO_2$ emission sources. The maximum gradient with MHD is observed at



GON (from 6.8 ppm in July to 27.5 ppm in December). Similarly to what is observed at the diurnal scale (section 3.1), differences of several ppm are observed at the seasonal scale between the continental stations and MHD, while the differences between the rural/peri-urban/urban stations is of the same order of magnitude. Thus, MHD appears not to be relevant as a background site for studying $CO_2$ in the Paris region at the seasonal scale as well.

At each station, the monthly mean $CO_2$ concentration follows a seasonal cycle that reaches its maximum in winter and its minimum in summer. This is expected due to: 1/ the seasonal cycle of the biosphere; 2/ the variability of anthropogenic emissions, mainly from the heating sector, which are directly linked to ambient temperature (see 3.2.2); and 3/ the seasonal cycle of the ABL height (section 3.2.3), which is at the lowest in wintertime. It is difficult to estimate the biases due to missing data points in the time series (section 2.2.2), however as an indicator of robustness, the data coverage for each month

and each station (given in Table 4) is very good overall.

To assess the variability of the seasonal cycle, Fig. 9b shows the $CO_2$ monthly means at each station with error bars representing the associated 1-σ standard deviation. Note that the 1-σ dispersion is the highest at GON and the lowest at MHD. More generally, the variability increases with the level of urbanization around the station and the distance to anthropogenic $CO_2$ emission sources. Therefore, increases in the variability from one month to the next can be used to track

down the influence of more local and thus fresh sources, as a complement to the "local" wind sector (wind speed < 3 m s$^{-1}$). Some specific seasonal patterns can be observed:

**Winter**. In winter, the lower biospheric activity makes the $CO_2$ concentration more sensitive to fluctuations in anthropogenic emissions (see Bréon et al, 2015). In Paris, January is usually the coldest month (meaning the month with the highest heating emissions). However, the months of December 2010 and February 2011 were characterized by cold episodes, while January

2011 was rather mild. This resulted in higher $CO_2$ concentrations in December and February than in January for MON and GON. In GIF, EIF and TRN, the secondary maximum (Feb.) is shifted to March. Indeed, in February, southerly winds prevailed (see S1 and S2), bringing Parisian anthropogenic $CO_2$ emissions in the direction of GON and MON and depleting the southern stations while in March, winds blew mostly from the NE/SE sectors bringing higher $CO_2$ levels to GIF, TRN, EIF and also MHD. The higher $CO_2$ concentration encountered in December compared to February or March can be

explained by the ABL height being minimal in December (Fig. 9a). However, in February the GON signal remains the highest of all stations, and the concentrations observed at MON are higher than those recorded at TRN. Here we may see the impact of air masses advected from the NE with higher $CO_2$ background levels, and a sensitivity to upwind emissions at GON especially. Such influence of meteorological conditions on the seasonal cycle of continental stations was also reported in the literature (e.g. Fang et al., 2014; Zhang et al., 2008) and will be further assessed in section 3.5.2.

**Spring**. Starting from April, we observe a decrease of $CO_2$ at all stations except GON, as regional photosynthesis activity develops (Bréon et al., 2015). In April, the high variability of the GON signal and the prevailing local, SW and NW wind sectors show that the station experiences strong influence from anthropogenic emissions, local or advected, and explains why the $CO_2$ concentration remains higher than in the other stations. From April to July, we observe that the $CO_2$ concentration at TRN180 is always equal to or below MHD, showing the strong influence of regional biospheric activity on concentrations





measured at continental stations. Indeed, this effect is also observed in TRN50 and MON in May when the biosphere is very active and winds blew mostly from the SE and SW, bringing air masses from the forests of the Centre region to IdF. During other spring and summer months, concentrations at TRN50 and MON remain higher than at MHD as the dominant winds were from the NE sector, likely bringing emissions from the Ruhr/Benelux to MON and TRN and/or from Paris to TRN.

**Summer**. For all stations except GON, the annual minimum of concentration follows the minimum of anthropogenic emission and occurs in August. In GON, the contribution of the local wind sector is strong in August, as confirmed by the large 1-σ deviation, explaining why the minimum of concentration is shifted to July, another month with reduced economic activity and emissions (on top of a high level of photosynthesis). The higher concentrations in August at GON are also associated with slow winds blowing from the northwest direction, indicating an impact of relatively local emissions, though
no noticeable large $CO_2$ source is located in the vicinity of the station in this direction (sec. 2.1.2).

**Autumn**. September is characterized by an increase of the monthly mean $CO_2$ concentrations at all stations, although the increase is higher in GON (+9 ppm) than elsewhere (+3 to +5 ppm). As there were several local and NW events during that month, we infer that this larger increase is due to urban emissions in the vicinity of GON (eg. from CDG airport) or a bit further to the NW side of GON.

The sensitivity of the stations to wind speed and direction will be analyzed in more detail in the next section, and especially the question of higher background $CO_2$ levels advected from the NE sector.

## 3.5 Wind study: from local to regional signals
### 3.5.1 Wind speed effect

Wind speed is a key factor in modulating the dispersion of $CO_2$ emissions. Figure 10 shows the mean hourly $CO_2$
concentrations and the associated standard deviations recorded at GON over the year of study for local afternoon hours only (11-15 h UTC) as a function of the wind speed and colored by wind direction. The $CO_2$ concentrations have been seasonally adjusted to avoid biases due to seasonal variability (section 3.4). The left panel of Fig. 10 shows that the amplitude of the $CO_2$ concentration range and especially the maximum values decrease exponentially with the wind speed because of the ventilation and dilution effects. Such behavior is observed at all the regional stations, although the wind speed maximum is
higher at TRN (~11 m s$^{-1}$) and even higher at EIF (~20 m s$^{-1}$) due to the elevation of these stations. The 1-σ dispersion from the hourly means (called variability on the right panel of Fig. 10) shows a similar dependency on wind speed. At low wind speed, the relatively high level of variability can be associated to the impact of fresh and regional anthropogenic $CO_2$ emissions. For high wind speeds, the hourly averaged $CO_2$ concentration converges towards a mean value and the 1-σ variability drops below 1 ppm. Such behavior was previously reported at former $CO_2$ urban stations for other cities (e.g.
Garcia et al., 2012; Rice et al., 2011; Massen and Beck, 2011). However, and contrary to those studies, we do not think that this mean value can be considered as an asymptote, as it originates only from a few sparse events (spread over 7 days of the period of study), nor that it can be considered as a background $CO_2$ concentration for the stations.


Indeed, Fig. 11 shows this $CO_2$ mean value at the different stations: a $CO_2$ horizontal gradient appears, with the maximum of difference (6.6 ppm) observed between GON and MHD. The high wind speed events that occurred during the period of study correspond only to winds blowing from the southwest sector, mostly from the 200-220° sector. GON was thus immediately downwind of Paris emissions, most likely the reason why it exhibits the highest mean constant value. A

gradient is also observed between TRN and MHD and between GIF and MHD. As both TRN and GIF are located upwind of Paris, we see once again here that MHD does not provide an adequate $CO_2$ concentration background level for Paris and other continental Western European cities. The peri-urban upwind station of GIF has quite a similar mean constant value as the rural downwind station of MON. Indeed, MON station was not in the path of Paris $CO_2$ urban plume in this 20° wind sector. The EIF value is also lower than at GIF and GON, supporting the fact that for such high winds, the top of the Eiffel

tower was not very sensitive to surface emissions, most likely because between 0 and 300m agl, ventilation of emissions was stronger than their vertical mixing.

### 3.5.2 Fine wind sector analysis

In order to distinguish the relative contributions of the local, the remote and finally the Paris megacity regional $CO_2$ fluxes on the $CO_2$ concentration observed at the 5 stations of the Paris network, we analyzed the dependence of the observed $CO_2$

concentration and its variability on the horizontal wind speed and direction. Considering the diurnal variability of vertical transport dynamics (section 3.2), we separately analyzed afternoon (11 h to 15 h UTC) and nighttime (22 h to 2 h UTC) data. For the TRN station, we consider that the TRN50 level is sufficient for this analysis.

Inner Paris extends within a 10 km diameter, while the Paris conurbation extends to a diameter of 30 to 50 km. The distance of the peri-urban stations GON and GIF to the Paris inner city is about 10 km and 15 km, respectively. The distance of the

rural stations MON and TRN to inner Paris is about 30 and 100 km, respectively. Taking into account these distances, we set the hypothesis that we can assess the influence of local emissions using hourly means observed in low wind speed conditions (less than 3 m s⁻¹) while the influence of remote emissions can be analyzed using data recorded in relatively high wind speed conditions (more than 8 m s⁻¹). In the middle range (3-8 m s⁻¹), we expect most of the $CO_2$ variability to be driven by the influence of the regional emissions out coming of the Paris megacity area.

For all of the regional stations, Fig. 12 shows the pollution roses of the mean afternoon $CO_2$ concentration binned by wind speed (ws) and wind direction (wd) with a resolution of 1 m s⁻¹ for ws and 10° for wd. Here as well, the $CO_2$ hourly concentration has first been seasonally adjusted. In order to assess the representativeness of each (ws, wd) bin, the contribution of each concentration mean for a given (ws, wd) bin on the total concentration is also calculated, after applying a square root transformation on the $CO_2$ concentration to reduce any bias from the highest $CO_2$ values. We also show the

mean 1-σ standard deviation of the $CO_2$ concentration at each bin. A similar figure for nighttime data is given in the supplementary material S5a. During daytime (nighttime), the color scale is limited to the 380-430 ppm interval for the $CO_2$ concentration and to the 0-5 ppm range for the standard deviation. There are a few values outside of these ranges that are




forced to the closest range bound value. To facilitate the comparison between the stations, the highest complete wind speed circle visible on the plots is set at 10 m s$^{-1}$ in all cases. For MON, GON and GIF, all the data are plotted when taking this wind speed threshold. For TRN and EIF, wind speeds can reach higher values due to the elevation of these stations (during the afternoon: up to 15 m s$^{-1}$ at TRN and 25.5 m s$^{-1}$ at EIF; at night: up to 15 m s$^{-1}$ at TRN and 22 m s$^{-1}$ at EIF). Although

they represent only a minor fraction of the datasets, some of the TRN and EIF data are thus not apparent on Fig. 12: the plots for the full wind speed ranges encountered at EIF and TRN are given in the Supplementary materiel S5b (daytime) and S5c (nighttime).

**Influence of remote emissions**

The back trajectories (S1) show that Paris was exposed to a range of synoptic air masses over the period of study, including

clean oceanic ones and others with $CO_2$ enriched by remote anthropogenic emissions especially from the Benelux, the Ruhr area and the London megacity. Relatively high $CO_2$ concentrations (> 410 ppm) were observed for high wind speeds (> 8 m s$^{-1}$) in the 0-45° NNE sector at the 3 stations located relatively close to the ground level (MON, GON and GIF). For the elevated stations (EIF and TRN), such concentration values also occur, but as expected at higher wind speeds (> 12-14 m s$^{-1}$), reaching at least the 410 to 420 ppm range at all of the stations. The fraction of data falling in these (ws, wd) bins is large

enough to consider these high concentration values to be statistically representative. Furthermore, the standard deviation of the signal at the upwind stations is quite low (less than 0.6 ppm), which indicates that the high concentration values observed upwind of Paris (GON and MON) are not associated with fresh emissions, but with imported pollution that was already well-mixed in the atmosphere. It is likely that we see here the signature of remote anthropogenic $CO_2$ emissions from hot spots such as the Benelux and the Ruhr areas that bring higher $CO_2$ background levels to all the stations. The high $CO_2$

concentrations observed in the 0-35° NE sector at the downwind stations (EIF, GIF and TRN50) for moderate to high wind conditions (≥ 3 m s$^{-1}$) appear thus to be due not only to the Paris $CO_2$ emissions plume, but also to enriched background $CO_2$ levels advected from the NE. By comparison, the background levels that are observed in the 200° (SE) to 280° (NW) sector of GIF and TRN50 are lower than 400 ppm, while the 0-35° NE background levels at GON and MON are often above 400 ppm, reaching concentrations in the 410-430 ppm range. This shows that the Paris megacity background values can vary by

several ppm depending on the wind direction, with the highest $CO_2$ concentrations advected in the 0-45° wind cone. We note also that EIF shows higher concentrations in the 295-360° NW sector at high wind speeds that could be associated with long-range transport of anthropogenic plumes from the northern emissions hot spots emissions mentioned and better seen at this elevated station. Also, TRN shows higher $CO_2$ concentrations in the 345-360° NW sector for high wind speeds, that could be attributed to these hot spots - but also to Paris.

During nighttime, for wind speeds higher than 8 m s$^{-1}$ all stations show higher $CO_2$ levels in the 0-45° NE sector than in the other wind directions (see Fig.S5a).

**Influence of local emissions**

In section 3.2.1, we questioned whether MON was under the strong influence of local signals. The MON $CO_2$ wind rose shows that for wind speeds in the 0-2 m s$^{-1}$ range, higher $CO_2$ concentration (400 ppm to more than 430 ppm) are observed



in different wind sectors. Note the 230°-240° SW sector, where the bin contribution is the highest (~0.8-1%). Since there is no known surface source of $CO_2$ near the MON station, these higher $CO_2$ concentrations can most likely be attributed to the influence of aircraft emissions. Indeed, Montgé-en-Goële is located in the path of aircraft departing from CDG for easterly winds and of aircraft arriving to that airport for westerly winds (http://www.advocnar.fr/Fluxdetrajectoires.html). The CDG platform is equipped with two runways (North and South) from which the planes both take off and land along two W-E axis and pass very close the station at altitudes between 0 and 1000 m AGL. The NW and SE sides of the station are exposed to aircraft flying respectively to and from the CDG northern runway, while the 260°-360° sector and the 180°-260° sectors are the most exposed to aircraft traffic from the southern runway. Tarmac and in flight aircraft traffic (below 915 m ASL) are estimated to represent ~60% of the airport emissions (ADP, 2013). Taking road traffic emissions from and to CDG apart, the airport infrastructure itself (building heating, stopover airplanes electricity supply…) could also influence the station (as it represents ~11% of the airport $CO_2$ emissions; ADP, 2013), although more likely at the regional scale (see below). A much weaker influence of the Le Bourget aircraft flight paths, passing a few km southern than CDG airplanes but also at low altitude, is also possible in the southern side of the station.

In sections 3.2.1 and 3.4, we questioned the influence of local sources on GON (such as CDG and Le Bourget airports), even in the NW sector of the station. Indeed, GON is also exposed to aircraft emissions as it lies close to the lowest flight paths (0-1000m AGL) from the CDG and Le Bourget airports (http://www.advocnar.fr/Fluxdetrajectoires.html). These emissions are due: (i) in the NW sector, to takeoffs from the CDG northern runway; (ii) in the SW sector, to takeoffs from the CDG southern runway and from Le Bourget runway; (iii) in the NE sector, to landing on both CDG runways; and (iv) in the SE sector, to landings on the southern runway of CDG and to a lesser extent on Le Bourget airport. Also, it is likely that GON gets exposed to emissions from the two airports themselves, located a few km away. Note that the standard deviation that stands higher than 1 ppm from 60° (NE) to 170° (SE) seems to indicate fresher emissions in this wind sector. Nearby highways (located about 1.2 km north and east) could contribute in these wind directions, but discriminating between the different emission sectors would require measurements of carbon isotopes and specific emissions tracers.

At EIF, the influence of local emissions is expected mostly between the late morning and the late afternoon since, as we have seen in section 3.2.2, the top of the Eiffel tower receives surface emissions in this time period during all seasons. The $CO_2$ pollution rose of Fig. 12 indicates high concentrations (400 ppm to more than 430 ppm) in all directions around the stations for wind speeds comprised between 0 and 2 m s$^{-1}$. The variability is quite large (1.5 to 5 ppm) indicating fresh emissions and reflecting the spatial and temporal variability of the emissions coupled to atmospheric transport variations. Carbon isotopes and $CO_2$ co-emitted species measurements would be useful here to estimate the role of the different emission sectors.

In GIF, a few high $CO_2$ spots are observed for low wind conditions in diverse wind directions. These spots are likely due to emissions from traffic and heating from the surrounding infrastructures, as observed from the corresponding relatively high standard deviation (> 5 ppm). Flight paths to and from Orly airport for westerly winds pass several km south of the station and likely have a weak local impact.



Similarly to what is observed at GIF, higher $CO_2$ concentrations are observed at TRN50 in the wind sector of the city of Orleans, located ~13 km SW of the station.

During nighttime, MON and GIF show a higher local influence that still remains moderate. At EIF, no specific local influence is observed apart from a couple of (ws, wd) bins, confirming that the station is quite disconnected from the surface where urban emissions are diluted into the nocturnal layer. At GON, the influence of local emissions is strongly evident, with $CO_2$ concentrations reaching greater than 460 ppm and standard deviation greater than 5 ppm. In the 2-3 m s$^{-1}$ range, the station shows the highest $CO_2$ concentration in the direction of the CDG airport, a source that seems to have an impact on GON even at night. Indeed, CDG is one of the only airports in Europe to have nocturnal activity. TRN seems to be less influenced by local emissions than during daytime. Indeed, TRN not being impacted by Paris urban heat island, the nocturnal boundary layer is very shallow there so that the 50 m level is probably often decoupled from fresh emissions during the night (Pal et al, 2012).

At all stations, except for a few points in the SW sector at MON and GIF, the bin contribution of the data recorded for wind speeds in the 0-3 m s$^{-1}$ range is quite low, which indicates that generally the low wind conditions do not weight data very much during the period of study and during daytime. However, since local sources can be relatively strong, for regional studies these local influences should be removed by filtering out the $CO_2$ concentrations collected at wind speeds lower than 3 m s$^{-1}$.

**Influence of regional emissions**

Most of the data correspond to wind speeds comprised between 3 and 8 m s$^{-1}$, values for which we expect the regional influence of the Paris megacity on the downwind observed $CO_2$ concentrations to be the highest.

In the 0°-45° (NNE) sector, we observe relatively high $CO_2$ signals (>400 ppm) and low standard deviation values, even in stations upwind of Paris (GON and MON). In MON, the $CO_2$ concentrations in this wind sector are even higher than the ones in the SW sector which is expected to be exposed to the Paris emissions plume. This large NE signal can be attributed to the impact of remote emissions advected from that wind sector, as observed for higher wind speeds. In EIF and GIF (over and downwind of Paris in that wind sector), the $CO_2$ concentration reaches even higher values (>430 ppm, especially in EIF), which indicates the additional impact of the urban regional emissions. The contribution of each (ws, wd) bin is in the 0.4-1% range and is thus significant. These high concentrations are associated with high standard deviations (> 1 ppm, and even > 5 ppm at EIF), which results both from the high spatial and temporal variability of fresh emissions at the surface and from small scale dynamic effects in the ABL such as turbulence (succession of updrafts bringing polluted air to the station and downdrafts bringing cleaner air). In TRN50, there are some bins where the signal is higher than in MON and GON, but overall, the $CO_2$ concentration is lower, indicating that the Paris plume does not pass the TRN tower (50 m level) very often.

In the 45-90° (ENE) sector, all stations but EIF show $CO_2$ concentrations mostly in the 390-400 ppm range with some bins in the 400-410 ppm range. EIF shows more bins in the 400-410 ppm range, showing a higher exposure to urban emissions. However, while the standard deviation is relatively low in MON and TRN50, this is not the case at the GON, EIF and GIF stations, likely due to a higher proximity to sources of emissions, that, for GON include the CDG airport.





In the SW wind direction, stations upwind of the Parisian emissions (TRN50 and GIF) mostly show $CO_2$ concentrations in the 380-400 ppm range. In EIF, and even more in GON, we observe higher $CO_2$ values reaching the 400-410 ppm range. Indeed, due to its geographical position, EIF is less exposed to Parisian emissions in this wind sector, while GON is directly downwind of Paris for the 175-235° wind sector, where the largest point contribution reaches 1.6%. The standard deviation in EIF is above 1 ppm although lower than in the NE sector, while it is less than 1 ppm in GON, indicating that the emissions were mixed before arriving at the station. The MON station does not show specifically higher $CO_2$ concentrations compared to the upwind GIF station, except in the direction of the CDG airport. This latter source seems to have more impact on the station than the Paris emissions plume, which does not appear to often advect to the station.

In the NW wind sector, all stations except EIF are mostly in the 390-400 ppm range, with some values in the 400-410 ppm range (like in the 45-90° sector or NNE sector). EIF exhibits higher concentrations in the 325-360° sector, with values often in the 410-430 ppm range, and even reaching more than 430 ppm. The associated standard deviation is also very high at EIF, in the 2-5 ppm range and even more, indicating that emissions from the NW of Paris strongly impact this station. On the contrary, the variability stands mostly below 1 ppm in the other stations. The highest values are observed at GIF in the 305-325° direction, which could be explained by the station receiving emissions from the Saint-Quentin-en-Yvelines conurbation that is located 10-15 km upwind of GIF in those wind directions.

In the SE wind sector, for moderate wind speeds the MON, GIF and TRN50 stations show $CO_2$ concentrations mostly below 400 ppm and a few (ws, wd) bins in the 400-410 ppm range, especially in GIF for the 3-4 m s$^{-1}$ range and in the 90-135° sector. This sector comprises the southern branch of the extension of the Paris megacity which likely impacts the station. It is surprising though, that the 70-85° (ENE) sector does not show similar concentration ranges as it is urbanized at a similar level. At GON, the station is mostly sensitive to emissions in the 135-180° (SSE) sector although the standard deviation is quite low indicating these emissions are not from nearby sources as they are already mixed into the atmosphere. The EIF signal is as high as in the NW sector, very variable from one wind direction to the next and shows a high standard deviation, again reflecting the large variability of surface emissions and possibly the impact of atmospheric turbulence on the observations.

During nighttime, MON exhibits the highest $CO_2$ concentrations in the 0-45° (NNE) sector with values reaching the 410-420 ppm range. Those higher concentrations probably correspond to the continental background signals of polluted air masses advected from the Benelux and Ruhr areas. At GON, the $CO_2$ concentration reaches similar values but in all directions, showing on top of higher NE background values an impact of the regional urban emissions. Like during daytime, EIF shows higher concentrations in the urbanized sectors upwind of the station (NE, SE and NW mainly), although the concentrations stay mostly below 410 ppm - as a result of the decoupling from surface emissions during nighttime. At GIF, the highest concentrations are encountered, like during daytime, mostly in the NE sector that is the most exposed to Paris emissions. At TRN some (ws, wd) bins show higher $CO_2$ concentration in the NE sector, although this remains at a moderate level. The levels of the standard deviation confirm these observations and the data distribution plots show that generally most of the regional signal is contained into the 3-6 m s$^{-1}$ range.





## 4 Conclusions

This work forms the first study of ~1-year of measurements of atmospheric $CO_2$ in the region of the Paris megacity. We analyzed the $CO_2$ diurnal, synoptic and seasonal variability at five stations in that region and carried out a comparison with

the $CO_2$ dataset recorded at the MHD remote continental site.

In all stations of the Paris network, the influence of anthropogenic emissions, biospheric fluxes, atmospheric dynamics and synoptic wind patterns were shown to be key factors of the diurnal, weekday/weekend and seasonal variability of the atmospheric $CO_2$ concentrations.

At low wind speed, the stations receive local emissions, leading to a build-up of the $CO_2$ concentration, especially over Paris

at the top of the Eiffel tower during daytime and at the peri-urban station of Gonesse, where the concentration increase can reach up to 60 ppm. For wind speed values comprised between 3 and 9 m s$^{-1}$, advection leads to a decrease of the $CO_2$ concentration at all stations by ventilation of the emissions. For wind speeds higher than 9 m s$^{-1}$, as it was mentioned in former urban studies, the $CO_2$ concentration tends toward a mean constant value. However, contrary to previous studies, we showed that this value is different at each site and increases with the level of urbanization surrounding the station, leading to

a gradient of a few ppm between upwind and downwind stations. We argued that this value is based only on sparse meteorological events so that it cannot be defined as an asymptotic value, nor should it be used as a regional background.

Our work shows large diurnal and seasonal differences in the $CO_2$ concentration between the MHD site and the Paris upwind sites, as advected air masses undergo the influence of sources and sinks of $CO_2$ encountered on their footprint before reaching the megacity. This demonstrates that such a remote coastal site should not be used as a background site to infer

atmospheric $CO_2$ signals from urbanized regions located several hundreds of kilometers away, as it was done in some previous studies. This was also highlighted by Turnbull et al (2015) when analyzing atmospheric $CO_2$ variability in the Indianapolis region. Furthermore, even at high wind speeds, higher $CO_2$ concentrations (up to several ppm) are observed for air masses advected from the 0-45° NNE sector at all of the regional stations, compared to those advected from the SW sector, highlighting the impact of anthropogenic emissions from remote hot spots like Benelux and the Ruhr valley on the

Paris region $CO_2$ background in the NNE sector. Indeed, the average $CO_2$ concentrations measured at a given station when it is located downwind the Paris megacity are not always higher than the concentrations measured at that same station when it is located upwind, and this concerns both the hourly, diurnal and seasonal averages. This shows that the $CO_2$ concentration advected from the polluted 0-45° NNE sector can overtake the sum of the $CO_2$ plume out coming from Paris for SW winds and of the relatively low SW oceanic $CO_2$ background signals. This leads to the conclusion that when developing future

urban $CO_2$ networks, efforts must be made to carefully set-up several regional background sites on the path of the different dominant wind directions and ideally at the peri-urban/rural border of the city to constrain its signal as much as possible.



Ideally, the network will also be designed to position the urban and peri-urban downwind sites on these same wind directions axes.

Furthermore, our analysis shows the strong coupling that exists between the $CO_2$ concentration diurnal cycle and the boundary layer height cycle at the elevated stations and especially at EIF. We also highlighted how the high variability observed at EIF in the afternoon reflects the coupling of the highly variable urban emissions in the vicinity of the station with fluctuations of the wind speed and direction but also possibly with atmospheric fine scale dynamic processes. These results have consequence on the assimilation of the EIF data for inverse modelling purposes. Tall towers have been for several years the first choice in matter of sites selection for studying atmospheric $CO_2$ at the regional to the continental scales (e.g. Andrews et al., 2014; Haszpra et al., 2015; Gloor et al., 2001; Vermeulen et al., 2011), but their use for understanding $CO_2$ in urban environment seems to be more complicated as this requires to properly represent the underlying dynamic processes (including turbulence) that occur inside the boundary layer, and their coupling with the highly variable ground anthropogenic $CO_2$ emissions. For these reasons, we are for now not able to use data from EIF in our inverse modelling framework (Bréon et al., 2015). Improving our sampling system on the Eiffel tower to gather vertical $CO_2$ profiles and meteorological data will be of great help in the future. This recalls as well that the altitude relative to ground level and the distance to the emissions of a station are very important factors to take into account in the network capacity to properly detect a $CO_2$ urban plume (see also the discussion about this topic in Boon et al., 2015).

The fine classification of the $CO_2$ concentrations collected at each site following wind directions and wind speeds allowed us to better define the footprint of each station and the impact of local, regional and remote $CO_2$ fluxes on each station. In each of the regional sites, the high $CO_2$ concentrations observed at low wind speeds (<3 m s$^{-1}$) revealed the impact of local sources including likely emissions from aircraft and airports. For moderate wind speeds (3 to 9 m s$^{-1}$), the impact of the $CO_2$ emissions of Paris is clearly seen at urban and peri-urban stations (GON, EIF and GIF) in the afternoon, and much less at night. This impact however is barely seen in the two rural stations (MON and TRN), and ultimately do not seem to be relevant sites to study the $CO_2$ emission plume from the Paris megacity.

At each station, the minimum of the seasonal cycle amplitude was found in summer due to high photosynthesis, lower anthropogenic emissions and higher ABL height. The maximum of the $CO_2$ seasonal cycle was found in winter when the biospheric activity reaches its minimum, the Paris anthropogenic emissions get to their maximum and the ABL height is at its lowest. However, we could not separate the anthropogenic and biospheric $CO_2$ signals, nor the role of the different emission sectors. This highlights the need for regular carbon isotopic measurements of $CO_2$ at the regional network stations, together with measurements of anthropogenic co-emitted species such as CO, NOx, black carbon and volatile organic compounds (e.g. Lopez et al., 2013; Ammoura et al., 2014; Ammoura et al., 2015). Finally, we show that ancillary data such as local meteorological data and parameters defining the structure of the atmosphere such as the ABL height are very important to understand the observed $CO_2$ variability. Ideally, such measurements should also be included in the development of urban $CO_2$ monitoring networks. The $CO_2$ datasets presented here provide the basis for a study conducted on atmospheric inversion modeling of the Paris $CO_2$ emissions (Staufer et al, 2016).




**Supplement link**

**Acknowledgments**

This work was mostly funded by the Agence Nationale de la Recherche (ANR) in the framework of the $CO_2$-Megaparis project and partly by the Ville de Paris through the "Le $CO_2$ parisien" (Paris 2030) project. We deeply acknowledge

AIRPARIF technical team for the maintenance of the $CO_2$-Megaparis stations. The authors are very grateful to the RAMCES-ICOS team and they also thank Sandip Pal for technical help. The GIF and TRN stations are funded by INSU and CEA (SNO RAMCES/ICOS). Most of the figures shown in this work were produced with the Openair package for R (Carslaw and Ropkins, 2012; Carslaw, 2015). We especially acknowledge David Carslaw (Openair package) for helpful advices. We thank very much SPR at CEA Saclay for providing us with the meteorological measurements. The first author

sends warm acknowledgments to Peter Rayner and Thomas Lauvaux for their scientific advices in building-up the $CO_2$-Megaparis project, and to Cecilia Garrec and Peter Rayner (once again) for their help in coordinating it. Special thanks to Steve Wofsy for his support and to Chris Rella from the Picarro company for his help with the CRDS analyzers.

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





**Figure captions**

Figure 1. Annual emissions of $CO_2$ from Île-de-France at a spatial resolution of 1x1 km$^2$ (AIRPARIF, 2010) and our Paris megacity $CO_2$ in-situ network: the red points indicate the $CO_2$-MEGAPARIS stations (MON, GON and EIF); the dark blue points are stations from the ICOS-France network (GIF, TRN). The QUALAIR station for monitoring the atmospheric boundary layer height in the Paris city is also shown (green point).

Figure 2. Location of the Paris megacity on a map of $CO_2$ anthropogenic emissions from Western Europe, adapted from the Edgar 2009 inventory (http://edgar.jrc.ec.europa.eu/, 2009). Emissions are given in Tg of $CO_2$-eq per grid cell (10 x 10 km$^2$). Some of the main emitting points in Western Europe are also given. The geographical position of the remote site of Mace Head (MHD) on the west coast of Ireland is also shown.

Figure 3. Wind rose at GIF given by season over the period of study (8 August 2010–13 July 2011) from the Meso-NH wind fields. Colors indicate the wind speed according to the given scale (in m s$^{-1}$).

Figure 4. Seasonal variation of the temperature at SAC (100 m AGL) close to the GIF station (hourly averages) on the period of study (8 August 2010 –13 July 2011).

Figure 5. Mean $CO_2$ diurnal cycles at the different sites of the Paris regional network and MHD averaged on the whole period of study (8 August 2010–13 July 2011) and computed from hourly $CO_2$ concentrations.

Figure 6 (a to d'). Left: Diurnal cycles of $CO_2$ from 1 h averages at (a) MON, (b) GON, (c) EIF and (d) GIF. Right: Diurnal cycles of $CO_2$ by season at (a') MON, (b') GON, (c') EIF and (d') GIF. Note that the left and right plot scales are not the same.

Figure 6 (e to g'). Left: Diurnal cycles of $CO_2$ from 1 h averages at: (e) TRN50, (f) TRN180 and (g) MHD. Right: Diurnal cycles of $CO_2$ by season at: (e') TRN50, (f') TRN180 and (g') MHD. Note that the left and right plot scales are not the same.

Figure 7. Diurnal cycles of the hourly LBLH estimate means (±1-σ) and of $CO_2$ hourly means observed by season at QUALAIR and EIF, respectively. Time is in hour UTC. The horizontal line is the elevation of EIF. The violet circles give the $CO_2$ concentration (according to the red scale) at the moments when the LBLH was measured as well.





Figure 8. Left: $CO_2$ diurnal cycle by day of the week at the different stations, calculated from $CO_2$ hourly concentrations over the whole period of study. Right: standard variation (1-σ) of the hourly $CO_2$ mean concentration.

Figure 9a. Seasonal cycles of $CO_2$ concentration at the six sites based on monthly means. Monthly averages of air temperature at 100 m (Saclay tower near GIF) and of the LBLH (Jussieu) are also shown.

Figure 9b. Seasonal cycle (Aug.2010-Jul.2011) of $CO_2$ at each of the Paris regional sites and at MHD, calculated from $CO_2$ monthly means of hourly averages, with error bars showing one standard deviation (±1-σ) of the $CO_2$ means.

Figure 10. Left: Hourly means of the $CO_2$ concentration recorded at GON as a function of wind speed and colored by wind direction (the color scale is in degrees). Right: same for the $CO_2$ standard deviation (1-σ of the hourly $CO_2$ concentration means).

Figure 11. Mean $CO_2$ concentration (in ppm) observed at the different stations of the Paris regional network (TRN represents the measurements at 50 m AGL) and at MHD for wind speed higher than 9 m s$^{-1}$ over the period of study (8 August 2010–13 July 2011). During such events, the synoptic conditions were mostly oceanic (wind blowing from the SW sector).

Figure 12. Left: $CO_2$ mean concentration as a function of wind speed (circles in m s$^{-1}$) and wind direction at MON, GON, EIF, GIF and TRN50 stations using daytime data (11-15 h UTC) for the whole period of study (4 Aug.2010-11 July 2011). Middle: mean 1-σ $CO_2$ variability of each concentration (ws, wd) point. Right: occurrence as the frequency of the (ws, wd) bin weighted by the square-root of the $CO_2$ concentration mean.





**Figures**

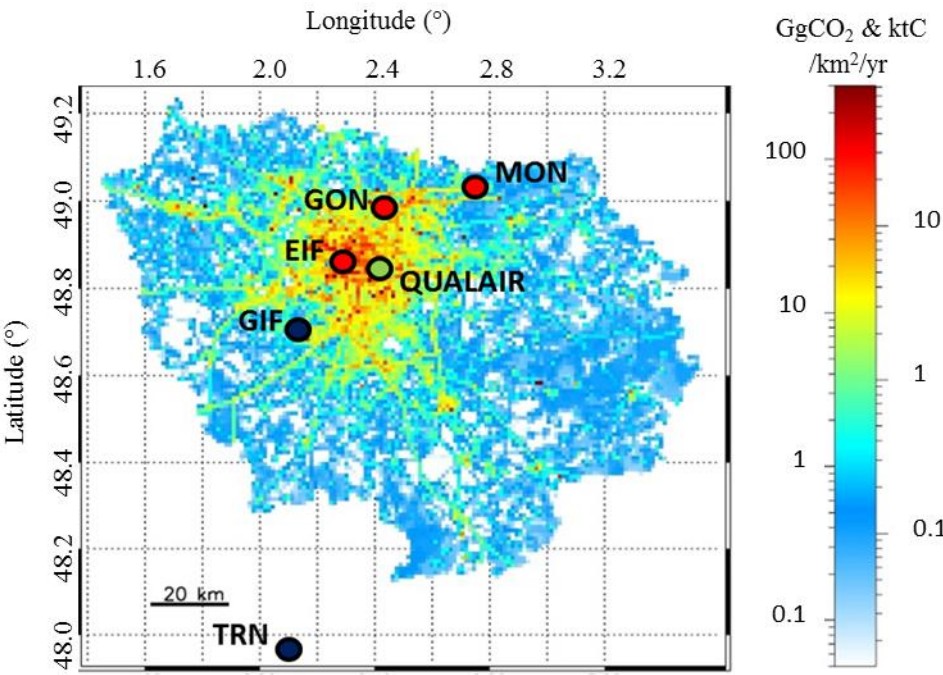

Figure 1. Annual emissions of $CO_2$ from Île-de-France at a spatial resolution of 1x1 km² (AIRPARIF, 2010) and our Paris megacity $CO_2$ in-situ network: the red points indicate the $CO_2$-MEGAPARIS stations (MON, GON and EIF); the dark blue

10  points are stations from the ICOS-France network (GIF, TRN). The QUALAIR station for monitoring the atmospheric boundary layer height in the Paris city is also shown (green point).



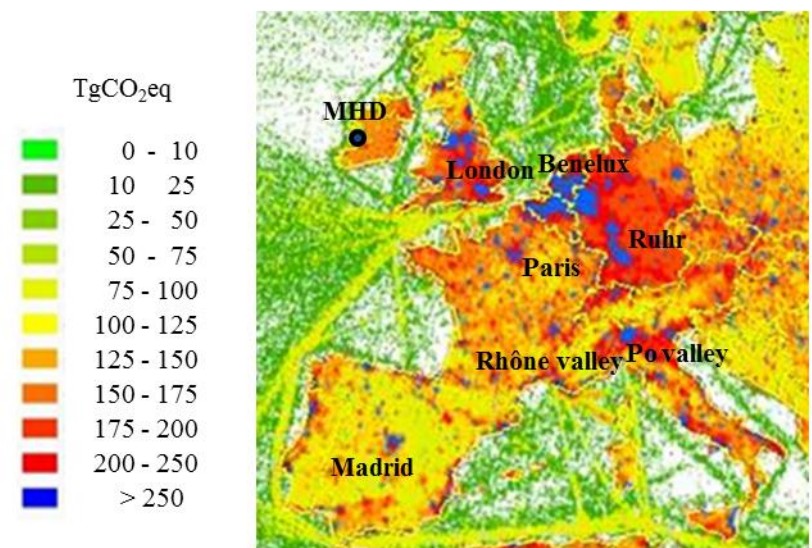

Figure 2. Location of the Paris megacity on a map of $CO_2$ anthropogenic emissions from Western Europe, adapted from the
10   Edgar 2009 inventory (http://edgar.jrc.ec.europa.eu/, 2009). Emissions are given in Tg of $CO_2$-eq per grid cell (10 x 10 km$^2$).
Some of the main emitting points in Western Europe are also given. The geographical position of the remote site of Mace
Head (MHD) on the west coast of Ireland is also shown.



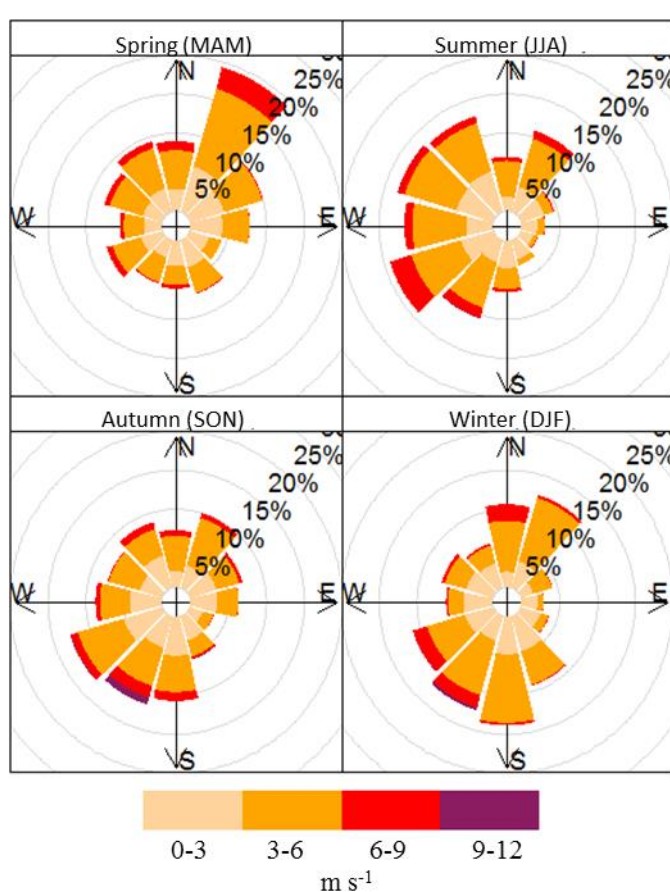

Figure 3. Wind rose at GIF given by season over the period of study (8 August 2010–13 July 2011) from the Meso-NH wind fields. Colors indicate the wind speed according to the given scale (in m s[-1]).

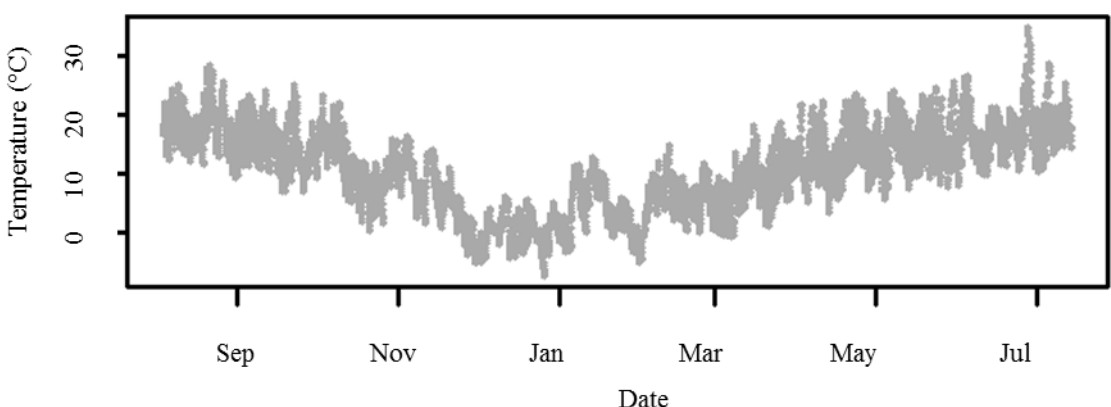

Figure 4. Seasonal variation of the temperature at SAC (100 m AGL) close to the GIF station (hourly averages) on the period
10    of study (8 August 2010 –13 July 2011).





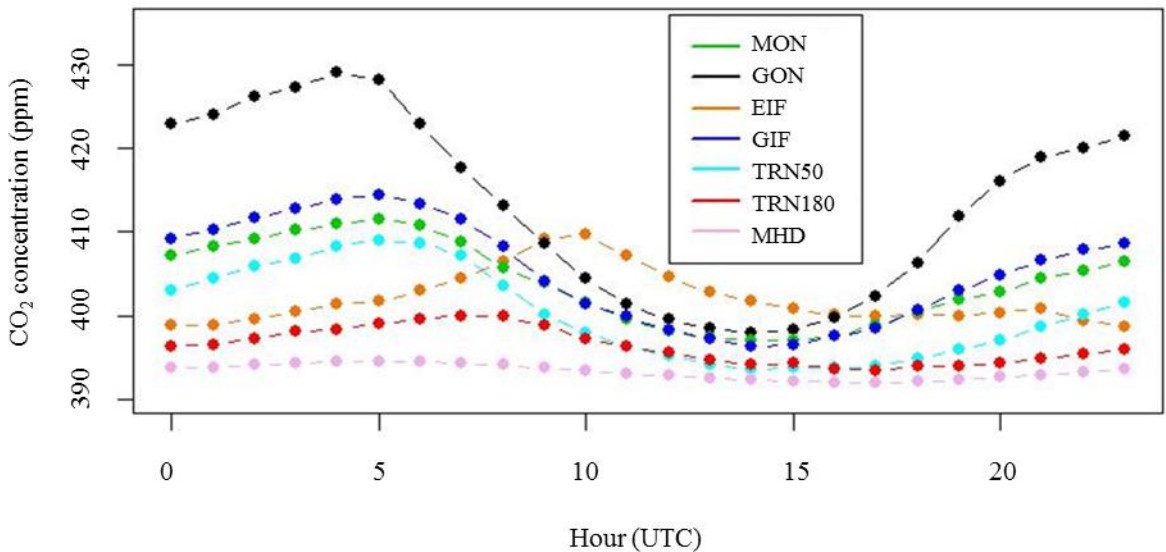

10  Figure 5. Mean $CO_2$ diurnal cycles at the different sites of the Paris regional network and MHD averaged on the whole period of study (8 August 2010–13 July 2011) and computed from hourly $CO_2$ concentrations.







Figure 6 (a to d'). Left: Diurnal cycles of $CO_2$ from 1 h averages at (a) MON, (b) GON, (c) EIF and (d) GIF. Right: Diurnal cycles of $CO_2$ by season at (a') MON, (b') GON, (c') EIF and (d') GIF. Note that the left and right plot scales are not the same.



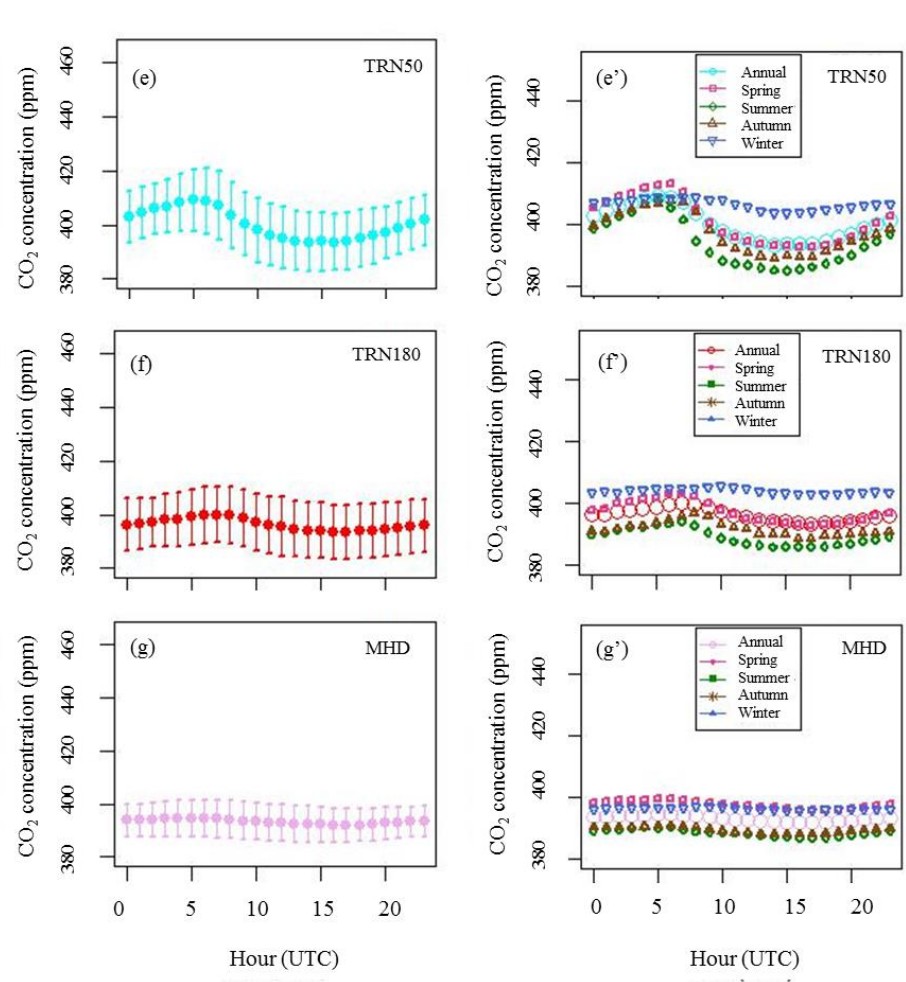

5    Figure 6 (e to g'). Left: Diurnal cycles of $CO_2$ from 1 h averages at: (e) TRN50, (f) TRN180 and (g) MHD. Right: Diurnal cycles of $CO_2$ by season at: (e') TRN50, (f') TRN180 and (g') MHD. Note that the left and right plot scales are not the same.

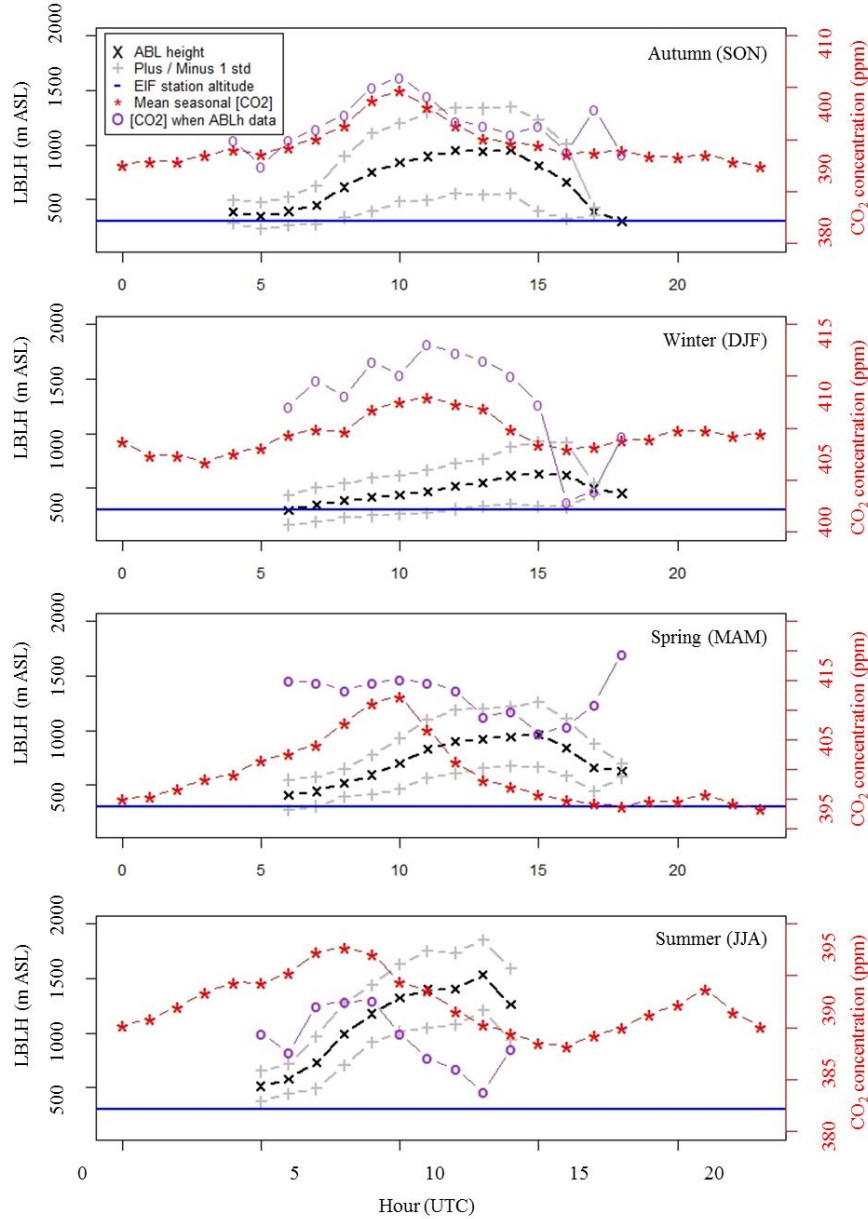

Figure 7. Diurnal cycles of the hourly LBLH estimate means ($\pm 1$-$\sigma$) and of $CO_2$ hourly means observed by season at QUALAIR and EIF, respectively. Time is in hour UTC. The horizontal line is the elevation of EIF. The violet circles give the $CO_2$ concentration (according to the red scale) at the moments when the LBLH was measured as well.




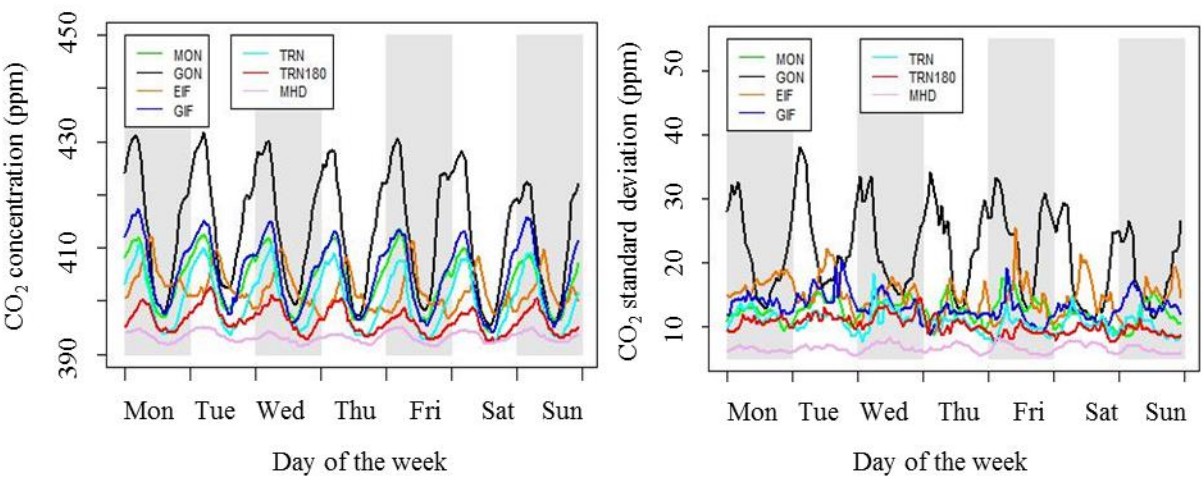

Figure 8. Left: $CO_2$ diurnal cycle by day of the week at the different stations, calculated from $CO_2$ hourly concentrations over the whole period of study. Right: standard variation (1-σ) of the hourly $CO_2$ mean concentration.





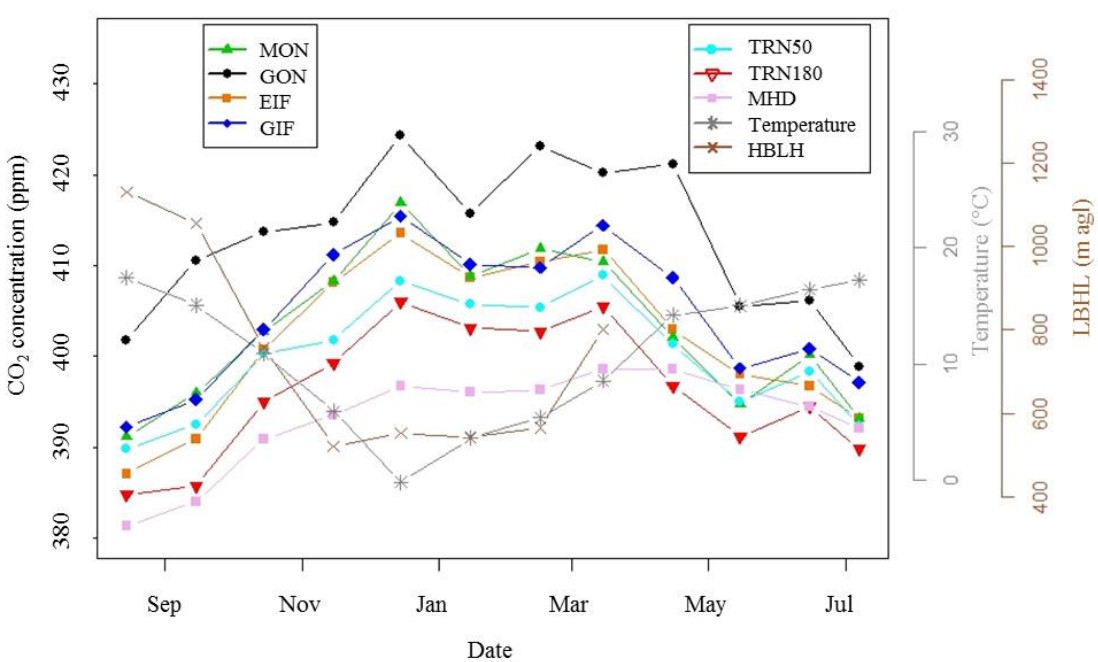

Figure 9a. Seasonal cycles of $CO_2$ concentration at the six sites based on monthly means. Monthly averages of air temperature at 100 m (Saclay tower near GIF) and of the LBLH (Jussieu) are also shown.





Figure 9b. Seasonal cycle (Aug.2010-Jul.2011) of $CO_2$ at each of the Paris regional sites and at MHD, calculated from $CO_2$ monthly means of hourly averages, with error bars showing one standard deviation ($\pm 1$-$\sigma$) of the $CO_2$ means.



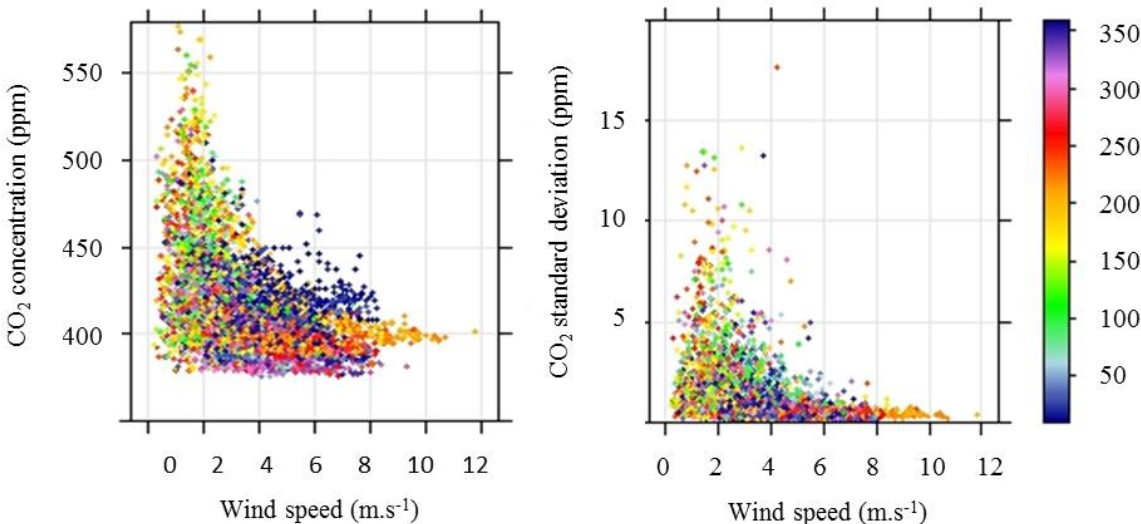

Figure 10. Left: Hourly means of the $CO_2$ concentration recorded at GON as a function of wind speed and colored by wind direction (the color scale is in degrees). Right: same for the $CO_2$ standard deviation (1-σ of the hourly $CO_2$ concentration means).




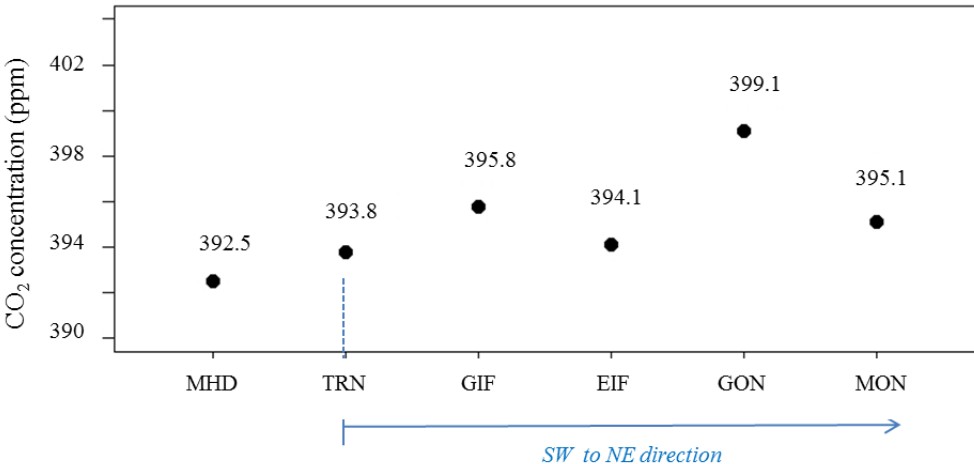

Figure 11. Mean $CO_2$ concentration (in ppm) observed at the different stations of the Paris regional network (TRN represents the measurements at 50 m AGL) and at MHD for wind speed higher than 9 m s[-1] over the period of study (8 August 2010–13 July 2011). During such events, the synoptic conditions were mostly oceanic (wind blowing from the SW sector).



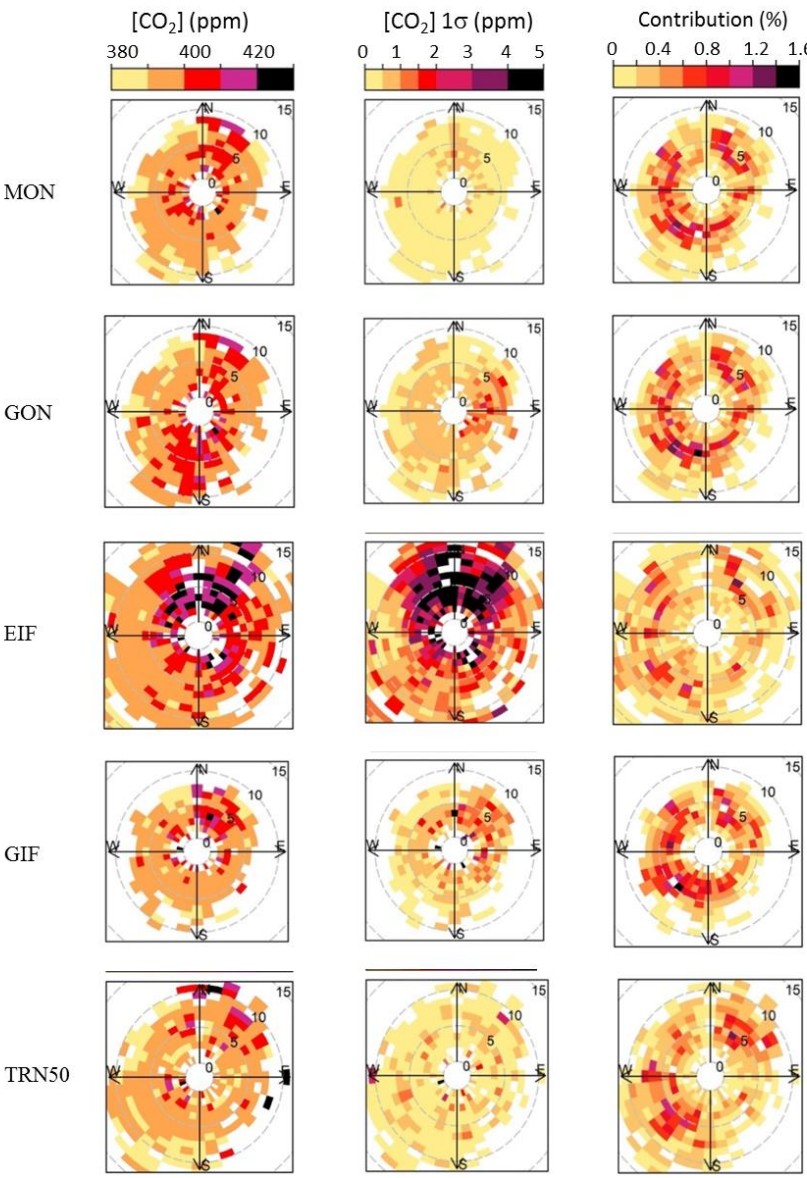

Figure 12. Left: $CO_2$ mean concentration as a function of wind speed (circles in m s$^{-1}$) and wind direction at MON, GON, EIF, GIF and TRN50 stations using daytime data (11-15 h UTC) for the whole period of study (4 Aug.2010-11 July 2011). Middle: mean 1-$\sigma$ $CO_2$ variability of each concentration (ws, wd) point. Right: occurrence as the frequency of the (ws, wd) bin weighted by the square-root of the $CO_2$ concentration mean.



**Tables**

Table 1. Coordinates of the stations used in this study (ASL stands for Above Sea Level; AGL for Above Ground Level).

| Station | Code | Latitude (°) | Longitude (°) | Site ground elevation ASL | Sampling height AGL |
|---|---|---|---|---|---|
| Montgé-en-Goële | MON | 49°01'41.79'' N | 2°44'55.54'' E | 160 m | 9 m |
| Gonesse | GON | 48°59'24.56'' N | 2°27'21.90'' E | 68 m | 4 m |
| Eiffel tower | EIF | 48°51'29.71'' N | 2°17'39.92'' E | 33 m | 317 m |
| Gif-sur-Yvette | GIF | 48°42'35.82'' N | 2°08'51.55'' E | 163 m | 7 m |
| Traînou | TRN | 47°57'53.08'' N | 2°06'45.42'' E | 133 m | 50 m , 180m |
| Mace Head | MHD | 53°19'33.00'' N | 9°54'12.00'' W | 25 m | 15 m |
| QUALAIR | QUA | 48°50'47.26'' N | 2°21'21.40'' E | 35 m | 25 m |



Table 2. Calibration and target frequencies, accuracy and repeatability of the $CO_2$-Megaparis stations. The accuracy is given as the difference of the target $CO_2$ concentrations measured by the CRDS analyzer and by the GC.

|  | EIF | MON | GON |
|---|---|---|---|
| Calibration sequence | 2 h every 3 months | 6 h every 2 weeks | 6 h every 2 weeks |
| Target sequence | 30 mn every 2 weeks | 30 mn every 12 h | 30 mn every 12 h |
| Accuracy (ppm) | 0.13 | -0.04 | -0.07 |
| Repeatability (ppm) | 0.38 | 0.10 | 0.07 |



Table 3. Mean altitude of the lowest estimate of the boundary layer height (LBLH) by season in the morning and early afternoon (hours are given UTC, altitude in meters AGL). The number of points used to calculate the means are also given (N).

| Time (UTC) | 5 h | 6 h | 7 h | 8 h | 9 h | 10 h | 11 h | 12 h | 13 h |
|---|---|---|---|---|---|---|---|---|---|
| | | | | Spring | | | | | |
| LBLH | NaN | 410 | 442 | 520 | 593 | 697 | 833 | 899 | 935 |
| N | 0 | 9 | 11 | 11 | 12 | 12 | 12 | 13 | 13 |
| | | | | Summer | | | | | |
| LBLH | 513 | 583 | 728 | 992 | 1178 | 1324 | 1400 | 1405 | 1531 |
| N | 7 | 13 | 13 | 13 | 13 | 13 | 11 | 11 | 7 |
| | | | | Autumn | | | | | |
| LBLH | 351 | 394 | 451 | 615 | 751 | 837 | 896 | 947 | 940 |
| N | 16 | 25 | 31 | 34 | 33 | 33 | 33 | 31 | 30 |
| | | | | Winter | | | | | |
| LBLH | NaN | 301 | 349 | 384 | 419 | 440 | 470 | 516 | 550 |
| N | 0 | 3 | 15 | 24 | 23 | 25 | 26 | 27 | 29 |





Table 4. Monthly means and standard deviation (± 1-σ) of the $CO_2$ concentration (in ppm) measured at each site and data coverage of each month (N, in percent).

|  | MON | GON | EIF | GIF | TRN50 | TRN180 | MHD |
|---|---|---|---|---|---|---|---|
|  | | | | Spring | | | |
| March | 410.4±9.4 | 420.3±19.1 | 411.8 ±16.7 | 414.4±13.7 | 408.9±9.3 | 405.5±7.9 | 398.6±4.4 |
| N | 99.9 | 97.3 | 95.6 | 93.0 | 57.7 | 66.8 | 87.6 |
| April | 402.1±11.0 | 421.2±32.6 | 403.0±13.2 | 408.7±15.3 | 401.3±11.2 | 396.8±7.1 | 398.6±4.9 |
| N | 100.0 | 95.3 | 94.6 | 94.2 | 69.0 | 79.6 | 77.6 |
| May | 394.7±8.9 | 405.5±20.0 | 398.0±10.6 | 398.7±11.2 | 395.0±9.9 | 391.2±5.9 | 396.3±2.4 |
| N | 99.9 | 97.3 | 98.8 | 98.3 | 81.2 | 82.8 | 95.6 |
|  | | | | Summer | | | |
| June | 400.1±11.9 | 406.2±27.3 | 396.9±8.2 | 400.9±12.8 | 398.4±10.7 | 394.5±4.7 | 394.5±3.5 |
| N | 98.1 | 0.65 | 95.3 | 84.9 | 88.2 | 69.3 | 92.9 |
| July | 393.1±6.9 | 398.6±17.3 | 393.4±6.6 | 397.2±8.3 | 392.4±6.2 | 389.8±3.2 | 392.1±5.0 |
| N | 96.8 | 96.8 | 78.1 | 62.4 | 51.4 | 78.1 | 97.1 |
| August | 390.8±10.2 | 401.9±29.6 | 387.1± 7.9 | 392.2±11.8 | 389.8±10.8 | 384.9±5.6 | 381.4±2.5 |
| N | 99.6 | 94.6 | 90.5 | 78.6 | 95.8 | 96.1 | 99.9 |





|  | Autumn | | | | | | |
|---|---|---|---|---|---|---|---|
| September | 395.3±12.7 | 410.9±34.0 | 391.0±11.1 | 395.3±11.1 | 392.5±11.8 | 385.7±5.7 | 384.0±3.3 |
| N | 72.9 | 96.0 | 97.8 | 83.1 | 91.1 | 90.4 | 96.8 |
| October | 402.8±9.8 | 413.9±24.7 | 400.8±12.0 | 403.0±11.3 | 400.3±10.6 | 395.0±7.2 | 390.9±6.2 |
| N | 100.0 | 96.0 | 98.9 | 82.7 | 92.5 | 90.5 | 98.7 |
| November | 408.3±10.4 | 414.9±15.9 | 407.7±15.1 | 411.2±12.9 | 401.8±9.4 | 399.3±8.6 | 393.6±3.8 |
| N | 100.0 | 97.2 | 99.6 | 67.4 | 34.3 | 31.5 | 97.1 |
|  | Winter | | | | | | |
| December | 417.0±13.9 | 424.5±17.9 | 414.2±16.9 | 415.4±13.9 | 408.3±9.5 | 406.0±10.4 | 396.8±3.8 |
| N | 100.0 | 73.9 | 71.9 | 77.4 | 82.4 | 87.5 | 97.2 |
| Jan uary | 408.9±9.4 | 415.8±16.7 | 408.4±13.2 | 410.1±13.0 | 405.7±10.1 | 403.1±9.3 | 396.1±2.3 |
| N | 100.0 | 96.2 | 78.9 | 78.5 | 95.6 | 94.5 | 98.7 |
| February | 411.9±12.2 | 423.1±20.7 | 410.5±14.7 | 409.8±10.5 | 405.4±7.8 | 402.8±7.3 | 396.3±2.0 |
| N | 100.0 | 97.0 | 93.2 | 97.0 | 84.8 | 88.5 | 98.4 |