# Peer review of "Diurnal, synoptic and seasonal variability of atmospheric CO2 in the Paris megacity area"

_Atmospheric Chemistry and Physics, 2016_

## Referee Comment (RC1) · J. Turnbull (Referee) · 20 Jun 2016

This paper describes a year-long series of in situ CO2 measurements from sites in and around Paris. The paper focuses on how and why the CO2 signals vary: the proximity to the city; height of the inlet above ground; variability in emission sources; wind direction and speed. They demonstrate that in many wind regimes, emissions from upwind sources can contribute as much or more CO2 than local Paris emissions. They show that urban CO2 variability is complex, implying that a strong understanding of these factors and the particular sampling network is needed to infer the emission flux from such measurements. Of particular note is that the Eiffel Tower sampling site is challenging to interpret since the inlet height is only sometimes within the boundary layer.

[Figure]

This is a very nice, detailed examination of urban CO2 source variability that will be useful for the existing and upcoming urban greenhouse gas researchers. This research area is still in its infancy, and this study gives a very good demonstration of how urban sampling networks should be designed and the types of problems that can be encountered. This paper is entirely appropriate for publication in ACP. I see no major issues with the paper, and recommend minor revisions for clarity and language usage.

Specific comments:

The authors should edit the full paper for correct English grammar. I point out some specific words in further comments, but there are many other cases where the grammar is comprehensible but incorrect.

Abstract page 1 line 31. "elevated" is used here and in other places through the paper to mean "sites where the inlet is well above ground level". This is confusing though, because "elevated" is also commonly used to mean "the CO2 is higher than background". Perhaps "two sites with inlets high above ground level"?

Introduction pg 3 line 3 (and several other times in the paper). "conurbation" is not commonly used in English – I am a native speaker and had to look up the meaning. Perhaps "metropolitan area" would be a better choice.

Pg 5 ln 3-12. Are there any large points sources in the metropolitan area? You mention some in the next section, but it would be helpful to first give them in this section.

Pg 6 lines 13-31. Are there any emissions directly from the buildings you are sampling on top of?

Pg 7 line 16. You say that this station is ideally located, but don't give any justification as to why it is ideal.

Pg 8 ln 6-7. "Only the last calibration..." it is not clear what is meant by this sentence. Please clarify.

Pg 8 ln 16. Please give a reference for the ICOS procedure.

Pg 8 ln 20. How were the very local influences (that were removed) identified?

Pg 8 ln 26. Please reference the WMO-X2007 scale.

Pg 10 ln 4. Please provide a link or reference for the Met Eireann met data.

Pg 10 ln 23. What met dataset was used in HySplit?

Pg 11 ln 11. I don't see the 1-sigma std devs on the plot. Did you mean to refer to figure 6 here?

Pg 11 ln 15. Please provide references to previous work that has discussed the biosphere and vertical dilution impacts on CO2.

Pg 12 ln 2. "During daytime..." do you mean mid-afternoon?

Pg 12 ln 3. "significant positive gradient". Perhaps "enhancement" would be a better word. (Also used elsewhere in the paper).

Pg 12 ln 12-14. Why does the lack of diurnal cycle at MHD make it a poor choice for background? If you are interested in examining the urban anthropogenic CO2 source, then this is probably correct, but if you are interested in the diurnal variability of the continental biosphere signal, then it might be a good choice. Please explain/clarify.

Pg 12 ln 22-23. Can you give an estimate of the magnitude of the biospheric flux through the seasons. It would be helpful to know how large it might be relative to the fossil fuel flux (even though the biosphere flux might be poorly constrained).

Pg 13 ln 14. I think you mean figure 5 and 6, not figure 7.

Pg 15 ln 5-11. I don't see what this discussion of the vertical gradients adds to the paper. It could either be cut out, or a sentence added to explain why it is useful.

Pg 15 ln 13-26. The AIRPARIF inventory, I believe, is fossil fuel CO2 flux only, whereas you measure total CO2 (both fossil and bio). Could it be that the smaller weekday/weekend differences in your observation be due to the fact that biospheric fluxes are constant through weekdays and weekends? I.e. the difference between weekdays and weekends would be proportionally smaller in the total CO2 observations than in the inventory, if there is a large (and constant) biosphere flux. Could this also explain why the GIF signal is more consistent between weekdays and weekends? I.e. perhaps the biosphere contribution is relatively more important at GIF than the urban sites?

Pg 15 ln 28. Does this seasonal cycle include all or only some hours of the day?

Pg 16 ln 5-9. Please reference previous work that has discussed this phenomenon of seasonality in BL height, biosphere emissions and fossil fuel emissions. See for example: Denning, A. S., P. J. Rayner, R. M. Law and K. R. Gurney (1995). Atmospheric tracer transport model intercomparison project (TransCom). IGBP/GAIM report series report #4. D. Sahagian. Turnbull, J. C., P. J. Rayner, J. B. Miller, T. Naegler, P. Ciais and A. Cozic (2009). "On the use of 14CO2 as a tracer for fossil fuel CO2: quantifying uncertainties using an atmospheric transport model." Journal of Geophysical Research 114, D22302.

Pg 16 ln 17-21. Indeed, the CO2 signals are higher in the winter, but the standard deviations do not seem to be higher in winter. Elsewhere in the paper, the higher standard deviations are used to identify higher anthropogenic emissions. Please justify why this is not the case here.

Pg 17 ln 5-10. I don't think you can conclude that fossil fuel emissions are lower in summer from this dataset, since photosynthetic drawdown confounds the signal so strongly.

Pg 17 ln 21-24. Please explain and/or reference how the seasonal adjustment was performed. Reference previous work that discusses relationship between concentration and wind speed/ventilation.

Pg 18 ln 20-22. Please clarify what the relationship is that justifies using the different

wind speed regimes to identify local and remote emissions. Another sentence or two would help to follow the logic of doing this.

Pg 18 ln 25-29. Please expand this explanation a little more and/or reference the method, particularly for the square root transformation that has been applied.

Pg 20 ln 1-23. Exactly how close are MON and GON to CDG airport? Are there any other industrial or commercial facilities that could be causing this signal? In section 2.1.1. You stated that airport emissions are 4% of the total, whereas industrial emissions are 14%, so industrial emissions are potentially more important. Are CDG emissions large enough to plausibly explain the signal at both sites?

Pg 20 ln 22-23. How would carbon isotopes and specific emission tracers help to discriminate between airport and traffic emissions? Does jetfuel have a different isotopic signature than petrol/diesel?

Pg 21 ln 3-11. See also previous comment – are the CDG emissions large enough at night and close enough to plausibly influence the GON site so strongly?

It would be helpful to include Figure S2 in the main paper, since that shows the actual $CO_2$ data which is the main focus of the paper. If there is a limitation on the number of figures, Figures 3 and4 could move to the supplementary material (since the wind directions are also shown in figure S2).

Figure 5 is essentially repeated in figure 6. Could these two figures be combined?

Figure 9a and b could be combined by plotting 9a as an 8th panel in figure 9b.

Tables are mentioned in the text in a different order than the order of their numbering.

---

## Referee Comment (RC2) · Anonymous Referee #2 · 13 Sep 2016

Journal: ACP Title: Diurnal, synoptic and seasonal variability of atmospheric CO2 in the Paris megacity area Author(s): I. Xueref-Remy et al. MS No.: acp-2016-218 MS Type: Research article

General Comments

This paper analyzes nearly 1 year of CO2 data from the Paris megacity greenhouse gas measurement network. The analysis focuses on deciphering the CO2 observations on diurnal and seasonal time scales, and includes a careful examination of the influence of the atmospheric boundary layer height (ABLH), wind speed and direction, and local anthropogenic emissions on these signals. The measurement network contains six total sites across Ile de France spanning a range of conditions from rural to the Eiffel Tower in the heart of Paris. The report presents measurements that pro-

vide an important baseline for emissions from Paris and for comparison to other global megacities.

Specific Comments The authors present a detailed analysis of the CO2 observations based on time, location, and wind speed/direction to infer the seasonal influence of local and background contributions at each site. This analysis is largely qualitative, but could be made far more quantitative and definitive if based around back trajectory analyses, such as those shown in Figure S1. We strongly suggest that the discussion of Section 3.1 be expanded and used to validate the conclusions of Section 3.5 which appear to be based on site wind measurements.

The study concludes that the level of CO2 enhancement varies with urbanization level local to the site; however, the paper does not directly discuss estimates of enhancement relative to background (or the concept of background) until much later in the paper. While diurnal and seasonal variability and the gradients between sites are the primary focus of this paper, background estimation is an important topic and which merits more introduction. Overall, there are two key points that should be incorporated: (1) the concept of background should be defined relative to the domain of interest and (2) a single site may not represent background CO2 mole fractions under all meteorological conditions. Additionally, the paper should use CO2 enhancement values relative to some chosen background rather than absolute CO2 values (eg 410 ppm) since the global background will surpass even these "elevated" values in the near future.

The challenges of analyzing these measurements raises several priority questions regarding the Paris network. We note that the INFLUX network in Indianapolis, IN USA contains 13 towers for a smaller, less populated urban area and approximately 1/10th the emissions of Paris/IdF [Turnbull, Jocelyn C., et al. "Toward quantification and source sector identification of fossil fuel CO2 emissions from an urban area: Results from the INFLUX experiment." Journal of Geophysical Research: Atmospheres 120.1 (2015): 292-312]. We would have expected some discussion of the density of the Paris network, the potential benefit of additional sites, and where they would ideally be located for maximum impact. This is particularly relevant for the "background" discussion since it is clear that Mace Head alone is insufficient for this analysis and that a full understanding of Paris $CO_2$ monitoring may well require observations from as far away as the Ruhr or the Benelux region. Given the topographical similarities of Paris and Indianapolis, we were also surprised that more discussion was not presented comparing the $CO_2$ concentration "plume" patterns from these urban areas.

Newman et al. [Newman, S., et al. "Diurnal tracking of anthropogenic CO 2 emissions in the Los Angeles basin megacity during spring 2010." Atmospheric Chemistry and Physics 13.8 (2013): 4359-4372] showed diurnal patterns for $CO_2$ from the Los Angeles megacity, but there was no comparison made with these data. This is particularly relevant since Los Angeles $CO_2$ emissions are well known to be dominated by vehicle/transportation and impart significant rush hour maxima (0700-1000 and 1500-1900) that are absent from all but the EIF signals in Paris. The arguments for winter vehicle emissions in Paris are not obvious from the figures as presented.

The Eiffel Tower (EIF) site offers unique observations that might be more fully exploited in future studies. Complete diurnal and day of week sampling at this site would enable greater understanding of variability across the network. Adding vertical profile measurements at eg 50, 100, and 200 m to complement the 300 m inlet height would add tremendously to understanding the ABLH/$CO_2$ linkages as well as providing different spatial sensitivity footprints within the Paris/IdF region. Increasing the sampling of meteorological fields at different heights would also prove valuable.

It would be useful to present the more details about the AIRPARIF inventory in the text, e.g., how it was constructed, its spatial resolution, etc.

Comments on treatment of MHD and "background": P.7, line 6: MHD is described as a remote location. State here that this site was specifically evaluated as a potential background site. See also comments below. P.16, line 3-4: The conclusion that MHD is not a relevant site for background on the seasonal scale does not seem to be fully

supported by results. In some instances, a site that is classified as rural or peri-urban (or possibly urban) could represent background mole fractions under certain meteorological conditions. Selection of background can performed with using many methods, including meteorological filtering, analyzing tracer/tracer correlations, or evaluating the stability of observations. There is a significant body of literature detailing methods for selecting observations that represent background mole fractions (as an example, see Ruckstuhl et al., 2012, http://www.atmos-meas-tech.net/5/2613/2012/).

P.18, lines 5-7: The conclusion here that MHD is not a relevant background site for Paris or other Western European cities also does not seem to be fully supported by the evidence. The definition of background depends on the domain of interest and also the timescale. For example, a single site may not be relevant for selecting background observations at all times and under all conditions. It is not clear whether there are ever any meteorological conditions that support MHD as a relevant local and/or regional background site. In general, the conclusions regarding MHD could be further supported by the evidence from the back trajectory and fine wind sector analysis (Sections 3.1 and 3.5.2) and/or the Supplemental materials (Figures S1 and S2).

Technical Corrections The manuscript could further benefit from more labeling figures to classify sites as "Urban" and "Periurban/Rural". Regarding analytical methods, the paper would also benefit from stating early on that all 7 sites (new and previously published) are on the same CO2 calibration scale (WMO X2007), use similar analytical procedures, and have relatively small uncertainties. This could be stated perhaps in the introduction or at the beginning of the methods section. Introduction: Suggest presenting the site code QUA to associate this site with the ABLH measurements from the time they are first introduced. Figure 6: May help to include inlet heights. Also, maybe label plots as Urban, peri-urban, rural/remote, etc.

P.4, line14: The reference Schmidt et al., (2014) first appears here, however it was not included in the list of references at the end of the paper.

[Figure]

**[ACPD](https://example.com)**

Interactive
comment

P.7, line 23: The authors mention the cell temperature of the analyzer at the EIF site was modified to undergo cell temperature set point at 60°C, however do not discuss what impact (if any) this may have on the results. Details of such analytical differences could be useful for others in the community conducting studies using similar analyzers.

Please be clear when meteorological data is measured vs. modeled—e.g. add the word modeled to Figure 3 caption.

Figure 5: might be useful to add inlet heights to the site key

Figure 7: What is the difference between the violet and red traces? Please describe in the text.

Figure 12, the wind roses highlighting CO2 concentrations and indicating the origin of the air masses being measured, was particularly interesting. Unfortunately, the discussion of this figure includes a lot of discussion of background, but it isn't clear exactly how the authors determined the background. I would also like to see explicit explanation of how the seasonal adjustments to the CO2 concentrations were made.

Table 4: The use of "N" is confusing since this is a percentage, not an integer. Consider renaming "coverage"?

Page 10 Line 20: shouldn't this section be titled, "Results and Discussion?"

Page 12 Lines 28-32: What about the effect of inlet height? MON is much lower than TRN50.

Page 13 Line 6: Max interseasonal difference is higher than the mean annual afternoon dispersion: what does this imply?

Page 13 Line 10: "strong impact of regional CO2 emissions variability:" why? Please elaborate a bit more.

Page 14 Lines 5-34: Please put the seasons in the same order in the text and in the plot.

Page 18 Lines 21-22: Define local in terms of spatial scale.

---

## Author Comment (AC1) · 24 Jul 2017

Atmos. Chem. Phys. Discuss., doi:10.5194/acp-2016-218-RC1, 2016 © Author(s) 2016. CC-BY 3.0 License.

Answer to Referee 1 (Jocelyn Turnbull) by Irène Xueref-Remy et al

to "Interactive comment on "Diurnal, synoptic and seasonal variability of atmospheric CO2 in the Paris megacity area" by Irène Xueref-Remy et al. "

REF 1 : This paper describes a year-long series of in situ CO2 measurements from sites in and around Paris. The paper focuses on how and why the CO2 signals vary: the proximity to the city; height of the inlet above ground; variability in emission sources; wind direction and speed. They demonstrate that in many wind regimes, emissions

from upwind sources can contribute as much or more CO2 than local Paris emissions. They show that urban CO2 variability is complex, implying that a strong understanding of these factors and the particular sampling network is needed to infer the emission flux from such measurements. Of particular note is that the Eiffel Tower sampling site is challenging to interpret since the inlet height is only sometimes within the boundary layer. This is a very nice, detailed examination of urban CO2 source variability that will be useful for the existing and upcoming urban greenhouse gas researchers. This research area is still in its infancy, and this study gives a very good demonstration of how urban sampling networks should be designed and the types of problems that can be encountered. This paper is entirely appropriate for publication in ACP. I see no major issues with the paper, and recommend minor revisions for clarity and language usage.

Authors : The authors thank very much Jocelyn Turnbull for her careful reading and for her constructive feedbacks. We answer to each point that she mentioned hereafter. The first author apologizes for the time that it took us to send our reply, due to her particular situation as she recently left LSCE to move to another institute in the southeast of France.

REF 1 : Specific comments: The authors should edit the full paper for correct English grammar. I point out some specific words in further comments, but there are many other cases where the grammar is comprehensible but incorrect.

Authors : We thank Referee 1 for this comment and we will edit the full paper for correct English grammar.

REF 1 : Abstract page 1 line 31. "elevated" is used here and in other places through the paper to mean "sites where the inlet is well above ground level". This is confusing though, because "elevated" is also commonly used to mean "the CO2 is higher than background". Perhaps "two sites with inlets high above ground level"?

Authors : The correction will appear according to Ref.1's suggestion.

REF 1 : Introduction pg 3 line 3 (and several other times in the paper). "conurbation" is not commonly used in English – I am a native speaker and had to look up the meaning. Perhaps "metropolitan area" would be a better choice.

Authors : The correction will appear according to Ref.1's suggestion.

REF 1 : Pg 5 ln 3-12. Are there any large points sources in the metropolitan area? You mention some in the next section, but it would be helpful to first give them in this section.

Authors : Yes there are large point sources, mostly from the industrial sectors. According to Ref.1's suggestion, we will add some information about these sources in section 2.1.

REF 1 : Pg 6 lines 13-31. Are there any emissions directly from the buildings you are sampling on top of?

Authors : The sites were carefully chosen so that none of them emits in a way that could directly contaminate the sampling inlet. We will add this information.

REF 1 : Pg 7 line 16. You say that this station is ideally located, but don't give any justification as to why it is ideal.

Authors : We will change the sentence as follows : " This station allows monitoring the height of the urban atmospheric boundary layer (ABL) in the megacity."

REF 1 : Pg 8 ln 6-7. "Only the last calibration: " it is not clear what is meant by this sentence. Please clarify.

Authors : We will explain that "gas equilibrium issues implied retaining only the last calibration cycle for the calibration equation."

REF 1 : Pg 8 ln 16. Please give a reference for the ICOS procedure.

Authors : The following reference will be added : Hazan et al, 2016 (Hazan, L.,

Tarniewicz, J., Ramonet, M., Laurent, O., and Abbaris, A.: Automatic processing of atmospheric $CO_2$ and $CH_4$ mole fractions at the ICOS Atmosphere Thematic Centre, Atmos. Meas. Tech., 9, 4719-4736, doi:10.5194/amt-9-4719-2016, 2016).

REF 1 : Pg 8 ln 20. How were the very local influences (that were removed) identified?

Authors : Very local influences were identified from the very short duration of the events (a few seconds to minutes) and from the large standard deviation of the $CO_2$ averages associated to these events.

REF 1 : Pg 8 ln 26. Please reference the WMO-X2007 scale.

Authors : The WMO-X2007 scale will be referenced to Zhao, C. L. and P. P. Tans (2006), estimating uncertainty of the WMO mole fraction scale for carbon dioxide in air, Journal of Geophysical Research-Atmospheres, 111(D8), 10.1029/2005JD006003. We will also provide the following link : https://www.esrl.noaa.gov/gmd/ccl/co2_scale.html/

REF 1 : Pg 10 ln 4. Please provide a link or reference for the Met Eireann met data.

Authors : The following link will be added : http://www.met.ie/.

REF 1 : Pg 10 ln 23. What met dataset was used in HySplit?

Authors : The Met dataset used in HySplit is the NOAA-NCEP/NCAR reanalysis at a $2.5° \times 2.5°$ and 6 h resolution (http://rda.ucar.edu/datasets/ds090.0). This reference was already given in the Supplementary material but will be given in the main text.

REF 1 : Pg 11 ln 11. I don't see the 1-sigma std devs on the plot. Did you mean to refer to figure 6 here?

Authors : Yes this is right, we will correct this.

REF 1 : Pg 11 ln 15. Please provide references to previous work that has discussed the biosphere and vertical dilution impacts on $CO_2$.

Authors : The following reference (dedicated to the TRN site) will be added : Schmidt

et al, 2014 (Schmidt, M., Lopez, M., Yver Kwok, C., Messager, C., Ramonet, M., Wastine, B., Vuillemin, C., Truong, F., Gal, B., Parmentier, E., Cloué, O., and Ciais, P.: High-precision quasi-continuous atmospheric greenhouse gas measurements at Trainou tower (Orléans forest, France), Atmos. Meas. Tech., 7, 2283-2296, https://doi.org/10.5194/amt-7-2283-2014, 2014).

REF 1 : Pg 12 ln 2. "During daytime: " do you mean mid-afternoon?

Authors : Yes this is right, this will be corrected.

REF 1 : Pg 12 ln 3. "significant positive gradient". Perhaps "enhancement" would be a better word. (Also used elsewhere in the paper).

Authors : The correction will be made according to Ref.1's suggestion.

REF 1 : Pg 12 ln 12-14. Why does the lack of diurnal cycle at MHD make it a poor choice for background? If you are interested in examining the urban anthropogenic CO2 source, then this is probably correct, but if you are interested in the diurnal variability of the continental biosphere signal, then it might be a good choice. Please explain/clarify.

Authors : This study is dedicated to the Paris megacity region ($\sim$150 km of diameter). The activity of the biosphere and other fluxes located between MHD and the Paris region impacts the amplitude of the regional " background " CO2 diurnal cycle i.e. existing without the contribution of the Paris megacity fluxes. We do not want to take this remote contribution into account in our regional study. We propose to reformulate the text as follows: " 2/the MHD signal is several ppm below the continental signals, even at the rural site of TRN that was already shown not to be much influenced by the Paris megacity fluxes in Schmidt et al (2014). Thus, MHD does not reproduce the background diurnal variability of the regional stations and is clearly not a relevant background site for continental-Europe urban studies at the diurnal scale and at the regional scale ($\sim$150 km). "

REF 1 : Pg 12 ln 22-23. Can you give an estimate of the magnitude of the biospheric flux through the seasons. It would be helpful to know how large it might be relative to the fossil fuel flux (even though the biosphere flux might be poorly constrained).

Authors : In this section, we refered to the Bréon et al (2015) paper as it gives the magnitude of the biospheric fluxes from the C-TESSEL model and of the fossil fuel fluxes from AIRPARIF, through the seasons. From this reference and to answer to Ref.1's comment, we will add some quantitative information to our text, that will allow an easy assessment of the relative contribution of both types of fluxes at different periods of the year.

REF 1 : Pg 13 ln 14. I think you mean figure 5 and 6, not figure 7.

Authors : Yes this is right, the correction will be made.

REF 1 : Pg 15 ln 5-11. I don't see what this discussion of the vertical gradients adds to the paper. It could either be cut out, or a sentence added to explain why it is useful.

Authors : Vertical gradients are important to consider regarding mesoscale modeling studies. We propose to add the following sentence to this section: " Quantifying such vertical gradients is of interest since these gradients have to be correctly reproduced in urban mesoscale modeling frameworks for accurate $CO_2$ atmospheric inversion purposes. "

REF 1 : Pg 15 ln 13-26. The AIRPARIF inventory, I believe, is fossil fuel $CO_2$ flux only, whereas you measure total $CO_2$ (both fossil and bio). Could it be that the smaller weekday/weekend differences in your observation be due to the fact that biospheric fluxes are constant through weekdays and weekends? I.e. the difference between weekdays and weekends would be proportionally smaller in the total $CO_2$ observations than in the inventory, if there is a large (and constant) biosphere flux. Could this also explain why the GIF signal is more consistent between weekdays and weekends? I.e. perhaps the biosphere contribution is relatively more important at GIF than the urban sites?

Authors : We agree with Ref.1 that this explanation is plausible. Furthermore, the influence of wind and of the CO2 background signal likely contributes to the discrepancy observed from the emissions and the observations datasets. We will rewrite this section according to these considerations.

REF 1 : Pg 15 ln 28. Does this seasonal cycle include all or only some hours of the day?

Authors : This seasonal cycle includes all hours of the day.

REF 1 : Pg 16 ln 5-9. Please reference previous work that has discussed this phenomenon of seasonality in BL height, biosphere emissions and fossil fuel emissions. See for example: Denning, A. S., P. J. Rayner, R. M. Law and K. R. Gurney (1995). Atmospheric tracer transport model intercomparison project (TransCom). IGBP/GAIM report series report #4. D. Sahagian. Turnbull, J. C., P. J. Rayner, J. B. Miller, T. Naegler, P. Ciais and A. Cozic (2009). "On the use of 14CO2 as a tracer for fossil fuel CO2: quantifying uncertainties using an atmospheric transport model." Journal of Geophysical Research 114, D22302.

Authors : We will add both references to the text.

REF 1 : Pg 16 ln 17-21. Indeed, the CO2 signals are higher in the winter, but the standard deviations do not seem to be higher in winter. Elsewhere in the paper, the higher standard deviations are used to identify higher anthropogenic emissions. Please justify why this is not the case here.

Authors : What we meant is that a signal with a higher standard deviation can be associated to the influence of fresher anthropogenic emissions, i.e. that are not well mixed in the atmosphere. We will make sure that this is clear enough throughout the paper.

REF 1 : Pg 17 ln 5-10. I don't think you can conclude that fossil fuel emissions are lower in summer from this dataset, since photosynthetic drawdown confounds the signal so

strongly.

Authors : We agree that the first sentence of this section was confusing regarding the influence of the biospheric activity. We will reformulate as follows: "For all stations except GON, the annual minimum of concentration occurs in August and follows: 1/ the minimum of anthropogenic emissions as given by the Airparif inventory; and 2/ the maximum of the photosynthetic activity."

REF 1 : Pg 17 ln 21-24. Please explain and/or reference how the seasonal adjustment was performed. Reference previous work that discusses relationship between concentration and wind speed/ventilation.

Authors : The seasonal adjustment was done on the $CO_2$ hourly mean dataset of each station by : 1/ computing the annual mean of the dataset ; 2/ computing the monthly seasonal index for each month by dividing the monthly mean by the annual mean of the dataset ; 3/ interpolating the monthly seasonal index dataset to an hourly scale dataset ; and 4/ dividing the hourly dataset by the hourly seasonal index. We agree that several previous studies discussed the relationship between concentration and wind speed / ventilation and we will add their references in this section.

REF 1 : Pg 18 ln 20-22. Please clarify what the relationship is that justifies using the different wind speed regimes to identify local and remote emissions. Another sentence or two would help to follow the logic of doing this.

Authors : To make it simple, this depends on the strength of atmospheric mixing of local emissions against their ventilation and the advection of remote signals. For exemple, at windspeed less than 3 m.s-1 (11 km.h-1), it takes one hour or more for the airmass to pass over the center of Paris ($\sim$10 km of diameter) while at 8 m.s-1 ($\sim$29 km.h-1) it takes only about 20 minutes or less: the influence of local emissions vs. remotes ones on the passing air mass thus gets higher with decreasing windspeed. We will add a few sentences in this section to make it clearer.

[Figure]

REF 1 : Pg 18 ln 25-29. Please expand this explanation a little more and/or reference the method, particularly for the square root transformation that has been applied.

Authors : We used the function polarFreq of the Openair software for R (http://www.openair-project.org/PDF/OpenAir_Manual.pdf) with the option "weighted mean". We will rewrite these sentences to make it clearer.

REF 1 : Pg 20 ln 1-23. Exactly how close are MON and GON to CDG airport? Are there any other industrial or commercial facilities that could be causing this signal? In section 2.1.1. You stated that airport emissions are 4% of the total, whereas industrial emissions are 14%, so industrial emissions are potentially more important. Are CDG emissions large enough to plausibly explain the signal at both sites?

Authors : MON and GON are located about 13 km and 9.5 km away from the middle of the CDG airport. We agree with Ref.1 that we may have underestimated the influence of some industrial sites that are closer to the MON and GON sites than is the CDG airport. We will take this into account and will modify the text accordingly.

REF 1 : Pg 20 ln 22-23. How would carbon isotopes and specific emission tracers help to discriminate between airport and traffic emissions? Does jetfuel have a different isotopic signature than petrol/diesel?

Authors : This sentence is incorrect indeed and will be removed.

REF 1 : Pg 21 ln 3-11. See also previous comment – are the CDG emissions large enough at night and close enough to plausibly influence the GON site so strongly? It would be helpful to include Figure S2 in the main paper, since that shows the actual CO2 data which is the main focus of the paper. If there is a limitation on the number of figures, Figures 3 and 4 could move to the supplementary material (since the wind directions are also shown in figure S2).

Authors : The CDG airport is operational day and night. But as mentioned earlier, we will better consider the influence of industrial emissions relatively to the one of the

airplanes and CDG airport, and rewrite the text accordingly. We will include Figure S2 in the main paper. We think informative for the reading to keep Figure 3 in the main paper. We will move Figure 4 to the supplementary material.

REF 1 : Figure 5 is essentially repeated in figure 6. Could these two figures be combined?

Authors : The authors agree with Ref.1. To keep the information readable, we will combine both figures in one single figure with two panels a/ and b/.

REF 1 : Figure 9a and b could be combined by plotting 9a as an 8th panel in figure 9b.

Authors : The correction will be made according to Ref.1's suggestion.

REF 1 : Tables are mentioned in the text in a different order than the order of their numbering.

Authors : This will be corrected in the text.

---

## Author Comment (AC2) · 24 Jul 2017

Atmos. Chem. Phys. Discuss., doi:10.5194/acp-2016-218-RC1, 2016 © Author(s) 2016. CC-BY 3.0 License.

Answer to Referee 2 by Irène Xueref-Remy et al

to "Interactive comment on "Diurnal, synoptic and seasonal variability of atmospheric CO2 in the Paris megacity area" by Irène Xueref-Remy et al."

General Comments

REF.2 : This paper analyzes nearly 1 year of CO2 data from the Paris megacity greenhouse gas measurement network. The analysis focuses on deciphering the CO2 ob-

servations on diurnal and seasonal time scales, and includes a careful examination of the influence of the atmospheric boundary layer height (ABLH), wind speed and direction, and local anthropogenic emissions on these signals. The measurement network contains six total sites across Ile de France spanning a range of conditions from rural to the Eiffel Tower in the heart of Paris. The report presents measurements that provide an important baseline for emissions from Paris and for comparison to other global megacities.

Authors : We thank Referee 2 very much for her/his careful reading of our paper and for her/his constructive comments. We answer to each point hereafter. The first author apologizes for the time that it took us to send our reply, due to her particular situation as she recently left LSCE to move to another institute in the south-east of France.

Specific Comments

REF.2 : The authors present a detailed analysis of the CO2 observations based on time, location, and wind speed/direction to infer the seasonal influence of local and background contributions at each site. This analysis is largely qualitative, but could be made far more quantitative and definitive if based around back trajectory analyses, such as those shown in Figure S1. We strongly suggest that the discussion of Section 3.1 be expanded and used to validate the conclusions of Section 3.5 which appear to be based on site wind measurements.

Authors : We will expand a bit more the discussion of Section 3.1 and will attempt to use this discussion to consolidate the conclusions of Section 3.5 as far as possible, but we think that the backtrajectories of Figure S1 deliver a qualitative information rather than a quantitative one. Indeed, we produced these backtrajectories using a public tool (HYSPLIT) with a 2.5° x 2.5° wind resolution, and this resolution is much too low to decipher differences between the Paris sites, that are distant by a few dozens of kilometers only. Furthermore, this low resolution can only give a gross estimate on the synoptic air mass fluxes between MHD or the Ruhr/Benelux area and the Paris

megacity region. A quantitative analysis of the wind trajectories would require a dedicated model with a much finer resolution. This would require consequent work in terms of development, time calculation and analysis, and we therefore think that it would represent another study in itself, that is out of the scope of this paper.

REF.2 : The study concludes that the level of $CO_2$ enhancement varies with urbanization level local to the site; however, the paper does not directly discuss estimates of enhancement relative to background (or the concept of background) until much later in the paper. While diurnal and seasonal variability and the gradients between sites are the primary focus of this paper, background estimation is an important topic and which merits more introduction. Overall, there are two key points that should be incorporated: (1) the concept of background should be defined relative to the domain of interest and (2) a single site may not represent background $CO_2$ mole fractions under all meteorological conditions.

Authors : We define ''background'' in the introduction as the $CO_2$ mole fraction without the contribution of the regional emissions (p. 3, l. 18). By regional, we mean the Paris megacity region i.e. a radius of about 100 km around the Paris center. We will make this spatial scale clearer to address point (1) mentioned above. Regarding point (2), we fully agree that a single site may not represent background $CO_2$ mole fractions under all meteorological conditions, as illustrated with our study of MHD in this paper, and in two previous papers of this team [Bréon, F. M., Broquet, G., Puygrenier, V., Chevallier, F., Xueref-Remy, I., Ramonet, M., Dieudonné, E., Lopez, M., Schmidt, M., Perrussel, O., and Ciais, P.: An attempt at estimating Paris area $CO_2$ emissions from atmospheric concentration measurements, Atmos. Chem. Phys., 15, 1707-1724, https://doi.org/10.5194/acp-15-1707-2015, 2015 ; Staufer, J., Broquet, G., Bréon, F.-M., Puygrenier, V., Chevallier, F., Xueref-Rémy, I., Dieudonné, E., Lopez, M., Schmidt, M., Ramonet, M., Perrussel, O., Lac, C., Wu, L., and Ciais, P.: The first 1-year-long estimate of the Paris region fossil fuel $CO_2$ emissions based on atmospheric inversion, Atmos. Chem. Phys., 16, 14703-14726, https://doi.org/10.5194/acp-16-14703-2016,

2016].

REF.2 : Additionally, the paper should use CO2 enhancement values relative to some chosen background rather than absolute CO2 values (eg 410 ppm) since the global background will surpass even these "elevated" values in the near future.

Authors : As a consequence of the previous point, the term ''background'' remains more a concept than a quantity in our study and is not given as a numerical quantity value. We therefore do not report enhancements here (not even with a fixed value like 410 ppm). We reported dynamic enhancements in the above-mentioned studies of Bréon et al. (2015, Fig. 7) and Staufer et al. (2016, Fig. 4) but under specific meteorological conditions. We are trying to get a more general assessment in an on-going study, but this one is at early stage and clearly distinct from the research that we are reporting here.

REF.2 : The challenges of analyzing these measurements raises several priority questions regarding the Paris network. We note that the INFLUX network in Indianapolis, IN USA contains 13 towers for a smaller, less populated urban area and approximately 1/10th the emissions of Paris/IdF [Turnbull, Jocelyn C., et al. "Toward quantification and source sector identification of fossil fuel CO2 emissions from an urban area: Results from the INFLUX experiment." Journal of Geophysical Research: Atmospheres 120.1 (2015): 292-312]. We would have expected some discussion of the density of the Paris network, the potential benefit of additional sites, and where they would ideally be located for maximum impact. This is particularly relevant for the "background" discussion since it is clear that Mace Head alone is insufficient for this analysis and that a full understanding of Paris CO2 monitoring may well require observations from as far away as the Ruhr or the Benelux region.

Authors: Through the CO2-Megaparis project, we were funded for 3 new sites on top of 2 existing national ICOS sites. We chose to deploy these new sites on the axis of the dominant winds (NE/SW) in order to optimize the amount of available data. The

extension of the network for inverse modeling purposes is discussed by Staufer et al. (2016, Section 4.3) who conclude to the need of 8 more sites in the suburban/rural border of the city. Longer prospects are the topic of Wu et al. (2016) [Wu, L., Broquet, G., Ciais, P., Bellassen, V., Vogel, F., Chevallier, F., Xueref-Remy, I., and Wang, Y.: What would dense atmospheric observation networks bring to the quantification of city $CO_2$ emissions?, Atmos. Chem. Phys., 16, 7743-7771, https://doi.org/10.5194/acp-16-7743-2016, 2016.]. Furthermore, in order to improve our understanding and modeling of the vertical transport of urban $CO_2$ emissions, we mentioned in our conclusion the need to develop more measurements in the center of Paris, especially $CO_2$ vertical profiles on the Eiffel tower (p.24 lines 13-15). We will extend the conclusion to synthesize all those elements.

REF.2 : Given the topographical similarities of Paris and Indianapolis, we were also surprised that more discussion was not presented comparing the $CO_2$ concentration "plume" patterns from these urban areas.

Authors : If we compare Figure 2d of Turnbull et al. (2015) and Figure 7 of Bréon et al. (2015), we see enhancements of a few ppm in both cases. We can report this information, but the background is defined differently in each paper and the comparison remains rather qualitative and artificial. Also remember that we have much less sites in Paris than in Indianapolis.

REF.2 : Newman et al. [Newman, S., et al. "Diurnal tracking of anthropogenic CO 2 emissions in the Los Angeles basin megacity during spring 2010." Atmospheric Chemistry and Physics 13.8 (2013): 4359-4372] showed diurnal patterns for $CO_2$ from the Los Angeles megacity, but there was no comparison made with these data. This is particularly relevant since Los Angeles $CO_2$ emissions are well known to be dominated by vehicle/transportation and impart significant rush hour maxima (0700-1000 and 1500-1900) that are absent from all but the EIF signals in Paris. The arguments for winter vehicle emissions in Paris are not obvious from the figures as presented.

Authors : We agree and we will add some words in the paper dedicated to the comparison with the Newman et al (2013) study. In Paris, according to the AIRPARIF inventory for the year 2010, the $CO_2$ emissions are dominated by the heating sectors (43%, summing home and commercial emissions) followed by the traffic sector (29%). In winter, the heating sector is clearly dominant as the air temperatures are low (see Bréon et al, 2015, Figure 3). This may explain the differences observed between LA and Paris, although the vicinity of each station to traffic and heating sources must also be taken into account.

REF.2 : The Eiffel Tower (EIF) site offers unique observations that might be more fully exploited in future studies. Complete diurnal and day of week sampling at this site would enable greater understanding of variability across the network. Adding vertical profile measurements at eg 50, 100, and 200 m to complement the 300 m inlet height would add tremendously to understanding the ABLH/CO2 linkages as well as providing different spatial sensitivity footprints within the Paris/IdF region. Increasing the sampling of meteorological fields at different heights would also prove valuable.

Authors : We thank REF.2 for this comment and we will mention more explicitly in our text that we plan to carry out such measurements within the Paris 2030 ''Le CO2 parisien" project that was funded by Ville de Paris.

REF.2 : It would be useful to present the more details about the AIRPARIF inventory in the text, e.g., how it was constructed, its spatial resolution, etc.

Authors : The AIRPARIF inventory is well detailed in the Bréon et al (2015) paper. We will pay attention to better refer to this paper and will also add some key information about this inventory in our paper.

REF.2 : Comments on treatment of MHD and "background": P.7, line 6: MHD is described as a remote location. State here that this site was specifically evaluated as a potential background site.

Authors : We will do so.

REF.2 : See also comments below. P.16, line 3-4: The conclusion that MHD is not a relevant site for background on the seasonal scale does not seem to be fully supported by results. In some instances, a site that is classified as rural or peri-urban (or possibly urban) could represent background mole fractions under certain meteorological conditions. Selection of background can performed with using many methods, including meteorological filtering, analyzing tracer/tracer correlations, or evaluating the stability of observations. There is a significant body of literature detailing methods for selecting observations that represent background mole fractions (as an example, see Ruckstuhl et al., 2012, http://www.atmos-meas-tech.net/5/2613/2012/).

Authors : We defined our background as the $CO_2$ mole fraction without the contribution of remote emissions. By remote, we mean out of the Paris megacity region (i.e. $\sim$100 km around the center of Paris). Our observations show clear differences of several ppm between MHD and the rural site of TRN for example, which has been already demonstrated to be poorly influenced by the Paris megacity emissions. This shows that MHD is not a relevant background for the Paris megacity region. Regarding background calculation, we are aware of the complexity of the question and of the different methods available, but as we explained above this question is out of the scope of this paper. We will rewrite the text to make these points clearer.

REF.2 : P.18, lines 5-7: The conclusion here that MHD is not a relevant background site for Paris or other Western European cities also does not seem to be fully supported by the evidence. The definition of background depends on the domain of interest and also the timescale. For example, a single site may not be relevant for selecting background observations at all times and under all conditions. It is not clear whether there are ever any meteorological conditions that support MHD as a relevant local and/or regional background site. In general, the conclusions regarding MHD could be further supported by the evidence from the back trajectory and fine wind sector analysis (Sections 3.1 and 3.5.2) and/or the Supplemental materials (Figures S1 and S2).

Authors : We explained above why MHD is not a relevant background site for the Paris megacity region or other continental Western cities. We will make this clearer in the paper and make our best to rely also on the backtrajectories to consolidate our argumentation.

Technical Corrections

REF.2 : The manuscript could further benefit from more labeling figures to classify sites as "Urban" and "Periurban/Rural".

Authors : We will do so.

REF.2 : Regarding analytical methods, the paper would also benefit from stating early on that all 7 sites (new and previously published) are on the same $CO_2$ calibration scale (WMO X2007), use similar analytical procedures, and have relatively small uncertainties. This could be stated perhaps in the introduction or at the beginning of the methods section.

Authors : We will follow REF.2's suggestion and will add a sentence about this earlier in the paper.

REF.2 : Introduction: Suggest presenting the site code QUA to associate this site with the ABLH measurements from the time they are first introduced.

Authors : We will do so.

REF.2 : Figure 6: May help to include inlet heights. Also, maybe label plots as Urban, peri-urban, rural/remote, etc.

Authors : We will do so.

REF.2 : P.4, line14: The reference Schmidt et al., (2014) first appears here, however it was not included in the list of references at the end of the paper.

Authors : This will be corrected.

REF.2 : P.7, line 23: The authors mention the cell temperature of the analyzer at the EIF site was modified to undergo cell temperature set point at 60_C, however do not discuss what impact (if any) this may have on the results. Details of such analytical differences could be useful for others in the community conducting studies using similar analyzers.

Authors : No specific impact of the set point of 60°C was observed in the results.

REF.2 : Please be clear when meteorological data is measured vs. modeledâËŸA ËĞTe.g. add the word modeled to Figure 3 caption.

Authors : We will make this clearer.

REF.2 : Figure 5: might be useful to add inlet heights to the site key

Authors : We will do so.

REF.2 : Figure 7: What is the difference between the violet and red traces? Please describe in the text.

Authors : The violet trace uses only CO2 hourly data that are concomitant to ABLh hourly data. The red trace uses all CO2 hourly data points available for the relevant season. We will explain this in the text.

REF.2 : Figure 12, the wind roses highlighting CO2 concentrations and indicating the origin of the air masses being measured, was particularly interesting. Unfortunately, the discussion of this figure includes a lot of discussion of background, but it isn't clear exactly how the authors determined the background. I would also like to see explicit explanation of how the seasonal adjustments to the CO2 concentrations were made.

Authors : As we explained here before, the term "background" is not quantitative in this paper, but is a concept and represents the contribution of remote fluxes (i.e. not from the Paris megacity area). We will make this clearer in our paper. The seasonal adjustment was done on the CO2 hourly mean dataset of each station by : 1/ computing the

annual mean of the dataset ; 2/ computing the monthly seasonal index for each month by dividing the monthly mean by the annual mean of the dataset ; 3/ interpolating the monthly seasonal index dataset to an hourly scale dataset ; and 4/ dividing the hourly dataset by the hourly seasonal index. REF.2 : Table 4: The use of "N" is confusing since this is a percentage, not an integer. Consider renaming "coverage"?

Authors : We will modify according to this suggestion.

REF.2 : Page 10 Line 20: shouldn't this section be titled, "Results and Discussion?"

Authors : Yes, this is right, we will modify according to this suggestion.

REF.2 : Page 12 Lines 28-32: What about the effect of inlet height? MON is much lower than TRN50.

Authors : We recognize that this point merits more consideration and will discuss it in the text.

REF.2 : Page 13 Line 6: Max interseasonal difference is higher than the mean annual afternoon dispersion: what does this imply?

Authors : This implies that the seasonal variability is higher than the mean dispersion of the fluxes.

REF.2 : Page 13 Line 10: "strong impact of regional $CO_2$ emissions variability:" why? Please elaborate a bit more.

Authors : We will elaborate this a bit more.

REF.2 : Page 14 Lines 5-34: Please put the seasons in the same order in the text and in the plot.

Authors : We will do so.

REF.2 : Page 18 Lines 21-22: Define local in terms of spatial scale.

Authors : Local is here define as "less than 10 km". We will make this clear in the

paper.

---

## Author Response (AR1)

*REF 1 : This paper describes a year-long series of in situ $CO_2$ measurements from sites in and around Paris. The paper focuses on how and why the $CO_2$ signals vary: the proximity to the city; height of the inlet above ground;*
15    *variability in emission sources; wind direction and speed. They demonstrate that in many wind regimes, emissions from upwind sources can contribute as much or more $CO_2$ than local Paris emissions. They show that urban $CO_2$ variability is complex, implying that a strong understanding of these factors and the particular sampling network is needed to infer the emission flux from such measurements. Of particular note is that the Eiffel Tower sampling site is challenging to interpret since the inlet height is only sometimes within the boundary*
20    *layer. This is a very nice, detailed examination of urban $CO_2$ source variability that will be useful for the existing and upcoming urban greenhouse gas researchers. This research area is still in its infancy, and this study gives a very good demonstration of how urban sampling networks should be designed and the types of problems that can be encountered. This paper is entirely appropriate for publication in ACP. I see no major issues with the paper, and recommend minor revisions for clarity and language usage.*

Authors : The authors thank very much Jocelyn Turnbull for her careful reading and for her constructive feedbacks. We answer to each point that she mentioned hereafter. The first author apologizes once again for the time that it took us to send our reply, due to her particular situation as she recently left LSCE to move to
30    another institute in the south-east of France.

*REF 1 : Specific comments:*
*The authors should edit the full paper for correct English grammar. I point out some specific words in further*
35    *comments, but there are many other cases where the grammar is comprehensible but incorrect.*

Authors : We thank Referee 1 for this comment and we edited the full paper for correct English grammar.

*REF 1 : Abstract page 1 line 31. "elevated" is used here and in other places through the paper to mean "sites where the inlet is well above ground level". This is confusing though, because "elevated" is also commonly used to mean "the $CO_2$ is higher than background".*
*Perhaps "two sites with inlets high above ground level"?*

Authors : The correction was made according to Ref.1's suggestion.

*REF 1 : Introduction pg 3 line 3 (and several other times in the paper). "conurbation" is not commonly used in English – I am a native speaker and had to look up the meaning. Perhaps "metropolitan area" would be a better*
10  *choice.*

Authors : The correction was made according to Ref.1's suggestion.

15  *REF 1 : Pg 5 ln 3-12. Are there any large points sources in the metropolitan area? You mention some in the next section, but it would be helpful to first give them in this section.*

Authors : Yes there are large point sources from the industrial sector. According to Ref.1's suggestion, in section 2.1 we added some information about the industrial sources located in the vicinity of each station of the Paris
20  network.  The source of this information is a national database (http://www.georisques.gouv.fr), which provides the location of the main industrial sites of the Paris region by county, and the greenhouse gases emissions estimated for each of these industrial sites.

25  *REF 1 : Pg 6 lines 13-31. Are there any emissions directly from the buildings you are sampling on top of?*

Authors : The sites were carefully chosen so that none of them emits in a way that could directly contaminate the sampling inlet. We added this piece of information.

*REF 1 : Pg 7 line 16. You say that this station is ideally located, but don't give any justification as to why it is ideal.*

Authors : We changed the sentence as follows : " This station allows monitoring the height of the urban
35  atmospheric boundary layer (ABL) above the Paris megacity."

*REF 1 : Pg 8 ln 6-7. "Only the last calibration: " it is not clear what is meant by this sentence. Please clarify.*

40  Authors : We modified this sentence to make it clearer. The new sentence is the following : "Gas equilibrium issues implied retaining only the last calibration cycle of the 4 cycles at MON and GON (and of the 2 cycles at EIF) to compute the calibration equation".

*REF 1 : Pg 8 ln 16. Please give a reference for the ICOS procedure.*

Authors : The following reference was added : Hazan et al, 2016 (Hazan, L., Tarniewicz, J., Ramonet, M., Laurent, O., and Abbaris, A.: Automatic processing of atmospheric $CO_2$ and $CH_4$ mole fractions at the ICOS Atmosphere Thematic Centre, Atmos. Meas. Tech., 9, 4719-4736, doi:10.5194/amt-9-4719-2016, 2016).

*REF 1 : Pg 8 ln 20. How were the very local influences (that were removed) identified?*

Authors : The following sentence was added : "Very local influences were identified from the short duration of the events (a few seconds to minutes) and from the large standard deviation of the $CO_2$ averages associated to these events."

*REF 1 : Pg 8 ln 26. Please reference the WMO-X2007 scale.*

Authors : The WMO-X2007 scale is now referenced to Zhao, C. L. and P. P. Tans (2006), estimating uncertainty of the WMO mole fraction scale for carbon dioxide in air, Journal of Geophysical Research-Atmospheres, 111(D8), 10.1029/2005JD006003. We also provided the following link : https://www.esrl.noaa.gov/gmd/ccl/co2_scale.html/

*REF 1 : Pg 10 ln 4. Please provide a link or reference for the Met Eireann met data.*

Authors : The following link was added : http://www.met.ie/.

*REF 1 : Pg 10 ln 23. What met dataset was used in HySplit?*

Authors : The Met dataset used in HySplit is the NOAA-NCEP/NCAR reanalysis at a 2.5° x 2.5° and 6 h resolution (http://rda.ucar.edu/datasets/ds090.0). This reference was given in the Supplementary material but is now part of the main paper :
"In order to get information about the origin of the air masses that reached our stations, back trajectories from the HYSPLIT model (Hybrid Single Particle Lagrangian Integrated Trajectory: http://www.arl.noaa.gov/HYSPLIT_info.php) model were calculated for the Paris city over the full period of study. We used wind fields from the NOAA-NCEP/NCAR reanalysis data archives, at a 2.5° x 2.5° and 6 h resolution (http://rda.ucar.edu/datasets/ds090.0/). The back trajectories were run for 72 h backwards and started at 10 m AGL. They were then aggregated on monthly plots that are shown in the supplementary material (Fig.S1)."

*REF 1 : Pg 11 ln 11. I don't see the 1-sigma std devs on the plot. Did you mean to refer to figure 6 here?*

Authors : Yes indeed, thank you. This was corrected.

*REF 1 : Pg 11 ln 15. Please provide references to previous work that has discussed the biosphere and vertical dilution impacts on $CO_2$.*

10  Authors : The following reference (dedicated to the TRN site) was added : Schmidt et al, 2014 **(**Schmidt, M., Lopez, M., Yver Kwok, C., Messager, C., Ramonet, M., Wastine, B., Vuillemin, C., Truong, F., Gal, B., Parmentier, E., Cloué, O., and Ciais, P.: High-precision quasi-continuous atmospheric greenhouse gas measurements at Trainou tower (Orléans forest, France), Atmos. Meas. Tech., 7, 2283-2296, https://doi.org/10.5194/amt-7-2283-2014, 2014).

*REF 1 : Pg 12 ln 2. "During daytime: " do you mean mid-afternoon?*

Authors : Yes this is right.  We modified the sentence accordingly.

*REF 1 : Pg 12 ln 3. "significant positive gradient". Perhaps "enhancement" would be a better word. (Also used elsewhere in the paper).*

Authors : Corrections were made according to Ref.1's suggestion through the whole paper using the terms
25  "enhancement" and "concentration difference" instead of "significant positive gradient".

*REF 1 : Pg 12 ln 12-14. Why does the lack of diurnal cycle at MHD make it a poor choice for background? If you are interested in examining the urban anthropogenic $CO_2$ source, then this is probably correct, but if you are*
30  *interested in the diurnal variability of the continental biosphere signal, then it might be a good choice. Please explain/clarify.*

Authors : This study is dedicated to the Paris megacity region (~200 km of diameter). The activity of the biosphere and other fluxes occuring between MHD and the Paris region impacts the amplitude of the regional
35  " background " $CO_2$ diurnal cycle i.e. existing without the contribution of the Paris megacity fluxes. Our study addresses the Paris regional scale (~100 km) and not the continental scale, therefore, we need a background that integrates the fluxes between MHD and IdF. We propose to reformulate the text as follows: "2/ the MHD signal is several ppm below the continental signals, even at the rural site of TRN that has already been shown not be significantly influenced by the Paris megacity fluxes (Schmidt et al, 2014). Thus, MHD does not reproduce
40  the background diurnal variability observed in the rural stations of IdF, and is clearly not a relevant background site for continental European urban studies at the diurnal scale and at the regional scale of ~100 km."

*REF 1 : Pg 12 ln 22-23. Can you give an estimate of the magnitude of the biospheric flux through the seasons. It would be helpful to know how large it might be relative to the fossil fuel flux (even though the biosphere flux might be poorly constrained).*

Authors : In this section of the paper, we refered to the Bréon et al (2015) paper as it gives the magnitude of the biospheric fluxes from the C-TESSEL model and of the fossil fuel fluxes from AIRPARIF, through the seasons. From this reference and to answer to Ref.1's comment, we added some quantitative information for allowing an easy assessment of the relative contribution of both types of fluxes at different periods of the year.
The following sentences were added :
"The biospheric fluxes show large diurnal and seasonal cycles, as mentioned in Bréon et al (2015) who reported net ecosystem exchange (NEE) outputs from the C-TESSEL model for the Paris region : NEE values are the highest in spring (-10 to -25 kt.hr$^{-1}$ during daytime and + 5 kt.hr$^{-1}$ during nighttime, and a daily mean of -5/-10 kt.yr$^{-1}$ which is the same order of magnitude as fossil fuel emissions i.e. 7 to 9 kt.hr$^{-1}$ in spring), a bit lower in summer and autumn  and much smaller in winter (-3 kt.hr$^{-1}$ during daytime and +2 kt.hr$^{-1}$ during nighttime, and a daily mean of -1 kt.hr$^{-1}$, which is much smaller than fossil fuel emissions that reach 10 kt.hr$^{-1}$ in winter)."

*REF 1 : Pg 13 ln 14. I think you mean figure 5 and 6, not figure 7.*

Authors : Yes this is right. The correction was made accordingly.

*REF 1 : Pg 15 ln 5-11. I don't see what this discussion of the vertical gradients adds to the paper. It could either be cut out, or a sentence added to explain why it is useful.*

Authors : Vertical gradients are important to consider regarding direct and inverse atmospheric $CO_2$ mesoscale modeling studies, that rely on the observations provided by regional networks such as the Paris one. Therefore, we think that this discussion of the vertical gradients is of interest for the community.
To make it clearer, we added the following sentence to this section: " Quantifying such vertical gradients is of interest since they have to be correctly reproduced in urban mesoscale modeling frameworks for accurate $CO_2$ atmospheric inversion purposes. "

*REF 1 : Pg 15 ln 13-26. The AIRPARIF inventory, I believe, is fossil fuel $CO_2$ flux only, whereas you measure total $CO_2$ (both fossil and bio). Could it be that the smaller week-day/weekend differences in your observation be due to the fact that biospheric fluxes are constant through weekdays and weekends? I.e. the difference between weekdays and weekends would be proportionally smaller in the total $CO_2$ observations than in the inventory, if there is a large (and constant) biosphere flux. Could this also explain why the GIF signal is more consistent between weekdays and weekends? I.e. perhaps the biosphere contribution is relatively more important at GIF than the urban sites?*

Authors : We agree with Ref.1 that this hypothesis was worth to think about. Indeed, we think that the smaller week-day/weekend differences in our observation could be either due to an overestimation of the inventory,

and/or due to the fact that anthropogenic emissions superimpose on biospheric fluxes (that are effectively constant through weekdays and weekends) and on the background signal, all being modulated by wind speed and direction conditions. These components likely soften the difference of anthropogenic $CO_2$ between weekdays and weekends in the atmosphere compared to the inventory. Regarding the fact that the diurnal cycle changes less through weekdays and weekends in GIF than if other stations, we agree with Ref.1 that this is likely due to the higher influence of biospheric fluxes at this site than at the others.

We added the following sentences to this section:
… "however, biospheric fluxes (eg Schmidt et al, 2015), wind speed and direction (see section 3.5) and $CO_2$ background signals (see section 3.1, and Turnbull et al, 2015) are also factors that modulate the observed $CO_2$ concentration at each site. Disentangling the role of each of these factors on the differences between the observed weekdays-to-weekend $CO_2$ concentration ratios versus the ones calculated from the inventory would require a dedicated analysis that is outside the scope of this paper."
… ", possibly because of a larger influence of the biospheric fluxes (that do not depend on weekday or weekends) at these stations compared to the contribution of anthropogenic emissions (that are different on weekdays and weekends according to AIRPARIF, see Fig. 4 in Bréon et al, 2015) and that are the strongest observed at GON (sections 3.2.1 and 3.5.2)."

*REF 1 : Pg 15 ln 28. Does this seasonal cycle include all or only some hours of the day?*

Authors : This seasonal cycle includes all hours of the day. This information was added in the paper.

*REF 1 : Pg 16 ln 5-9. Please reference previous work that has discussed this phenomenon of seasonality in BL height, biosphere emissions and fossil fuel emissions. See for example: Denning, A. S., P. J. Rayner, R. M. Law and K. R. Gurney (1995). Atmospheric tracer transport model intercomparison project (TransCom). IGBP/GAIM report series report #4. D. Sahagian. Turnbull, J. C., P. J. Rayner, J. B. Miller, T. Naegler, P. Ciais and A. Cozic (2009). "On the use of 14CO₂ as a tracer for fossil fuel $CO_2$: quantifying uncertainties using an atmospheric transport model." Journal of Geophysical Research 114, D22302.*

Authors : Both references were added to the text.

*REF 1 : Pg 16 ln 17-21. Indeed, the $CO_2$ signals are higher in the winter, but the standard deviations do not seem to be higher in winter. Elsewhere in the paper, the higher standard deviations are used to identify higher anthropogenic emissions. Please justify why this is not the case here.*

Authors : What we meant is that a signal with a higher standard deviation can be associated to the influence of fresher anthropogenic emissions, i.e. that are not well mixed in the atmosphere. This was made clearer throughout the paper.

*REF 1 : Pg 17 ln 5-10. I don't think you can conclude that fossil fuel emissions are lower in summer from this dataset, since photosynthetic drawdown confounds the signal so strongly.*

5    Authors : We agree that the first sentence of this section was confusing regarding the influence of the biospheric activity.
We reformulated as follows: "For all stations except GON, the annual minimum of concentration is observed in August when the following occurs : 1/ the minimum of anthropogenic emissions as given by the AIRPARIF inventory (see Fig.3 in Bréon et al, 2015); 2/ the maximum of photosynthetic activity (see Fig. 4 in Bréon et al);
10    and 3/ the maximum development of the ABLH (Fig. 8a)."

*REF 1 : Pg 17 ln 21-24. Please explain and/or reference how the seasonal adjustment was performed. Reference previous work that discusses relationship between concentration and wind speed/ventilation.*

15    Authors : The following sentences were added : "The $CO_2$ concentrations have been seasonally adjusted to avoid biases due to seasonal variability (section 3.4), by applying the following treatment to the $CO_2$ hourly dataset of each station : 1/ computing the annual mean of the dataset ; 2/ computing the monthly seasonal index for each month by calculating the ratio between the monthly mean and the annual mean of the dataset ; 3/ interpolating the monthly seasonal indexes at an hourly scale over the full period of study ; and 4/ dividing
20    the $CO_2$ hourly dataset by the hourly seasonal index. "
We agree that several previous studies discussed the relationship between concentration and wind speed / ventilation. We completed the first sentence of the section with some of these references, that were already cited elsewhere in the paper : Idso et al., 2002; Moriwaki et al., 2006, Rice et al, 2011 ; Garcia et al, 2012 ; Lac et al, 2013 ; Turnbull et al, 2015. .

*REF 1 : Pg 18 ln 20-22. Please clarify what the relationship is that justifies using the different wind speed regimes to identify local and remote emissions. Another sentence or two would help to follow the logic of doing this.*

Authors : To make it simple, this depends on the strength of atmospheric mixing of local emissions against their
30    ventilation and the advection of remote signals. To make it clearer, the following sentences were added : "This relies on considering the time given for atmospheric mixing of local and regional emissions (dominant at low to mid windspeeds) versus their ventilation (dominant at high windspeeds) : the integration of local and regional emissions into an air mass, which carries the signature of remote emissions when it is upwind of Paris, gets higher with decreasing windspeeds. For example, for windspeeds lower than 3 m.s$^{-1}$ (11 km.h$^{-1}$), it takes one
35    hour or more for any airmass to flow over the center of Paris (~10 km of diameter), allowing some time for local emissions to get mixed into the airmass, while at 8 m.s$^{-1}$ or more (~29 km.h$^{-1}$) it takes about 20 minutes or less, allowing less time for the atmospheric integration of local to regional emissions. In the middle range of windspeed (3-8 m s$^{-1}$), we expect most of the $CO_2$ variability to be driven by the influence of the regional emissions coming from Paris."

*REF 1 : Pg 18 ln 25-29. Please expand this explanation a little more and/or reference the method, particularly for the square root transformation that has been applied.*

Authors : We used the function polarFreq of the Openair software for R ([http://www.openair-project.org/PDF/OpenAir_Manual.pdf](http://www.openair-project.org/PDF/OpenAir_Manual.pdf)) with the option "weighted mean". This information was added to the paper.

*REF 1 : Pg 20 ln 1-23. Exactly how close are MON and GON to CDG airport? Are there any other industrial or commercial facilities that could be causing this signal? In section 2.1.1. You stated that airport emissions are 4% of the total, whereas industrial emissions are 14%, so industrial emissions are potentially more important. Are CDG emissions large enough to plausibly explain the signal at both sites?*

Authors : MON and GON are located about 13 km and 9.5 km away from the middle of the CDG airport. We agree with Ref.1 that we may have underestimated the influence of industrial sites that are closer to the MON and GON sites than is the CDG airport, and about which we added information in section 2.1.2 according to Ref.1's suggestion. We modified section 3.5.2 accordingly to discuss the possible influence of these sites on the

15 $CO_2$ concentration measured at MON and GON, referring to the information on the industrial sites provided in section 2.1.2.

*REF 1 : Pg 20 ln 22-23. How would carbon isotopes and specific emission tracers help to discriminate between airport and traffic emissions? Does jetfuel have a different isotopic*

20 *signature than petrol/diesel?*

Authors : This sentence is incorrect indeed and was removed, thank you.

*REF 1 : Pg 21 ln 3-11. See also previous comment – are the CDG emissions large enough at night and close*

25 *enough to plausibly influence the GON site so strongly? It would be helpful to include Figure S2 in the main paper, since that shows the actual $CO_2$ data which is the main focus of the paper. If there is a limitation on the number of figures, Figures 3 and 4 could move to the supplementary material (since the wind directions are also shown in figure S2).*

30 Authors : The CDG airport is operational day and night. But as mentioned earlier, we better considered the influence of industrial emissions relatively to the one of the airplanes and CDG airport, and rewrited the text accordingly. We included Figure S2 in the main paper. We think that Fig.3 provides to the reader an overview of the seasonal wind patterns in IdF, while Fig.S2 rather illustrate the day-to-day variability of wind speed and direction together with the variability of the $CO_2$ concentration observed at each site. Therefore we kept Fig.3 in

35 the main paper. We agree that Fig.4 is not essential in the main paper and we moved it to the supplementary material.

*REF 1 : Figure 5 is essentially repeated in figure 6. Could these two figures be combined?*

40 Authors : The authors agree with Ref.1 that there is some redundancy of information on these two figures. Therefore, after some tests, Figure 5 was moved to the Supplementary material eventually.

*REF 1 : Figure 9a and b could be combined by plotting 9a as an 8th panel in figure 9b.*

Authors : We firstly thought about following Ref.1's suggestion. We gave a try, but this is not satisfying as Fig.9a would be too much shrunk. Therefore we decided not to modify the layout of these two figures, to keep them readable.

*REF 1 : Tables are mentioned in the text in a different order than the order of their numbering.*

Authors : The order of Table 3 and Table 4 was reversed.

Thank you very much again and best regards,

Irène Xueref-Remy
Sanary-sur-Mer, Dec.8th, 2017

Atmos. Chem. Phys. Discuss.,
doi:10.5194/acp-2016-218-RC1, 2016

**Answer to Referee 2**
**by Irène Xueref-Remy et al**

10 **to "Interactive comment on "Diurnal, synoptic and seasonal variability of atmospheric CO$_2$ in the Paris megacity area" by Irène Xueref-Remy et al."**

_General Comments_

_REF.2 : This paper analyzes nearly 1 year of CO$_2$ data from the Paris megacity greenhouse gas measurement network. The analysis focuses on deciphering the CO$_2$ observations on diurnal and seasonal time scales, and includes a careful examination of the influence of the atmospheric boundary layer height (ABLH), wind speed and direction, and local anthropogenic emissions on these signals. The measurement network contains six total_
20 _sites across Ile de France spanning a range of conditions from rural to the Eiffel Tower in the heart of Paris. The report presents measurements that provide an important baseline for emissions from Paris and for comparison to other global megacities._

Authors : We thank Referee 2 very much for her/his careful reading of our paper and for her/his constructive
25 comments. We answer to each point hereafter. The first author apologizes again for the time that it took us to send our reply, due to her particular situation as she recently left LSCE to move to another institute in the south-east of France.

_Specific Comments_

_REF.2 : The authors present a detailed analysis of the CO$_2$ observations based on time, location, and wind speed/direction to infer the seasonal influence of local and background contributions at each site. This analysis is_
35 _largely qualitative, but could be made far more quantitative and definitive if based around back trajectory analyses, such as those shown in Figure S1. We strongly suggest that the discussion of Section 3.1 be expanded and used to validate the conclusions of Section 3.5 which appear to be based on site wind measurements._

Authors : We thank Ref.2 for these suggestions. We expanded a bit more the discussion of Section 3.1 and attempted to use this discussion to consolidate the conclusions of Section 3.5 as far as possible, but the backtrajectories (Figure S1) deliver a qualitative information rather than a quantitative one. Indeed, we produced these backtrajectories using a public tool (HYSPLIT) with a 2.5° x 2.5° wind resolution, and this
5   resolution is much too low to decipher differences between the Paris sites, that are distant by a few kilometers to dozens of kilometers only. Furthermore, this low resolution can only give a gross estimate on the synoptic air mass fluxes between MHD or the Ruhr/Benelux area and the Paris megacity region. A quantitative analysis of the wind trajectories would require a dedicated model with a much finer resolution. This would require consequent work in terms of development, time calculation and analysis, and we therefore think that it would
10  represent another study in itself, that is outside the scope of this paper. This information and the limitation of the HYSPLIT backtrajectories analysis was added to the section.

Furthermore, we moved Fig. S2 as Fig. 4a and Fig. 4b into the main text and we added the following sentences to the discussion to make it more complete :
15  "On Fig. 4a and Fig. 4b, as expected, wind direction and windspeed appear to be part of the main controlling factors of the $CO_2$ mixing ratio values recorded in the different stations. The urban and peri-urban stations are characterized by higher mixing ratios and a much larger variability than the rural and background sites. The highest variability is observed on the GON timeseries, followed by EIF and GIF. We note as well that the highest mixing ratios recorded at the southern rural sites (TRN50 and TRN180) and remote station of MHD occur usually
20  during local events, likely from the influence of local emissions, or remote events with northeast winds that passed over Benelux and Ruhr areas (see backtrajectories in S1) and got loaded with anthropogenic emissions (Xueref-Remy et al, 2011) before reaching IdF. We also observe simultaneous variations between the sites for the local wind class: for example peaks of $CO_2$ mixing ratio are observed in all the stations of IdF in mid-February and the end of March 2011, which correspond to two pollution events reported by AIRPARIF
25  (www.airparif.asso.fr). However, there are some other dates (not reported by AIRPARIF as pollution events) during which the $CO_2$ mixing ratio peaks at the urban and peri-urban stations and also sometimes at the rural stations (ex: 20-25 August and 22-25 October). The wind classification applied on the datasets will be further used to better assess the general features of the $CO_2$ seasonal cycles, and a much finer wind analysis will be conducted in section 3.5.2 to assess the role of local, regional and remote emissions on the $CO_2$ timeseries
30  collected within the Paris observation network."

REF.2 : The study concludes that the level of $CO_2$ enhancement varies with urbanization level local to the site; however, the paper does not directly discuss estimates of enhancement relative to background (or the concept
35  of background) until much later in the paper. While diurnal and seasonal variability and the gradients between sites are the primary focus of this paper, background estimation is an important topic and which merits more introduction. Overall, there are two key points that should be incorporated: (1) the concept of background should be defined relative to the domain of interest and (2) a single site may not represent background $CO_2$ mole fractions under all meteorological conditions.
40
Authors : We defined ''background'' in the Introduction section as the $CO_2$ mole fraction without the contribution of the regional emissions (p. 3, l. 18). By regional, we mean the Paris megacity region i.e. a radius of about 100 km around the Paris center.

We made this spatial scale clearer in the Introduction section and elsewhere in the paper to address the point (1) mentioned above, by quantifying our domain of interest (regional scale limited to ~100km of radius around Paris).

Regarding point (2), we fully agree that a single site may not represent background $CO_2$ mole fractions under all meteorological conditions, as illustrated with our study of MHD in this paper, and in two previous papers of this team [Bréon, F. M., Broquet, G., Puygrenier, V., Chevallier, F., Xueref-Remy, I., Ramonet, M., Dieudonné, E., Lopez, M., Schmidt, M., Perrussel, O., and Ciais, P.: An attempt at estimating Paris area $CO_2$ emissions from atmospheric concentration measurements, Atmos. Chem. Phys., 15, 1707-1724, https://doi.org/10.5194/acp-15-1707-2015, 2015 ; Staufer, J., Broquet, G., Bréon, F.-M., Puygrenier, V., Chevallier, F., Xueref-Rémy, I., Dieudonné, E., Lopez, M., Schmidt, M., Ramonet, M., Perrussel, O., Lac, C., Wu, L., and Ciais, P.: The first 1-year-long estimate of the Paris region fossil fuel $CO_2$ emissions based on atmospheric inversion, Atmos. Chem. Phys., 16, 14703-14726, https://doi.org/10.5194/acp-16-14703-2016, 2016]. We addressed this point more specifically in section 3.4.

*REF.2 : Additionally, the paper should use $CO_2$ enhancement values relative to some chosen background rather than absolute $CO_2$ values (eg 410 ppm) since the global background will surpass even these "elevated" values in the near future.*

Authors : As a consequence of the previous point, the term ''background'' remains more a concept than a quantity in our study and is not given as a numerical quantity value. We therefore do not report enhancements here (not even with a fixed value like 410 ppm). We reported dynamic enhancements in the above-mentioned studies of Bréon et al. (2015, Fig. 7) and Staufer et al. (2016, Fig. 4) but under specific meteorological conditions. We are trying to get a more general assessment in an on-going study, but this one is at early stage and clearly distinct from the research that we are reporting here.

*REF.2 : The challenges of analyzing these measurements raises several priority questions regarding the Paris network. We note that the INFLUX network in Indianapolis, IN USA contains 13 towers for a smaller, less populated urban area and approximately 1/10th the emissions of Paris/IdF [Turnbull, Jocelyn C., et al. "Toward quantification and source sector identification of fossil fuel $CO_2$ emissions from an urban area: Results from the INFLUX experiment." Journal of Geophysical Research: Atmospheres 120.1 (2015): 292-312]. We would have expected some discussion of the density of the Paris network, the potential benefit of additional sites, and where they would ide ally be located for maximum impact. This is particularly relevant for the "background" discussion since it is clear that Mace Head alone is insufficient for this analysis and that a full understanding of Paris $CO_2$ monitoring may well require observations from as far away as the Ruhr or the Benelux region.*

Authors: Through the $CO_2$-Megaparis project, we were funded for 3 new sites on top of 2 existing national ICOS sites. We chose to deploy these new sites on the axis of the dominant winds (NE/SW) in order to optimize the amount of available data. The extension of the network for inverse modeling purposes is discussed by Staufer et al. (2016, Section 4.3) who conclude to the need of 8 more sites in the suburban/rural border of the city. Longer prospects are the topic of Wu et al. (2016) [Wu, L., Broquet, G., Ciais, P., Bellassen, V., Vogel, F., Chevallier, F., Xueref-Remy, I., and Wang, Y.: What would dense atmospheric observation networks bring to the quantification of city $CO_2$ emissions?, Atmos. Chem. Phys., 16, 7743-7771, https://doi.org/10.5194/acp-16-7743-2016, 2016.]. Furthermore, in order to improve our understanding and modeling of the vertical transport of urban $CO_2$

emissions, we mentioned in our conclusion the need to develop more measurements in the center of Paris, especially $CO_2$ vertical profiles on the Eiffel tower (p.24 lines 13-15).

We completed the abstract, section 3.2.2, section 3.4 and the conclusion to better address this point.

*REF.2 : Given the topographical similarities of Paris and Indianapolis, we were also surprised that more discussion was not presented comparing the $CO_2$ concentration "plume" patterns from these urban areas.*

Authors : If we compare Figure 2d of Turnbull et al. (2015) and Figure 7 of Bréon et al. (2015), we see
10 enhancements of a few ppm in both cases. We could report this information, but the background is defined differently in each paper and the comparison would remain rather qualitative and artificial. Also, we would like to remember that we have much less sites in Paris than in Indianapolis.

*REF.2 : Newman et al. [Newman, S., et al. "Diurnal tracking of anthropogenic $CO_2$ emissions in the Los Angeles*
15 *basin megacity during spring 2010." Atmospheric Chemistry and Physics 13.8 (2013): 4359-4372] showed diurnal patterns for $CO_2$ from the Los Angeles megacity, but there was no comparison made with these data. This is particularly relevant since Los Angeles $CO_2$ emissions are well known to be dominated by vehicle/transportation and impart significant rush hour maxima (0700-1000 and 1500-1900) that are absent from all but the EIF signals in Paris. The arguments for winter vehicle emissions in Paris are not obvious from the figures as presented.*

Authors : We thank Ref.2 to advice us that it was worthwhile to make some comparison between our study and the Newman et al (2013)'s one. In IdF, according to the AIRPARIF 2010 inventory, 29% of the $CO_2$ emissions are due to traffic. And indeed, in winter, the signature of traffic can be seen on the MON, GON and GIF diurnal cycles, as in the Los Angeles study of Newman et al (2013), through two peaks at rush hours (cf Fig.8). This
25 feature was already observed in Paris center and GIF as reported in the Lopez et al (2013) $CO_2$, CO and NOx winter study. In other seasons in Paris, when vegetation is active, the signature of traffic is hidden by the biospheric activity and also by the boundary layer dynamics, which both drive the shape of the diurnal cycle. This information was added to section 3.2.1.

30 *REF.2 : The Eiffel Tower (EIF) site offers unique observations that might be more fully exploited in future studies. Complete diurnal and day of week sampling at this site would enable greater understanding of variability across the network. Adding vertical profile measurements at eg 50, 100, and 200 m to complement the 300 m inlet height would add tremendously to understanding the ABLH/$CO_2$ linkages as well as providing different spatial sensitivity footprints within the Paris/IdF region. Increasing the sampling of meteorological fields at different*
35 *heights would also prove valuable.*

Authors : We thank Ref.2 for this comment and we mentioned more explicitly in our text that we effectively do indeed plan to carry out such measurements (mentioned both in section 3.2.2 and in the conclusion=.

40 *REF.2 : It would be useful to present the more details about the AIRPARIF inventory in the text, e.g., how it was constructed, its spatial resolution, etc.*

Authors : The AIRPARIF inventory is well detailed in the Bréon et al (2015) paper. We paid attention to better refer to this paper and we also added some key information about this inventory in the Introduction section.

*REF.2 : Comments on treatment of MHD and "background": P.7, line 6: MHD is described as a remote location.*
*State here that this site was specifically evaluated as a potential background site.*

Authors :  This was modified accordingly.

*REF.2 : See also comments below. P.16, line 3-4: The conclusion that MHD is not a relevant site for background on the seasonal scale does not seem to be fully supported by results. In some instances, a site that is classified as rural or peri-urban (or possibly urban) could represent background mole fractions under certain meteorological conditions. Selection of background can performed with using many methods, including meteorological filtering, analyzing tracer/tracer correlations, or evaluating the stability of observations. There is a significant body of literature detailing methods for selecting observations that represent background mole fractions (as an example, see Ruckstuhl et al., 2012, http://www.atmos-meas-tech.net/5/2613/2012/).*

Authors : We defined our background as the $CO_2$ mole fraction without the contribution of remote emissions. By remote, we mean out of the Paris megacity region (i.e. ~100 km around the center of Paris). Our observations show clear differences of several ppm between MHD and the rural site of TRN for example, which has been already demonstrated to be poorly influenced by the Paris megacity emissions. This shows that MHD is not a relevant background for the Paris megacity region. Regarding background calculation, we are aware of the complexity of the question and of the different methods available, but as we explained above this question is outside the scope of this paper. We modified the text to make these points clearer, as follows:
"Ignoring the specific case of EIF (section 3.2.3), throughout the year we observe that the monthly mean $CO_2$ concentration increases with the vicinity of the station to larger $CO_2$ emission sources. The maximum $CO_2$ enhancement compared to MHD is observed at GON which is our most anthropogenically influenced station (from 6.8 ppm in July to 27.5 ppm in December). Similarly to what is observed at the diurnal scale (section 3.2), differences of several ppm are also observed between our rural sites and MHD, while the differences between the rural/peri-urban/urban stations in IdF is of the same order of magnitude. These differences of concentration between the stations located in IdF and MHD vary with the season, the seasonal cycle being much more well defined in the Paris rural stations than in MHD due to a higher biospheric activity in the IdF region than on the western coast of Ireland. This implies that background values of $CO_2$ in IdF (i.e. without the impact of Paris emissions) should be defined at the regional scale near Paris (~100 km) and not at the continental scale in MHD. Furthermore, in Section 3.1 we explained that the $CO_2$ concentration fluctuates with the origin of the airmasses that can be much variable, and therefore, specific regional background should be selected in function of the wind direction, as also mentioned for the case of Indianapolis (Turnbull et al, 2015). In conclusion, MHD appears not to be relevant as a background site for defining the atmospheric plume of $CO_2$ in the Paris region at the seasonal scale as well. Regional background stations (~100 km) seem to be much better suited for urban regional studies in Paris and elsewhere in the European continent. Several methods are available to extract a background signal from a timeseries (e.g. Ruckstuhl et al, 2012 ; Ammoura et al, 2016). Quantifying precisely the Paris background signals values as well as the Paris plume and its variability requires a dedicated analysis that is outside the scope of the present paper : it will specifically adressed within another dedicated study."

*REF.2 : P.18, lines 5-7: The conclusion here that MHD is not a relevant background site for Paris or other Western European cities also does not seem to be fully supported by the evidence. The definition of background depends on the domain of interest and also the timescale. For example, a single site may not be relevant for selecting background observations at all times and under all conditions. It is not clear whether there are ever any meteorological conditions that support MHD as a relevant local and/or regional background site. In general, the conclusions regarding MHD could be further supported by the evidence from the back trajectory and fine wind sector analysis (Sections 3.1 and 3.5.2) and/or the Supplemental materials (Figures S1 and S2).*

Authors : We explained above why MHD is not a relevant background site for the Paris megacity region or other continental Western cities. We made this point clearer through the paper and also made the best use of the backtrajectories presented in section 3.1 to consolidate our argumentation, given the limitation inherent to their low spatial resolution.

*Technical Corrections*

*REF.2 : The manuscript could further benefit from more labeling figures to classify sites as "Urban" and "Periurban/Rural".*

Authors : This was done through the paper.

*REF.2 : Regarding analytical methods, the paper would also benefit from stating early on that all 7 sites (new and previously published) are on the same $CO_2$ calibration scale (WMO X2007), use similar analytical procedures, and have relatively small uncertainties. This could be stated perhaps in the introduction or at the beginning of the methods section.*

Authors : We thank Ref.2 for this suggestion and we mentioned that point in the Introduction section.

*REF.2 : Introduction: Suggest presenting the site code QUA to associate this site with the ABLH measurements from the time they are first introduced.*

Authors : This was modified accordingly to Ref.2's suggestion.

*REF.2 : Figure 6: May help to include inlet heights. Also, maybe label plots as Urban, peri-urban, rural/remote, etc.*

Authors : We followed Ref.2's advice and this was done throught the paper.

*REF.2 : P.4, line14: The reference Schmidt et al., (2014) first appears here, however it was not included in the list of references at the end of the paper.*

Authors : This was corrected accordingly.

*REF.2 : P.7, line 23: The authors mention the cell temperature of the analyzer at the EIF site was modified to undergo cell temperature set point at 60_C, however do not discuss what impact (if any) this may have on the results. Details of such analytical differences could be useful for others in the community conducting studies using similar analyzers.*

Authors : Indeed, no specific impact of the set point of 60°C was observed in the results. We added this piece of information in the corresponding section.

*REF.2 : Please be clear when meteorological data is measured vs. modeled e.g. add theword modeled to Figure 3 caption.*

Authors : This was made clearer.

*REF.2 : Figure 5: might be useful to add inlet heights to the site key*

Authors : We made some tests and adding this information to the site key made it quite busy. Therefore, we added this information in the legend of the figure.

*REF.2 : Figure 7: What is the difference between the violet and red traces? Please describe in the text.*

Authors : The violet trace uses only $CO_2$ hourly data that are concomitant to ABLh hourly data. The red trace uses all $CO_2$ hourly data points available for the relevant season. This was better explained in the legend of the figure.

*REF.2 : Figure 12, the wind roses highlighting $CO_2$ concentrations and indicating the origin of the air masses being measured, was particularly interesting. Unfortunately, the discussion of this figure includes a lot of discussion of background, but it isn't clear exactly how the authors determined the background. I would also like to see explicit explanation of how the seasonal adjustments to the $CO_2$ concentrations were made.*

Authors : As we explained here before, the term ''background'' is not quantitative in this paper, but is a concept and represents the contribution of remote fluxes (i.e. not from the Paris megacity area). We made this clearer through the whole paper.
The seasonal adjustment was done on the $CO_2$ hourly mean dataset of each station by : 1/ computing the annual mean of the dataset ; 2/ computing the monthly seasonal index for each month by dividing the monthly mean by the annual mean of the dataset ; 3/ interpolating the monthly seasonal index dataset to an hourly scale dataset ; and 4/ dividing the hourly dataset by the hourly seasonal index. This information was addedto the text.

*REF.2 : Table 4: The use of "N" is confusing since this is a percentage, not an integer. Consider renaming "coverage"?*

Authors : We followed Ref.2's suggestion and changed N for Coverage in Table 4 (now Table 3).

*REF.2 : Page 10 Line 20: shouldn't this section be titled, "Results and Discussion?"*

Authors : Yes, thank you, this was modified according to this suggestion.

*REF.2 : Page 12 Lines 28-32: What about the effect of inlet height? MON is much lower than TRN50.*

Authors : We recognize that this point merits more consideration and it is now discussed in the paper. The following changes were done :
"It is noticeable that the mean winter concentration is about 6 ppm higher at MON than in TRN50. Both stations are in rural environment, but MON is closer to Paris than TRN. As the signals are quite similar in summer, this difference can not likely be explained by the biospheric activity, and is more probably partly due to a higher anthropogenic influence in MON. However, we need here to take into account the difference of the stations inlet height (9 m AGL at MON, 50 m AGL at TRN50) : as shown in Schmidt et al (2014) for the 2010 winter season at Trainou, during daytime $CO_2$ concentration measured at 10 m AGL and 50 m AGL are similar, but this is not the case during nighttime when the $CO_2$ concentration is about 3 ppm higher at 10 m AGL than at 50 m AGL because atmospheric mixing is not existent at night and $CO_2$ sources accumulate near the surface (Denning et al, 1995). This means that the difference between MON and TRN at the inlet height of MON is of the order of 6 ppm during daytime and twice as low during nighttime. This is consistent with the hypothesis of a higher impact of anthropogenic emissions in MON than in TRN, that according to AIRPARIF are lower during nighttime than during daytime, although we do not observe the same order of magnitude (AIRPARIF gives a ratio of daytime over nighttime emissions equals to 3 to 4 in wintertime, while we observe a ratio of 2 ; see Fig.3 in Bréon et al, 2015). Remember though that the diurnal cycle of the emissions inventory is an average for the whole IdF region, and not only for the MON area. The impact from local sources and/or the $CO_2$ emission plume of the Paris megacity on MON will be further inferred from the wind analysis in section 3.5."

*REF.2 : Page 13 Line 6: Max interseasonal difference is higher than the mean annual afternoon dispersion: what does this imply?*

Authors : This implies that the seasonal variability is higher than the mean dispersion of the fluxes. We added this information in the text.

*REF.2 : Page 13 Line 10: "strong impact of regional $CO_2$ emissions variability:" why? Please elaborate a bit more.*

Authors : We removed the term regional that was confusing here.

*REF.2 : Page 14 Lines 5-34: Please put the seasons in the same order in the text and in the plot.*

Authors : The text was modified accordingly.

*REF.2 : Page 18 Lines 21-22: Define local in terms of spatial scale.*

Authors : Local is here defined as "less than 10 km". We made this clear in the corresponding section.

We thank again very much Referee 2 for all these constructive comments that helped us to improve our analysis
10  and our paper.

Best regards,
Irène Xueref-Remy
Dec. 11[th], 2017

**Revised manuscript (marked-up version)**

[revised manuscript text omitted]

Based on the 2010 population censuseriteria, the Paris conurbation is with metropolitan area has 10.5 million inhabitants and is ranked the 21$^{st}$ megacity in the world and the 2$^{nd}$ in Europe after Moscow (United Nations, 2011b). Paris is centered in the region Île-de-France (IdF) that contains 18% of the French population (INSEE, 2012) while covering only 2% of the territory. The emission inventory reported by AIRPARIF (Association de surveillance de la qualité de l'air en IDF: http://www.airparif.asso.fr) estimates that IdF emitted a total of 41.9 Mt of $CO_2$ in 2010, i.e. 12% of French anthropogenic $CO_2$ emissions (source: CITEPA, 2012, www.statistiques.dvpt-durable.gouv.fr). It is based on the combination of benchmark emission factors and activity data for about 80 emission sectors and delivered every year (3 years after the year of the emissions reporting). It is built at a high spatio-temporal resolution (1x1 km$^2$, 1 h) for the whole IdF domain. The temporal resolution is based on the interpolation of mean hourly diurnal cycles of emissions constructed for 5 typical months (January, April, July, August and October). Detailed information can be found in Bréon et al (2015). However, there is no independent assessment of the regional $CO_2$ emission estimates given by the AIRPARIF inventory, which is based on the combination of benchmark emission factors and activity data. The associated uncertainties are estimated to be 20% of the total $CO_2$ emitted by month, but they are also sector dependent and can reach several tens of percent for some sectors, as also discussed in Rayner et al. (2010).

In the recent last years, there has been a growing international interest in quantifying urban $CO_2$ fluxes from atmospheric top-down approaches (e.g. Duren and Miller, 2012; Mc Kain et al., 2012). Large projects developed emerged in Indianapolis (Influx: http://influx.psu.edu ; e.g. Turnbull et al, 2015 ; Lauvaux et al, 2015), Boston (http://www.bu.edu/today/2013/the-climate-crisis-measuring-boston-carbon-metabolism/ ; McKain et al, 2012), Los Angeles (Megacities: http://megacities.jpl.nasa.gov/portal/ ; e.g. Newman et al, 2013 ; Verhulst et al, 2016) and in our case Paris ($CO_2$-Megaparis: http://co2-megaparis.lsce.ipsl.fr ; e.g. Lac et al, 2013 ; Bréon et al, 2015 ; Ammoura et al, 2016 ; Staufer et al, 2016). These projects rely on the development of urban atmospheric in-situ $CO_2$ monitoring networks that should ideally include, all along the dominant wind paths: 1/ regional stations upwind of the city to characterize the regional background $CO_2$ dry air mole fraction (i.e. without having the impact of the regional emissions - regional is here defined within a radius of ~100 km around the center of Paris); and 2/ regional stations in the city and downwind of it (that will integrate both the background signal and the peri-urban/urban ones). In the following, the term dry air mole fraction is simplified by concentration and is expressed in the part per million (ppm) unit.

[revised manuscript text omitted]

5 the case of the Eiffel tower station. We also estimate the weekday versus weekend variability (section 3.3) and . We then analyze the seasonal variations of the $CO_2$ concentration at each site (section 3.4). Finally, we study the role of wind speed and direction on the $CO_2$ signal collected at the five regional network stations (section 3.5) and we assess the impact of local (<10 km), regional (10-100 km) and remote (> 100 km) fluxes on the observed $CO_2$ concentrations. We come to conclusionsconclude on the representativeness of each site for assessing the how the Paris $CO_2$ emissions impact the

10 atmospheric $CO_2$ concentration at the regional scale, plume and on the lessons learned for regional urban network design that we learned from this study.

**2 Experimentals**

**2.1 The measurement network**

**2.1.1 Geography of IdF and $CO_2$ emissions from the Paris region and Western Europe**

15 Paris is located in the region of IdF in a relatively flat area and benefits from a temperate climate, with frequent rain events in all seasons and changing weather conditions. IdF covers 12011 km$^2$ i.e. only 2.2% of the national territory. In 2010, land usage was 47% by agriculture, 31% by forests and natural areas and 22% by urbanized areas (http://www.insee.fr/fr/themes/tableau.asp?reg_id=20&ref_id=tertc01201), the last sector increasing in recent decades (United Nations, 2011b). In 2010, anthropogenic $CO_2$ emissions of IdF came from the residential and commercial buildings

20 (43%), road traffic (29%), industry and energy production (14%), agriculture (5%), wastes (4%), aircrafts (0-915 m ASL) and airport infrastructures (4%) and worksites (1%) (AIRPARIF, 2010). The CDG airport (relatively close to GON, see below) represents about 78% of the aircrafts and airports $CO_2$ emissions in IDF, with ~60% emittissued from airplane traffic on the tarmac and in flight (below 915 m ASL) (ADP, 2013; AIRPARIF, 2013). The Orly airport (16 km east of GIF) emits ~27% of the CDG airport $CO_2$ emissions (AIRPARIF, 2013). Le Bourget airport (close to GON, see below) $CO_2$ emissions

25 are much smaller (~1.6% of the CDG one, AIRPARIF, 2013).

Figure 1 shows the total annual $CO_2$ emissions emitted from IdF at the resolution of 1x1 km$^2$ (AIRPARIF, 2010). As shown on Fig. 1, there is a large spatial variability of $CO_2$ emissions in IdF which is mainly driven by the population density and the location of highways. Each year, average emissions in the center of Paris are estimated to be ~70 000 t$CO_2$ km$^{-2}$ compared to ~5000 t$CO_2$ km$^{-2}$ at the surburban borders. Emissions have a temporal variability on diurnal, synoptic and seasonal scales,

30 mainly because $CO_2$ emitted by heating varies with temperature and season, and $CO_2$ emitted by traffic changes with the time of the day, day of the week and vacation periods (see Fig.3, Bréon et al, 2015). Figure 1 also shows emissions from the industry and energy production, that come from point sources here distributed on 1x1 km gridcells. According to

AIRPARIF, these sources are located mostly in the north and north-eastern areas of Paris compared with the southern part of Paris (Lopez et al, 2013). Detailed and public information on a total of one hundred and twenty three point sources of $CO_2$ in IdF can be found online for the year 2010 at the following address: http://www.georisques.gouv.fr/dossiers/irep/form-etablissement/resultats?annee=2010®ion=11&polluant=131#/. Some of these point sources are located within a few kilometers of the sampling sites as detailed in section 2.1.2 and may have an impact on the observed $CO_2$ concentration, as discussed in section 3.5.2. Figure 2 shows the distribution of fossil fuel and cement $CO_2$ emissions in Western Europe extracted from the EDGAR v4.0 emission inventory (http://edgar.jrc.ec.europa.eu/, 2009), highlighting large anthropogenic emissions spots in the Paris megacity, but also in the Benelux area, the Ruhr valley and the London megacity that may enrich the synoptic air masses with high $CO_2$ concentrations before they reach the Paris region.

**2.1.2 Sampling sites**

The location of the observation sites are represented on Fig. 1 and Fig. 2. Table 1 gives their exact geographical coordinates. The sites were carefully chosen so that they would not contaminate the $CO_2$ measurements by their own emissions.
The Eiffel tower station (EIF) was installed on the highest floor accessible to tourists, in a closed room of 1.5 m$^2$ under the stairs providing access to the Tower communication antennas. To prevent contamination by the visitors' respiration, the air inlet was elevated to about 15 m above the last floor accessible to tourists, at the antenna level (317m AGL), where it was protected from uplifted air by several intermediate metallic floors. The instrument was set-up into a Faraday cage to avoid interferences fromwith strong electromagnetic radiations from the antennase. The location of the Eiffel tower is not exactlyfully central within Paris, and t. The 0-180° (N, E and S) wind sector of the station is exposed to a larger urbanized and industrialized- area than the 180-270° sector (S to W). In the 0-180° wind sector, the urbanized area covers a radius of about 20 km and includes two large emitting point sources that are the waste burning facility of Ivry (in the SE direction of the Eiffel Tower) and the heating facility of Saint-Ouen (in the North). In the 180-270° wind sector, the urbanized area extends barely within a 10 km radius before entering into broad-leaved trees forests covering ~2300 ha. The 270°-360° wind sector is also mostly urbanized over a radius of about 15 km, although it comprises the woods of Boulogne (about 840 ha) which are located only 2 km NW of the Eiffel tower.

The Gonesse station (GON) was set-up about 20 km north east of the Eiffel tower at the local fire station in a residential area comprising a combination of streets and lawn gardens with a few trees around. The analyzer was hosted in a shelter equipped with a mast of ~4 m standing below the canopy level (~15m AGL). However the distance from the mast to the closest trees was at least 20m and the station was well exposed to wind from all directions. GON is located on a small hill relative to the centre of Paris and in the southerly direction, the station benefits from an open view of the Paris mega city. About 3 to 4 km to the southeast and east of the station is a highway which carries high traffic during rush hours, as early as 5 am local time. The highway connects the centre of Paris and CDG airport, which is located about 7 km northeast of GON. The station is also close to the Bourget Airport located about 2.5 km to the south. Finally, in the W-NW sector, two noticeable industrial

sources located at about 5 km from Gonesse (Fig.1) should be mentioned as they might have an influence on the $CO_2$ measurements (section 3.5.2) : a thermal plant in Sarcelles that emitted 44 kt$CO_2$/year in 2010, and an energy production plant in Le Plessis-Gassot that emitted 128 kt$CO_2$/year in 2010 (source: http://www.georisques.gouv.fr).

The Montgé-en-Goële station (MON) was set up in the small village of the same name with approximately 700 inhabitants located on the middle of the slope of a small hill (~20m high). The analyzer was installed on the top of the 3-floor city hall building (~9 m AGL). The air inlet was set-up on an arm pointing about 1.5m outside of the window towards the south (200°) opening onto fields. The north sector was covered by a few houses situated at the edge of a wood of broad-leaved trees. The city hall is located on the southern side of the main road of the village which approximatively follows a northwest- southeast axis. Most of its close surroundings are agricultural fields and small villages connected by secondary roads. Montgé-en-Goële is located approximately 10 km east of CDG airport. Two noticeable point sources are relatively close to the station (Fig.1) and could influence the measurements (section 3.5.2) : a cement plant 3 km east in Saint Soupplets (43 kt$CO_2$/year in 2010, source: http://www.georisques.gouv.fr/) and a waste burning facility 7 km east in Monthyon (106 kt$CO_2$/year in 2011, source: http://www.georisques.gouv.fr/). MON was considered as a NE rural site for the Paris megacity.

The Gif-sur-Yvette station (GIF), previously described in Lopez et al (2012) and Lopez et al (2013), has been running continuously since 2001 at LSCE (Laboratoire des Sciences du Climat et de l'Environnement). The air inlet is set up on the roof of a building at 7 m AGL. The site is located ~20 km south-west of the centre of Paris on the Plateau de Saclay and surrounded mainly in the 0°-90° sector by agricultural fields and by a few villages. A few hundred meters further in this direction, a national road passes on a north-south axis with high traffic levels during the morning and in the evening during rush hours. About 1 km further in the 270-360° sector, the atomic and environmental research agency (CEA of Saclay) holds approximately 7000 employees and is equipped with a thermal plant (17 kt$CO_2$ in 2010, source : http://www.georisques.gouv.fr/) that is further surrounded by agricultural fields. In the last wind sector (90°-270°), a band of forest of about 1 km depth extends along the west to east axis down to the bed of the Yvette river. A noticeable point source in the vicinity of GIF, a thermal plant located in Les Ulis, is located about 5 km further south-east (98.5 kt$CO_2$ in 2010, source: http://www.georisques.gouv.fr/). The GIF station is located roughly at the same distance from the Eiffel tower as GON. However, the environment is more rural in GIF than in GON so that we can label GON as a residential peri-urban site and GIF as a remote peri-urban site - although it is not as rural as the site at MON. Orly airport is located about 16 km East of GIF.

[revised manuscript text omitted]

**2.2.2 Data processing and quality control**

The CRDS $CO_2$ data were calibrated by applying a linear fit to the $CO_2$ concentration of the calibration tanks
5    as measured by the CRDS analyzer vs the  $CO_2$ concentration as measured by the GC. Gas equilibrium issues implied retaining only the last calibration cycle over the 4 cycles at MON and GON (and of the 2 cycles at EIF) to compute the calibration equation. For all of the calibration and target gas cylinders, the CRDS $CO_2$ concentration was calculated as the average of the last 5 minutes of each gas. The accuracy of the datasets was calculated as the mean difference between the $CO_2$ concentration reported by the CRDS
10    analyzer and by the GC for the target gas. The long-term repeatability of each dataset was calculated as the standard deviation of the mean concentration of the target gas reported by the CRDS analyzer over the year of observations.

Table 2 summarizes the accuracy ($\leq$ 0.13 ppm) and repeatability ($\leq$ 0.38 ppm) calculated from the 5 minute averaged data for MON, GON and EIF. As expected, the dataset of EIF shows larger deviations compared to GON and MON due to less frequent calibration and target gas measurements and a shorter calibration procedure.

15    The data of GON, EIF and MON were automatically filtered against cavity pressure (P) and cavity temperature (T) departure to the set points ($P_0$ and $T_0$) according to the ICOS procedure (Hazan et al, 2016), keeping only points for which $|P-P_0|<0.1$ Torr and $|T-T_0|<0.004°$ C for MON and EIF ($0.006°$ C for EIF). Furthermore, dead volumes in the set-up lead to instability in the response of the analyzer for 2 minutes after switching from one gas line to  another. These 2 minute periods were automatically removed from the datasets.
20

The data was also manually inspected to remove  $CO_2$ spikes due to very local influences (e.g. fire training at the GON station, breathing of a maintenance operator on the sampling inlet…). Very local influences were indentified from the short duration of the events (a few seconds to some minutes) and from the large standard deviation of the $CO_2$ averages associated with these events. This amounted to – less than 1% of the total

[revised manuscript text omitted]
 were calculated for the Paris city over the full period of study. We used wind fields from the NOAA-NCEP/NCAR reanalysis data archives, at a 2.5° x 2.5° and 6 h resolution (http://rda.ucar.edu/datasets/ds090.0/). The back trajectories were run for 72 h backwards and started at 10 m AGL. They were then aggregated on monthly plots that are shown in the supplementary material (Fig.S1). In all cases, the monthly clusters illustrate the high variability of the origin of the air masses, which could pass over high $CO_2$ emissions areas such as the megacity of London, the Benelux or the Ruhr regions before reaching IdF. The air masses could also be advected from clean areas such as the Atlantic Ocean, or from biospheric regions such as in the middle of France. This high atmospheric transport variability implies that the Paris regional $CO_2$ background signal may be highly variable depending on the synoptic conditions and that wind direction and speed are key parameters to take into account in order to understand the $CO_2$ concentrations recorded at the different sites. The Hysplit model does not have a sufficient resolution to get a more precise and quantitative information on the influence of local, regional and remote emissions on our $CO_2$ observations, and getting higher resolved transport information would require a very specific (and expensive) modeling work that is out of the scope of this study. Therefore, in order to go further into the analysis, we used the modeled meteorological fields presented in section 2.4 to classify the $CO_2$ hourly timeseries into six wind classes (Figure 4a and Figure 4b).

The hourly time series of $CO_2$ used in this work are shown in the supplementary material S2 and are colored according to six wind classes. The *local class* is defined for wind speed less than 3 m s$^{-1}$ and the *remote class* for wind speed higher than 9 m s$^{-1}$. For wind speeds between 3 and 9 m s$^{-1}$, we defined four remaining classes according to the wind direction: *northeast* (NE), *northwest* (NW), *southeast* (SE) and *southwest* (SW). As an example, in GIF the partition of the air masses between the different wind sectors over the full period of study is the following: 16% from the NE, 15% from the NW, 24% from the SW, 7.5% from the SE, 36% from the *local class* and 1.5% from the *remote class*. These classes will be used to better assess the general features of the $CO_2$ seasonal cycles, although a much finer wind analysis will be conducted in section 3.5.2.

On Fig. 4a and Fig. 4b, as expected, wind direction and windspeed appear to be part of the main controlling factors of the $CO_2$ mixing ratio values recorded in the different stations. The urban and peri-urban stations are characterized by higher mixing ratios and a much larger variability than the rural and background sites. The highest variability is observed on the GON timeseries, followed by EIF and GIF. We note as well that the highest mixing ratios recorded at the southern rural sites (TRN50 and TRN180) and remote station of MHD occur usually during local events, likely from the influence of local emissions, or remote events with northeast winds that passed over Benelux and Ruhr areas (see backtrajectories in S1) and got loaded with anthropogenic emissions (Xueref-Remy et al, 2011) before reaching IdF. We also observe simultaneous variations between the sites for the local wind class: for example peaks of $CO_2$ mixing ratio are observed in all the stations of IdF in mid February and the end of March 2011, which correspond to two pollution events reported by AIRPARIF (www.airparif.asso.fr). However, there are some other dates (not reported by AIRPARIF as pollution events) during which

the $CO_2$ mixing ratio peaks at the urban and peri-urban stations and also sometimes at the rural stations (ex: 20-25 August and 22-25 October).  The wind classification applied on the datasets will be further used to better assess the general features of the $CO_2$ seasonal cycles, and a much finer wind analysis will be conducted in section 3.5.2 to assess the role of local, regional and remote emissions on the $CO_2$ timeseries collected within the Paris observation network.

**3.2 $CO_2$ diurnal cycles**

**3.2.1 Mean $CO_2$ diurnal cycles**

Diurnal cycles of atmospheric $CO_2$ are affected by local sources and sinks, regional transport and ABL dynamics (Fang et al., 2014; Garcia et al., 2012; Rice et al., 2011; Artuso et al., 2009; Gerbig et al., 2006). The mean $CO_2$ diurnal cycles and

10 associated 1-$\sigma$ standard deviation are shown in Fig. S3 for the different stations.

Noticeable differences are observed between the sites.

The diurnal amplitude of the $CO_2$ concentration from the lowest to the highest is 2.6 ppm (MHD), 6.5 ppm (TRN180), 11.2 ppm (EIF), 14.9 ppm (MON), 15.5 ppm (TRN50), 18.2 ppm (GIF) and 30.6 ppm (GON). While the $CO_2$ diurnal pattern at TRN can mostly be explained by  biosphere activity and vertical dilution in the ABL (Schmidt et al, 2014), the peri-

15 urban and urban stations are also expected to be strongly influenced by the diurnal cycle of  Parisian anthropogenic sources. For all sites except EIF, the maximum concentration occurs in the late night/early morning (4-5 h for TRN50, MON, GIF and GON; 7-8 h for TRN180) when the ABL is the most shallow, vegetation respires and rush hours traffic occurs (5 h - 9 h, source : http://www.dir.ile-de-france.developpement-durable.gouv.fr/les-comptages-a174.html). The minimum of the cycle occurs in the afternoon (14 h to 17 h) when the ABL is the deepest and well mixed and during

20 seasons when the vegetation photosynthesis is active. Note that, as for the case of Los Angeles (Newman et al, 2013), the annual mean $CO_2$ concentration does not peak during rush hours, meaning that traffic is not the primary driver of the shape of the annual $CO_2$ diurnal cycle at the Paris surface stations, nor are other anthropogenic sources, but rather, the main drivers seem to be the biospheric activity and the ABL dynamics, deadening the diurnal features of anthropogenic emissions. The case of EIF is specific due to its elevation and a strong interaction of urban $CO_2$ emissions with the ABL cycle (see section

25 3.2.3). As a consequence, the maximum  $CO_2$ concentration at EIF occurs in the mid-morning (10 h) and its minimum is at night (0 h).

Comparing the 50 and 180 m levels at TRN, we observe  a vertical gradient of  $CO_2$ concentration, along with a phase shift of the diurnal cycle: the maximum concentration is observed at 5 h  at TRN50 versus 7 h  at TRN180, due to the coupling of the $CO_2$ fluxes with the ABL cycle.  $CO_2$ emitted during the night and early morning

30 by anthropogenic sources and by the biosphere's respiration accumulates near the ground into the shallow nocturnal boundary layer (Schmidt et al., 2014) until the ABL develops in the morning, uplifting $CO_2$ (from 5 h to 7 h ) to the 180m level. In the afternoon, when the ABL is well-mixed and deeper than 180m, the mean difference between the

concentration at the 50 and 180 m levels is very low (0.3 ppm). Furthermore, as noticed in Schmidt et al (2014), the amplitude of the diurnal cycle decreases with increased sampling height as elevated sampling levels are decoupled from the $CO_2$ sources during the night. As reported in Fang et al (2014), this covariance between  biospheric $CO_2$ activities and the ABL dynamics can make it difficult for inversion models to properly reproduce the $CO_2$ vertical gradient and thus,

5 use nighttime data for inversions. During mid-afternoon, the ABL is well mixed and the vertical bias would be very tiny.

There is a significant enhancement in the $CO_2$ concentration observed at the regional stations  compared to MHD. that increases the closer a station is to Paris city (apart from EIF). The  between two sites depends on the time of the day and its variation is mainly

10 driven by the $CO_2$ diurnal cycle at the continental sites. Apart from EIF, the more the station is surrounded by urbanization, the higher is the concentration enhancement compared to MHD, as the average levels of the $CO_2$ concentration recorded at a station increases with a higher proximity to anthropogenic emissions from Paris. The left panels (a-g) on Fig. 5 show that the hourly 1-$\sigma$ variability of the mean diurnal cycle remains quite constant over the day at TRN50, TRN180 and MHD. It is a bit more variable for the rural and remote peri-urban stations that are

15 located within IdF (MON and GIF). The variability changes significantly with the time of the day at EIF and even more at GON. We can conclude that: 1/ the more the station is within the urbanized part of the city, the more variable is the measured $CO_2$ signal, which reflects the spatial and temporal variability of anthropogenic emissions coupled to atmospheric transport fluctuations; and 2/ the MHD signal is several ppm below the continental signals. even at the rural site of TRN that has already been shown not to be significantly influenced by the Paris megacity fluxes (Schmidt et al, 2014).

20 Thus, MHD does not reproduce the background diurnal variability observed in the rural stations of IdF, and is clearly not a relevant background site for continental European urban studies at the diurnal scale and at the regional scale of ~100 km.

The right panels (a'-g') of Fig. 5 show the mean diurnal cycle at each site by season. The influence of anthropogenic

25 activities on the observed $CO_2$ concentration is expected to be the highest in wintertime when emissions from heating are superimposed on traffic and other sources, photosynthesis is minimal and the diurnal ABL is thinner. Although they vary with the time of the day, on average $CO_2$ emissions from traffic are quite constant throughout the year but they vary at the hourly and daily scales (according to the AIRPARIF 2010 inventory : on average, 1.5kt.yr$^{-1}$ during weekends and 2.5 kt.hr$^{-1}$ during weekdays, and up to 4 kt.hr$^{-1}$ during traffic peaks ;

30 see Fig. 4 in Bréon et al, 2015)). On the contrary, emissions from gas combustion (from the residential, the public and the commercial infrastructures that include mostly heating, production of hot water, air conditioning and cooking) show a seasonal cycle (mainly from heating), releasing about 2.5 kt.hr$^{-1}$ of $CO_2$ in the atmosphere in winter versus approximately 1.5 kt.hr$^{-1}$ in summer (AIRPARIF, 2010 ; Bréon et al, 2015). The biospheric fluxes show large diurnal and seasonal cycles, as mentioned in Bréon et al (2015) who

reported net ecosystem exchange (NEE) outputs from the C-TESSEL model for the Paris region : NEE values are  the highest in spring (-10 to -25 kt.hr$^{-1}$ during daytime and + 5 kt.hr$^{-1}$ during nighttime, and a daily mean of -5/-10 kt.yr$^{-1}$ which is the same order of magnitude as fossil fuel emissions i.e. 7 to 9 kt.hr$^{-1}$ in spring), a bit lower in  summer and autumn  and much smaller in winter (-3 kt.hr$^{-1}$ during daytime and +2 kt.hr$^{-1}$ during nighttime, and a daily mean of -1 kt.hr$^{-1}$, which is much smaller than fossil fuel emissions that reach 10 kt.hr$^{-1}$ in winter). In the Supplementary material S43  we give for each site the annual and seasonal averages of the daily minimum and of the daily maximum of the hourly concentration, along with the annual and seasonal averages of the diurnal cycle amplitude (max-min concentration difference). The lines entitled "variation" give the mean of the hourly 1-σ standard deviation of the min and of the max of each diurnal cycle.

It is noticeable that the mean winter concentration is about 6 ppm higher at MON than in TRN50. Both stations are in rural environment, but MON is closer to Paris than TRN. As the signals are quite similar in summer, this difference can not likely be explained by the biospheric activity, and is more probably partly due to a higher anthropogenic influence in MON. However, we need here to take into account the difference of the stations inlet height (9 m AGL at MON, 50 m AGL at TRN50) : as shown in Schmidt et al (2014) for the 2010 winter season at Trainou, during daytime $CO_2$ concentration measured at 10 m AGL and 50 m AGL are similar, but this is not the case during nighttime when the $CO_2$ concentration is about 3 ppm higher at 10 m AGL than at 50 m AGL because atmospheric mixing is not existent at night and $CO_2$ sources accumulate near the surface (Denning et al, 1995). This means that the difference between MON and TRN at the inlet height of MON is of the order of 6 ppm during daytime and twice as low during nighttime. This is consistent with the hypothesis of a higher impact of anthropogenic emissions in MON than in TRN, that according to AIRPARIF are lower during nighttime than during daytime, although we do not observe the same order of magnitude (AIRPARIF gives a ratio of daytime over nighttime emissions equals to 3 to 4 in wintertime, while we observe a ratio of 2 ; see Fig.3 in Bréon et al, 2015). Remember though that the diurnal cycle of the emissions inventory is an average for the whole IdF region, and not only for the MON area. This impact from local sources and/or  the $CO_2$ emission plume of the Paris megacity  on MON will be further inferred from the wind analysis in section 3.5.

The influence of  urban emissions in GIF, MON and GON results in a higher mean diurnal concentration of atmospheric $CO_2$ at these sites compared to the others for all seasons (and mainly in winter) and of its variability. The impact of traffic emissions is well visible in GIF, MON and GON in  the winter season only with two $CO_2$ maxima during rush hours (morning and evening). Although traffic occurs throughout the year, these peaks are likely more or less masked by the biospheric activity and the ABLH dynamics during the other seasons (see above). In addition, the ABL is shallower during winter leading to higher $CO_2$ concentrations. The amplitude of the morning and evening peaks is higher in GON than in GIF and MON and denotes a stronger impact of traffic emissions in GON than in the two other stations. GON also shows the maximum inter seasonal difference between summer and winter (31.3 ppm in the afternoon) which is higher than the mean annual afternoon dispersion, meaning in other terms that the seasonal variability is higher than the mean annual dispersion of the fluxes in the afternoon. 
[revised manuscript text omitted]
 317 m AGL (and likely higher). Quantifying such vertical gradients is of interest since they have to be correctly reproduced in urban mesoscale modeling frameworks for accurate atmospheric $CO_2$ inversion purposes. This vertical gradient can be roughly estimated by subtracting the EIF signal from the GON or the GIF signal. In the early morning (4-5 h ) the GON-EIF (respectively GIF-EIF) gradient is +35 ppm (+18 ppm) in spring, +31 ppm (+17 ppm) in summer, +30 ppm (+10 ppm) in autumn, and +14 ppm (+4ppm) in winter. In the afternoon (14-16 h), the GON-EIF (respectively GIF-EIF) gradient is lower in absolute values and changes of sign: -7 ppm (-8 ppm) in spring, -4 ppm (-3 ppm) in summer, -4 ppm (-7 ppm) in autumn and -2 ppm (-5 ppm) in winter. The gradient is thus at its maximum at night and in the warm seasons, which may also reflect the influence of the biospheric respiration at the stations close to the ground level, compared to EIF. In the future, we plan to equipy the Eiffel tower with two supplementary levels of sampling to collect observations that will allow us to well characterize the $CO_2$ vertical profile over the Paris city and its temporal variability, and its relation with ground emissions variations and their coupling with atmospheric dynamics.

**3.3 Weekday versus weekend**

According to the AIRPARIF inventory, the total $CO_2$ emissions of IdF are lower during weekends than during weekdays, with mean differences of the order of 30-40% during daytime and 50-60% during nighttime. We infer here the impact of such variations on the atmospheric concentrations. In Fig. 7, we show the mean diurnal cycles of the $CO_2$ concentrations at each site for each day of the week, as well as the associated standard deviation (1-$\sigma$).

In GON, the $CO_2$ concentrations are systematically lower over the weekend , especially on Sundays (5-10% of decrease during daytime, 25-35% of decrease during nighttime). A similar pattern is observed for MON. The weekdays-to-weekend ratios observed for the $CO_2$ concentrations are lower than those computed from the emissions given by the inventories. This could be due to an overestimation of the difference from the inventory ; however, biospheric fluxes (eg Schmidt et al, 2015), wind speed and direction (see section 3.5) and $CO_2$ background signals (see section 3.1, and Turnbull et al, 2015) are also factors that modulate the observed $CO_2$ concentration at each site. Disentangling the role of each of these factors on the differences between the observed weekdays-to-weekend $CO_2$ concentration ratios versus the ones calculated from the inventory would require a dedicated analysis that is outside the scope of this paper. Note that while the variability of the $CO_2$ means is very large in GON, it is lower during weekends than during weekdays. The $CO_2$ diurnal cycle does not change  much in GIF between a working weekday and a weekend (except for a small decrease during nighttime over the weekend), nor at EIF and TRN, possibly because of a larger influence of the biospheric fluxes (that do not

depend on weekday or weekends) at these stations compared to the contribution of anthropogenic emissions (that are different on weekdays and weekends according to AIRPARIF, see Fig. 4 in Bréon et al, 2015) and that are the strongest observed at GON (sections 3.2.1 and 3.5.2). DAnd, during nighttime at GIF we observed the highest concentrations from Sundays to Wednesdays, with concentrations lower by 3-5 ppm (a 20-25% decrease) from Thursdays to Saturdays. This could be due to a specific traffic pattern within the footprint of the station, but we currently do not have access to local traffic data for each day of the week to verify this hypothesis.

**3.4 $CO_2$ seasonal cycle**

We computed the seasonal cycle of $CO_2$ at each site, based on the monthly means of our ~1 year datasets and including all hours of the day (Fig. 89a). The seasonal cycles of the air temperature and available LBLH data (at QUA) are also shown on the same figure.

Ignoring the specific case of EIF (section 3.2.3), throughout the year we observe that the monthly mean $CO_2$ concentration increases with the vicinity of the station to larger $CO_2$ emission sources. The maximum maximum $CO_2$ enhancementgradient compared towith MHD is observed at GON which is our most anthropogenically influenced station (from 6.8 ppm in July to 27.5 ppm in December). Similarly to what is observed at the diurnal scale (section 3.21), differences of several ppm are also observed abetweent our rural sites the seasonal scale between the continental stations and MHD, while the differences between the rural/peri-urban/urban stations in IdF is of the same order of magnitude. These differences of concentration between the stations located in IdF and MHD vary with the season, the seasonal cycle being much more well defined in the Paris rural stations than in MHD due to a higher biospheric activity in the IdF region than on the western coast of Ireland. This implies that background values of $CO_2$ in IdF (i.e. without the impact of Paris emissions) should be defined at the regional scale near Paris (~100 km) and not at the continental scale in MHD. Furthermore, in Section 3.1 we explained that the $CO_2$ concentration fluctuates with the origin of the airmasses that can be much variable, and therefore, specific regional background should be selected in function of the wind direction, as also mentioned for the case of Indianapolis (Turnbull et al, 2015). In conclusionThus, MHD appears not to be relevant as a background site for studying defining the atmospheric plume of $CO_2$ in the Paris region at the seasonal scale as well. Regional background stations (~100 km) seem to be much better suited for urban regional studies in Paris and elsewhere in the European continent. Several methods are available to extract a background signal from a timeseries (e.g. Ruckstuhl et al, 2012 ; Ammoura et al, 2016). Quantifying precisely the Paris background signals values as well as the Paris plume and its variability requires a dedicated analysis that is outside the scope of the present paper : it will specifically adressed within another dedicated study.

At each station, the monthly mean $CO_2$ concentration follows a seasonal cycle that reaches its maximum in winter and its minimum in summer. This is expected due to: 1/ the seasonal cycle of the biosphere; 2/ the variability of anthropogenic emissions, mainly from the heating sector, which are directly linked to ambient temperature (see 3.2.2); and 3/ the seasonal cycle of the ABL height (section 3.2.3), which is at the lowest in wintertime (e.g. Denning et al, 1995 ; Turnbull et al, 2009).


[revised manuscript text omitted]

Inner Paris extends to a diameter of 10 km , while the Paris  metropolitan area extends to a diameter of 30 to 50 km. The distance of the peri-urban stations GON and GIF to the Paris inner city is about 10 km and 15 km, respectively. The distance of the rural stations MON and TRN to inner Paris is about 30 and 100 km, respectively. Taking into account these distances, we set the hypothesis that we can assess the influence of local emissions using hourly means observed in low wind speed conditions (less than 3 m s$^{-1}$) while the influence of remote emissions can be analyzed using data recorded in relatively high wind speed conditions (more than 8 m s$^{-1}$). This relies on considering the time given for atmospheric mixing of local and regional emissions (dominant at low to mid windspeeds) versus their ventilation (dominant at high windspeeds) : the integration of local and regional emissions into an air mass, which carries the signature of remote emissions when it is upwind of Paris, gets higher with decreasing windspeeds. For example, for windspeeds lower than 3 m.s$^{-1}$ (11 km.h$^{-1}$), it takes one hour or more for any airmass to flow over the center of Paris (~10 km of diameter), allowing some time for local emissions to get mixed into the airmass, while at 8 m.s$^{-1}$ or more (~29 km.h$^{-1}$) it takes about 20

minutes or less, allowing less time for the atmospheric integration of local to regional emissions. In the middle range of windspeed (3-8 m s$^{-1}$), we expect most of the $CO_2$ variability to be driven by the influence of the regional emissions coming from Paris.

For all of the regional stations, Fig. 11 shows the pollution roses of the mean afternoon $CO_2$ concentration binned by wind speed (ws) and wind direction (wd) with a resolution of 1 m s$^{-1}$ for ws and 10° for wd. We use here the $CO_2$ hourly concentration dataset that has been  seasonally adjusted (section 3.5.1). In order to assess the representativeness of each (ws, wd) bin, the contribution of each concentration mean for a given (ws, wd) bin on the total

10 concentration is also calculated, after applying a square root transformation on the $CO_2$ concentration to reduce any bias from the highest $CO_2$ values (we used the polarFreq function from the OpenAir workpackage for R with the option "weighted mean" – more information can be found online here : http://www.openair-project.org/PDF/OpenAir_Manual.pdf). We also show the mean 1-σ standard deviation of the $CO_2$ concentration at each bin. A similar figure for nighttime data is given in the supplementary material S6a. During daytime (nighttime), the color scale is limited to the 380-430 ppm interval

[revised manuscript text omitted]

In sections 3.2.1 and 3.4, we questioned the influence of local sources on GON (such as CDG and Le Bourget airports, but also of point sources mentioned in section 2.1.2 and diffuse sources around the station).

As for MON, all these types of sources in the vicinity of GON will likely influence it at low windspeed.  GON is also exposed to aircraft emissions as it lies close to the lowest flight paths (0-1000m AGL) from the CDG and Le Bourget airports (http://www.advocnar.fr/Fluxdetrajectoires.html). These emissions are due: (i) in the NW sector, to takeoffs from the CDG northern runway; (ii) in the SW sector, to takeoffs from the CDG southern runway and from Le Bourget runway; (iii) in the NE sector, to landing on both CDG runways; and (iv) in the SE sector, to landings on the southern runway of CDG and to a lesser extent on Le Bourget airport. Also, it is likely that GON gets exposed to emissions from the two airports themselves, located a few km away. Note that the standard deviation which is more that  1 ppm higher from 60° (NE) to 170° (SE) seems to indicate fresher emissions in this wind sector. Nearby highways (located about 1.2 km north and east) could contribute in these wind directions. Discriminating between the different emission sources influencing the GON or the MON stations at low windspeed would require dedicated fine scale modeling studies that are outside the scope of this study.

At EIF, the influence of local emissions is expected mostly between the late morning and the late afternoon since, as we have seen in section 3.2.2, the top of the Eiffel tower receives surface emissions in this time period during all seasons. The $CO_2$ pollution rose of Fig. 11 indicates high concentrations (400 ppm to more than 430 ppm) in all directions around the stations for wind speeds comprised between 0 and 2 m s$^{-1}$. The variability is quite large (1.5 to 5 ppm) indicating fresh emissions and reflecting the spatial and temporal variability of the emissions coupled to atmospheric transport variations. Carbon isotopes and $CO_2$ co-emitted species measurements would be useful here to estimate the role of the different emission sectors (ex. Lopez et al, 2013).

[revised manuscript text omitted]

10  tower with two additional sampling heights to gather vertical $CO_2$ profiles and associated meteorological data : this will be of great help to understand the coupling between $CO_2$ sources and atmospheric dynamics over the Paris megacity in the future. This recalls as well that the altitude relative to ground level and the distance to the emissions of a station are very important factors to take into account in the network capacity to properly detect a $CO_2$ urban plume (see also the discussion about this topic in Boon et al., 2015).

15  About gaining lessons on urban $CO_2$ network design, with 13 observation towers located in and just around the city, the Indianapolis network is a good example to follow (see Turnbull et al, 2015) - as long as the budget allow it - that fulfills the urban network constrains we inferred from our analysis in Paris. Longer prospects on the Paris network design with cheaper sensors are discussed in the study of Wu et al (2016). Note that these lessons are appropriate to cities having a flat continental topography. The situation would be different for coastal or mountain/valley cities, where complex meteorological

20  features occur (breezes, katabatic winds, thermal inversion…).

The fine classification of the $CO_2$ concentrations collected at each site following wind directions and wind speeds allowed us to better define the footprint of each station and the impact of local, regional and remote $CO_2$ fluxes on each station. In each of the regional sites, the high $CO_2$ concentrations observed at low wind speeds (<3 m s$^{-1}$) revealed the impact of local

25  sources including likely emissions from aircraft and airports, cement plants and thermal plants. For moderate wind speeds (3 to 9 m s$^{-1}$), the impact of the $CO_2$ emissions of Paris is clearly seen at urban and peri-urban stations (GON, EIF and GIF) in the afternoon, and much less at night. This impact however is barely seen in the two rural stations (MON and TRN), and ultimately do not seem to be relevant sites to study the $CO_2$ emission plume from the Paris megacity.

At each station, the minimum of the seasonal cycle amplitude was found in summer due to high photosynthesis, lower

30  anthropogenic emissions and higher ABL height. The maximum of the $CO_2$ seasonal cycle was found in winter when the biospheric activity reaches its minimum, the Paris anthropogenic emissions get to their maximum and the ABL height is at its lowest. However, we could not separate the anthropogenic and biospheric $CO_2$ signals, nor the role of the different emission sectors. This highlights the need for regular carbon isotopic measurements of $CO_2$ at the regional network stations, together with measurements of anthropogenic co-emitted species such as CO, NOx, black carbon and volatile organic

compounds (e.g. Lopez et al., 2013; Ammoura et al., 2014; Ammoura et al., 2015). Finally, we show that ancillary data such as local meteorological data and parameters defining the structure of the atmosphere such as the ABL height are very important to understand the observed $CO_2$ variability. Ideally, such measurements should also be included in the development of future urban $CO_2$ monitoring networks.

**5 Data availability**

The $CO_2$-Megaparis datasets are available from the AERIS/ESPRI data center via the following secure FTP link: http://cds-espri.ipsl.fr/espri/pubipsl/co2-megaparis/ftp.html upon simple request to the first author. The ICOS datasets are available 10 from the ICOS database at LSCE. Please contact the first author for further information (irene.remy-xueref@univ-amu.fr).

**Supplement link**

**Acknowledgments**

This work was mostly funded by the Agence Nationale de la Recherche (ANR) in the framework of the $CO_2$-Megaparis 15 project and partly by the Ville de Paris through the "Le $CO_2$ parisien" (Paris 2030) project. We deeply acknowledge AIRPARIF technical team for the maintenance of the $CO_2$-Megaparis stations. The authors are very grateful to the RAMCES-ICOS team and they also thank Sandip Pal for technical help. The GIF and TRN stations are funded by INSU and CEA (SNO RAMCES/ICOS). Most of the figures shown in this work were produced with the Openair package for R (Carslaw and Ropkins, 2012; Carslaw, 2015). We especially acknowledge David Carslaw (Openair package) for helpful 20 advices. We thank very much SPR at CEA Saclay for providing us with the meteorological measurements. The first author sends warm acknowledgments to Peter Rayner and Thomas Lauvaux for their scientific advices in building-up the $CO_2$-Megaparis project, and to Cecilia Garrec and Peter Rayner (once again) for their help in coordinating it. Special thanks to Steve Wofsy for his support and to Chris Rella from the Picarro company for his help with the CRDS analyzers.

5  **Figure captions**

10  Figure 1. Annual emissions of $CO_2$ from Île-de-France at a spatial resolution of 1x1 km$^2$ (AIRPARIF, 2010) and our Paris megacity $CO_2$ in-situ network: the red points indicate the $CO_2$-MEGAPARIS stations (MON = NE rural site, 9 m AGL, GON = NE peri-urban site, 4 m AGL and EIF = urban site, 317 m AGL); the dark blue points are stations from the ICOS-France network (GIF = SW peri-urban site, 7 m AGL, TRN = SW rural site, 50 & 180 m AGL). The QUALAIR station for monitoring the atmospheric boundary layer height in the Paris city is also shown (green point).

Figure 2. Location of the Paris megacity on a map of $CO_2$ anthropogenic emissions from Western Europe, adapted from the Edgar 2009 inventory (http://edgar.jrc.ec.europa.eu/, 2009). Emissions are given in Tg of $CO_2$-eq per grid cell (10 x 10 km$^2$). Some of the main emitting points in Western Europe are also given. The geographical position of the remote site of Mace Head (MHD) on the west coast of Ireland is also shown.

20  Figure 3. Wind rose at GIF given by season over the period of study (8 August 2010–13 July 2011) from the Meso-NH wind fields. Colors indicate the wind speed according to the given scale (in m s$^{-1}$).

Figure 3. Wind rose at GIF (7 m AGL, SW peri-urban site) given by season over the period of study (8 August 2010–13 July 2011) from the Meso-NH modeled wind fields. Colors indicate the wind speed according to the given scale (in m s$^{-1}$).

Figure 4. Seasonal variation of the temperature at SAC (100 m AGL) close to the GIF station (hourly averages) on the period 25  of study (8 August 2010 – 13 July 2011).

Figure 4a. Time series of $CO_2$ concentration (1 hour averages) recorded during the $CO_2$-Megaparis period and colored by wind classed for sites MON (NE rural site, 9 m AGL), GON (peri-urban site, 4 m AGL), EIF (urban site, 317 m AGL) and GIF (SW peri-urban site, 7 m AGL).

Figure 4b. Time series of $CO_2$ concentration (1 hour averages) recorded during the $CO_2$-Megaparis period and colored by wind classed for sites TRN50 (rural SW, 50 m  AGL) and MHD (coastal remote site, 15 m AGL).

Figure 5. Mean $CO_2$ diurnal cycles at the different sites of the Paris regional network and MHD averaged on the whole period of study (8 August 2010–13 July 2011) and computed from hourly $CO_2$ concentrations.

Figure 6 (a to d'). Left: Diurnal cycles of $CO_2$ from 1 h averages at (a) MON, (b) GON, (c) EIF  and (d) GIF. Right: Diurnal cycles of $CO_2$ by season at (a') MON, (b') GON, (c') EIF and (d') GIF. Note that the left and right plot scales are not the same.

Figure 5 (a to d'). Left:  Diurnal cycles of $CO_2$ from 1 h averages at (a) MON (NE rural site, 9 m AGL), (b) GON (NE peri-urban site, 4 m AGL), (c) EIF (urban site, 317 m AGL)  and (d) GIF (SW peri-urban site, 7 m AGL). Right: Diurnal cycles of $CO_2$ by season at (a') MON, (b') GON, (c') EIF and (d') GIF. Note that the left and right plot scales are not the same.

Figure 6 (e to g'). Left: Diurnal cycles of $CO_2$ from 1 h averages at: (e) TRN50, (f) TRN180 and (g) MHD. Right: Diurnal cycles of $CO_2$ by season at: (e') TRN50, (f') TRN180 and (g') MHD. Note that the left and right plot scales are not the same.

Figure 5 (e to g'). Left: Diurnal cycles of $CO_2$ from 1 h averages at: (e) TRN50 (SW rural site, 50 m AGL), (f) TRN180 (SW rural site, 180 m AGL) and (g) MHD (remote site, 15 m AGL). Right: Diurnal cycles of $CO_2$ by season at: (e') TRN50, (f') TRN180 and (g') MHD. Note that the left and right plot scales are not the same.

Figure 7. Diurnal cycles of the hourly LBLH estimate means (±1 σ) and of $CO_2$ hourly means observed by season at QUALAIR and EIF, respectively. Time is in hour UTC. The horizontal line is the elevation of EIF. The violet circles give the $CO_2$ concentration (according to the red scale) at the moments when the LBLH was measured as well.

Figure 6. Diurnal cycles of the hourly LBLH (Lower BLH) estimate means (in black) ±1-σ standard deviation (in grey) and of the $CO_2$ hourly means (in red) observed by season at QUALAIR (urban site, 25 m AGL) and EIF (urban site, 317 m AGL), respectively. Time is in hour UTC. The blue horizontal line is the elevation of EIF. The violet circles give the $CO_2$ concentration (according to the red scale) at the same moments when the LBLH (in black) was measured.

Figure 7. Left: $CO_2$ diurnal cycle by day of the week at the different stations, calculated from $CO_2$ hourly concentrations over the whole period of study. Right: standard variation (1-σ) of the hourly $CO_2$ mean concentration.

Figure 8a. Seasonal cycles of $CO_2$ concentration at the six sites based on monthly means. Monthly averages of air temperature at 100 m (Saclay tower near GIF) and of the LBLH (QUALAIR urban site, 25 m AGL) are also shown. Memo (in m AGL) : MON = 9 m (NE rural site), GON = 4 m (NE peri-urban site), EIF = 317 m (urban site), GIF = 7 m (SW peri-urban site), TRN50 = 50 m (SW rural site), TRN180 = 180 m (SW rural site), MHD = 15 m (remote site).

Figure 8b. Seasonal cycle (Aug.2010-Jul.2011) of $CO_2$ at each of the Paris regional sites and at MHD, calculated from $CO_2$ monthly means of hourly averages, with error bars showing one standard deviation (±1-σ) of the $CO_2$ means. Memo (in m AGL) : MON = 9 m (NE rural site), GON = 4 m (NE peri-urban site), EIF = 317 m (urban site), GIF = 7 m (SW peri-urban site), TRN50 = 50 m (SW rural site), TRN180 = 180 m (SW rural site), MHD = 15 m (remote site).

Figure 9. Left: Hourly means of the $CO_2$ concentration recorded at GON (NE peri-urban site, 4 m AGL) as a function of wind speed and colored by wind direction (the color scale is in degrees). Right: same for the $CO_2$ standard deviation (1-σ of the hourly $CO_2$ concentration means).

Figure 10. Mean $CO_2$ concentration (in ppm) observed at the different stations of the Paris regional network (TRN represents the measurements at 50 m AGL) and at MHD for wind speed higher than 9 m s$^{-1}$ over the period of study (8 August 2010–13 July 2011). During such events, the synoptic conditions were mostly oceanic (wind blowing from the SW sector). Memo (in m AGL) : MON = 9 m (NE rural site), GON = 4 m (NE peri-urban site), EIF = 317 m (urban site), GIF = 7 m (SW peri-urban site), TRN50 = 50 m (SW rural site), TRN180 = 180 m (SW rural site), MHD = 15 m (remote site).

Figure 12. Left: CO₂ mean concentration as a function of wind speed (circles in m s⁻¹) and wind direction at MON, GON, EIF, GIF and TRN50 stations using daytime data (11-15 h UTC) for the whole period of study (4 Aug.2010-11 July 2011). Middle: mean 1-σ CO₂ variability of each concentration (ws, wd) point. Right: occurrence as the frequency of the (ws, wd) bin weighted by the square-root of the CO₂ concentration mean.

Figure 11. Left: $CO_2$ mean concentration as a function of wind speed (circles in m s$^{-1}$) and wind direction at MON (NE rural), GON (NE peri-urban), EIF (urban), GIF (SW peri-urban) and TRN50 (rural) stations using daytime data (11-15 h UTC) for the period of study (4 Aug.2010-11 July 2011). Middle: mean 1-$\sigma$ $CO_2$ variability of each concentration (ws, wd) point. Right: occurrence as the frequency of the (ws, wd) bin weighted by the square-root of the $CO_2$ concentration mean.

Seasonal variation of the temperature at SAC (100 m AGL) close to the GIF station (hourly averages) on the period of study (8 August 2010–13 July 2011).

**Tables**

Table 1. Coordinates of the stations used in this study (ASL stands for Above Sea Level; AGL for Above Ground Level).

| Station | Code | Latitude (°) | Longitude (°) | Site ground elevation ASL | Sampling height AGL |
|---------|------|--------------|---------------|---------------------------|---------------------|
| Montgé-en-Goële | MON | 49°01'41.79'' N | 2°44′55.54'' E | 160 m | 9 m |
| Gonesse | GON | 48°59'24.56'' N | 2°27'21.90'' E | 68 m | 4 m |
| Eiffel tower | EIF | 48°51'29.71'' N | 2°17'39.92'' E | 33 m | 317 m |
| Gif-sur-Yvette | GIF | 48°42'35.82'' N | 2°08'51.55'' E | 163 m | 7 m |
| Traînou | TRN | 47°57'53.08'' N | 2°06'45.42'' E | 133 m | 50 m , 180m |
| Mace Head | MHD | 53°19'33.00" N | 9°54'12.00" W | 25 m | 15 m |
| QUALAIR | QUA | 48°50'47.26" N | 2°21'21.40" E | 35 m | 25 m |

Table 2. Calibration and target frequencies, accuracy and repeatability of the $CO_2$-Megaparis stations. The accuracy is given as the difference of the target $CO_2$ concentrations measured by the CRDS analyzer and by the GC.

|                      | EIF                | MON               | GON               |
|----------------------|--------------------|-------------------|-------------------|
| Calibration sequence | 2 h every 3 months | 6 h every 2 weeks | 6 h every 2 weeks |
| Target sequence      | 30 mn every 2 weeks | 30 mn every 12 h | 30 mn every 12 h |
| Accuracy (ppm)       | 0.13               | -0.04             | -0.07             |
| Repeatability (ppm)  | 0.38               | 0.10              | 0.07              |

Table 3 Monthly means and standard deviation (± 1-σ) of the $CO_2$ concentration (in ppm) measured at each site and data coverage of each month (Coverage, in percent).

| | MON | GON | EIF | GIF | TRN50 | TRN180 | MHD |
|---|---|---|---|---|---|---|---|
| | | | | Spring | | | |
| March | 410.4±9.4 | 420.3±19.1 | 411.8 ±16.7 | 414.4±13.7 | 408.9±9.3 | 405.5±7.9 | 398.6±4.4 |
| Coverage | 99.9 | 97.3 | 95.6 | 93.0 | 57.7 | 66.8 | 87.6 |
| April | 402.1±11.0 | 421.2±32.6 | 403.0±13.2 | 408.7±15.3 | 401.3±11.2 | 396.8±7.1 | 398.6±4.9 |
| Coverage | 100.0 | 95.3 | 94.6 | 94.2 | 69.0 | 79.6 | 77.6 |
| May | 394.7±8.9 | 405.5±20.0 | 398.0±10.6 | 398.7±11.2 | 395.0±9.9 | 391.2±5.9 | 396.3±2.4 |
| Coverage | 99.9 | 97.3 | 98.8 | 98.3 | 81.2 | 82.8 | 95.6 |
| | | | | Summer | | | |
| June | 400.1±11.9 | 406.2±27.3 | 396.9±8.2 | 400.9±12.8 | 398.4±10.7 | 394.5±4.7 | 394.5±3.5 |
| Coverage | 98.1 | 0.65 | 95.3 | 84.9 | 88.2 | 69.3 | 92.9 |
| July | 393.1±6.9 | 398.6±17.3 | 393.4±6.6 | 397.2±8.3 | 392.4±6.2 | 389.8±3.2 | 392.1±5.0 |
| Coverage | 96.8 | 96.8 | 78.1 | 62.4 | 51.4 | 78.1 | 97.1 |

| | | | | | | | |
|---|---|---|---|---|---|---|---|
| August | 390.8±10.2 | 401.9±29.6 | 387.1± 7.9 | 392.2±11.8 | 389.8±10.8 | 384.9±5.6 | 381.4±2.5 |
| Coverage | 99.6 | 94.6 | 90.5 | 78.6 | 95.8 | 96.1 | 99.9 |

**Autumn**

| | | | | | | | |
|---|---|---|---|---|---|---|---|
| September | 395.3±12.7 | 410.9±34.0 | 391.0±11.1 | 395.3±11.1 | 392.5±11.8 | 385.7±5.7 | 384.0±3.3 |
| Coverage | 72.9 | 96.0 | 97.8 | 83.1 | 91.1 | 90.4 | 96.8 |
| October | 402.8±9.8 | 413.9±24.7 | 400.8±12.0 | 403.0±11.3 | 400.3±10.6 | 395.0±7.2 | 390.9±6.2 |
| Coverage | 100.0 | 96.0 | 98.9 | 82.7 | 92.5 | 90.5 | 98.7 |
| November | 408.3±10.4 | 414.9±15.9 | 407.7±15.1 | 411.2±12.9 | 401.8±9.4 | 399.3±8.6 | 393.6±3.8 |
| Coverage | 100.0 | 97.2 | 99.6 | 67.4 | 34.3 | 31.5 | 97.1 |

**Winter**

| | | | | | | | |
|---|---|---|---|---|---|---|---|
| December | 417.0±13.9 | 424.5±17.9 | 414.2±16.9 | 415.4±13.9 | 408.3±9.5 | 406.0±10.4 | 396.8±3.8 |
| Coverage | 100.0 | 73.9 | 71.9 | 77.4 | 82.4 | 87.5 | 97.2 |
| January | 408.9±9.4 | 415.8±16.7 | 408.4±13.2 | 410.1±13.0 | 405.7±10.1 | 403.1±9.3 | 396.1±2.3 |
| Coverage | 100.0 | 96.2 | 78.9 | 78.5 | 95.6 | 94.5 | 98.7 |
| February | 411.9±12.2 | 423.1±20.7 | 410.5±14.7 | 409.8±10.5 | 405.4±7.8 | 402.8±7.3 | 396.3±2.0 |
| Coverage | 100.0 | 97.0 | 93.2 | 97.0 | 84.8 | 88.5 | 98.4 |

Table 4. Mean altitude of the lowest estimate of the boundary layer height (LBLH) by season in the morning and early afternoon (hours are given UTC, altitude in meters AGL). The number of points used to calculate the means are also given (N).

| Time (UTC) | 5 h | 6 h | 7 h | 8 h | 9 h | 10 h | 11 h | 12 h | 13 h |
|---|---|---|---|---|---|---|---|---|---|
| Spring | | | | | | | | | |
| LBLH | NaN | 410 | 442 | 520 | 593 | 697 | 833 | 899 | 935 |
| N | 0 | 9 | 11 | 11 | 12 | 12 | 12 | 13 | 13 |
| Summer | | | | | | | | | |
| LBLH | 513 | 583 | 728 | 992 | 1178 | 1324 | 1400 | 1405 | 1531 |
| N | 7 | 13 | 13 | 13 | 13 | 13 | 11 | 11 | 7 |
| Autumn | | | | | | | | | |
| LBLH | 351 | 394 | 451 | 615 | 751 | 837 | 896 | 947 | 940 |
| N | 16 | 25 | 31 | 34 | 33 | 33 | 33 | 31 | 30 |
| Winter | | | | | | | | | |
| LBLH | NaN | 301 | 349 | 384 | 419 | 440 | 470 | 516 | 550 |
| N | 0 | 3 | 15 | 24 | 23 | 25 | 26 | 27 | 29 |

**Figures**

[Figure]

10

Figure 1. Annual emissions of $CO_2$ from Île-de-France at a spatial resolution of 1x1 km$^2$ (AIRPARIF, 2010) and our Paris
megacity $CO_2$ in-situ network: the red points indicate the $CO_2$-MEGAPARIS stations (MON = NE rural site, 9 m AGL,
GON = NE peri-urban site, 4 m AGL and EIF = urban site, 317 m AGL); the dark blue points are stations from the ICOS-

France network (GIF = SW peri-urban site, 7 m AGL, TRN = SW rural site, 50 & 180 m AGL). The QUALAIR station for monitoring the atmospheric boundary layer height in the Paris city is also shown (green point).

[Figure]

Figure 2. Location of the Paris megacity on a map of $CO_2$ anthropogenic emissions from Western Europe, adapted from the Edgar 2009 inventory (http://edgar.jrc.ec.europa.eu/, 2009). Emissions are given in Tg of $CO_2$-eq per grid cell (10 x 10 km$^2$). Some of the main emitting points in Western Europe are also given. The geographical position of the remote site of Mace Head (MHD) on the west coast of Ireland is also shown.

[Figure]

Frequency of counts by wind direction (%)

Figure 3. Wind rose at GIF (7 m AGL, SW peri-urban site) given by season over the period of study (8 August 2010–13 July 2011) from the Meso-NH modeled wind fields. Colors indicate the wind speed according to the given scale (in m s[-1]).

[Figure]

Figure 4a. Time series of $CO_2$ concentration (1 hour averages) recorded during the $CO_2$-Megaparis period and colored by wind classed for sites MON (NE rural site, 9 m AGL), GON (peri-urban site, 4 m AGL), EIF (urban site, 317 m AGL) and GIF (SW peri-urban site, 7 m AGL).

[Figure]

[Figure]

Figure 4. Seasonal variation of the temperature at SAC (100 m AGL) close to the GIF station (hourly averages) on the period of study (8 August 2010 - 13 July 2011).

[Figure]

Figure 5. Mean $CO_2$ diurnal cycles at the different sites of the Paris regional network and MHD averaged on the whole period of study (8 August 2010–13 July 2011) and computed from hourly $CO_2$ concentrations.

[Figure]

Figure 6 (a to d'). Left: Diurnal cycles of $CO_2$ from 1 h averages at (a) MON, (b) GON, (c) EIF and (d) GIF. Right: Diurnal cycles of $CO_2$ by season at (a') MON, (b') GON, (c') EIF and (d') GIF. Note that the left and right plot scales are not the same.

Figure 5 (a to d'). Left: Diurnal cycles of CO_2 from 1 h averages at (a) MON (NE rural site, 9 m AGL), (b) GON (NE peri-urban site, 4 m AGL), (c) EIF (urban site, 317 m AGL)  and (d) GIF (SW peri-urban site, 7 m AGL). Right: Diurnal cycles of CO_2 by season at (a') MON, (b') GON, (c') EIF and (d') GIF. Note that the left and right plot scales are not the same.

[Figure]

Figure 6 (e to g'). Left: Diurnal cycles of CO_2 from 1 h averages at: (e) TRN50, (f) TRN180 and (g) MHD. Right: Diurnal cycles of CO_2 by season at: (e') TRN50, (f') TRN180 and (g') MHD. Note that the left and right plot scales are not the same.

Figure 5 (e to g'). Left: Diurnal cycles of $CO_2$ from 1 h averages at: (e) TRN50 (SW rural site, 50 m AGL), (f) TRN180 (SW rural site, 180 m AGL) and (g) MHD (remote site, 15 m AGL). Right: Diurnal cycles of $CO_2$ by season at: (e') TRN50, (f') TRN180 and (g') MHD. Note that the left and right plot scales are not the same.

[Figure]

Figure 7. Diurnal cycles of the hourly LBLH estimate means (±1 σ) and of CO₂ hourly means observed by season at QUALAIR and EIF, respectively. Time is in hour UTC. The horizontal line is the elevation of EIF. The violet circles give the CO₂ concentration (according to the red scale) at the moments when the LBLH was measured as well.

Figure 6. Diurnal cycles of the hourly LBLH (Lower BLH) estimate means (in black) ±1-σ standard deviation (in grey) and of the $CO_2$ hourly means (in red) observed by season at QUALAIR (urban site, 25 m AGL) and EIF (urban site, 317 m AGL), respectively. Time is in hour UTC. The blue horizontal line is the elevation of EIF. The violet circles give the $CO_2$ concentration (according to the red scale) at the same moments when the LBLH (in black) was measured.

[Figure]

Figure 8. Left: $CO_2$ diurnal cycle by day of the week at the different stations, calculated from $CO_2$ hourly concentrations over the whole period of study. Right: standard variation (1-σ) of the hourly $CO_2$ mean concentration. Figure 7. Left: $CO_2$

diurnal cycle by day of the week at the different stations, calculated from $CO_2$ hourly concentrations over the whole period of study. Right: standard variation (1-σ) of the hourly $CO_2$ mean concentration.

[Figure]

Figure 9a. Seasonal cycles of CO$_2$ concentration at the six sites based on monthly means. Monthly averages of air temperature at 100 m (Saclay tower near GIF) and of the LBLH (Jussieu) are also shown.

Figure 8a. Seasonal cycles of CO$_2$ concentration at the six sites based on monthly means. Monthly averages of air temperature at 100 m (Saclay tower near GIF) and of the LBLH (QUALAIR urban site, 25 m AGL) are also shown. Memo (in m AGL) : MON = 9 m (NE rural site), GON = 4 m (NE peri-urban site), EIF = 317 m (urban site), GIF = 7 m (SW peri-urban site), TRN50 = 50 m (SW rural site), TRN180 = 180 m (SW rural site), MHD = 15 m (remote site).

[Figure]

Figure 8b. Seasonal cycle (Aug.2010-Jul.2011) of $CO_2$ at each of the Paris regional sites and at MHD, calculated from $CO_2$ monthly means of hourly averages, with error bars showing one standard deviation ($\pm1$-$\sigma$) of the $CO_2$ means. Memo (in m

AGL) : MON = 9 m (NE rural site), GON = 4 m (NE peri-urban site), EIF = 317 m (urban site), GIF = 7 m (SW peri-urban site), TRN50 = 50 m (SW rural site), TRN180 = 180 m (SW rural site), MHD = 15 m (remote site).

Figure 9b. Seasonal cycle (Aug.2010 Jul.2011) of $CO_2$ at each of the Paris regional sites and at MHD, calculated from $CO_2$ monthly means of hourly averages, with error bars showing one standard deviation (±1 σ) of the $CO_2$ means.

[Figure]

Figure 10. Left: Hourly means of the $CO_2$ concentration recorded at GON as a function of wind speed and colored by wind direction (the color scale is in degrees). Right: same for the $CO_2$ standard deviation (1 σ of the hourly $CO_2$ concentration means).

[Figure]

Figure 10. Mean $CO_2$ concentration (in ppm) observed at the different stations of the Paris regional network (TRN represents the measurements at 50 m AGL) and at MHD for wind speed higher than 9 m s$^{-1}$ over the period of study (8 August 2010–13 July 2011). During such events, the synoptic conditions were mostly oceanic (wind blowing from the SW sector). Memo (in m AGL) : MON = 9 m (NE rural site), GON = 4 m (NE peri-urban site), EIF = 317 m (urban site), GIF = 7 m (SW peri-urban site), TRN50 = 50 m (SW rural site), TRN180 = 180 m (SW rural site), MHD = 15 m (remote site).

[Figure]

Figure 11. Left: $CO_2$ mean concentration as a function of wind speed (circles in m s$^{-1}$) and wind direction at MON (NE rural), GON (NE peri-urban), EIF (urban), GIF (SW peri-urban) and TRN50 (rural) stations using daytime data (11-15 h UTC) for the period of study (4 Aug.2010-11 July 2011). Middle: mean 1-σ $CO_2$ variability of each concentration (ws, wd) point. Right: occurrence as the frequency of the (ws, wd) bin weighted by the square-root of the $CO_2$ concentration mean.

[revised manuscript text omitted]